# Adapting to Online Label Shift with Provable Guarantees

**Yong Bai**[1*], **Yu-Jie Zhang**[2,1*], **Peng Zhao**[1], **Masashi Sugiyama**[3,2], **Zhi-Hua Zhou**[1†]

[1] National Key Laboratory for Novel Software Technology, Nanjing University, Nanjing, China
[2] The University of Tokyo, Chiba, Japan
[3] RIKEN AIP, Tokyo, Japan

## Abstract

The standard supervised learning paradigm works effectively when training data shares the same distribution as the upcoming testing samples. However, this stationary assumption is often violated in real-world applications, especially when testing data appear in an online fashion. In this paper, we formulate and investigate the problem of *online label shift* (OLaS): the learner trains an initial model from the labeled offline data and then deploys it to an unlabeled online environment where the underlying label distribution changes over time but the label-conditional density does not. The non-stationarity nature and the lack of supervision make the problem challenging to be tackled. To address the difficulty, we construct a new unbiased risk estimator that utilizes the unlabeled data, which exhibits many benign properties albeit with potential non-convexity. Building upon that, we propose novel online ensemble algorithms to deal with the non-stationarity of the environments. Our approach enjoys optimal *dynamic regret*, indicating that the performance is competitive with a clairvoyant who knows the online environments in hindsight and then chooses the best decision for each round. The obtained dynamic regret bound scales with the intensity and pattern of label distribution shift, hence exhibiting the adaptivity in the OLaS problem. Extensive experiments are conducted to validate the effectiveness and support our theoretical findings.

## 1 Introduction

One of the fundamental challenges for modern machine learning is the distribution shift [1, 2, 3, 4]. The learned model's testing performance would significantly drop when the distribution is different from the initial training distribution. More severely, in many real-world applications, testing data often come in an *online* fashion after deploying the trained model such that the underlying distribution might continuously change over time. Hence, it is necessary to develop learning methods to handle distribution shift in online and open environments [5]. Another practical concern is the *label scarcity* issue in real tasks, particularly those tasks emerging in online scenarios. For example, in the species monitoring task [6, 7], a learned model is deployed to detect species of wild animals. The data consist of received signals from sensors and hence are naturally in the streaming form. The data distribution of upcoming animals will change due to the variety of species across different geographic locations and seasons, and moreover, it is hard to gather the labels of streaming data in time.

Motivated by the above real demand, this paper is concerned with the following problem: how to design an algorithm that can adapt to non-stationary environments with a few labeled data or even unlabeled data observed at every time? In addition to getting empirical performance gain, the overall method is desired to have clear and strong theoretical guarantees. The problem is generally hard due

---

[*]Equal contribution.
[†]Correspondence to: Zhi-Hua Zhou <zhouzh@lamda.nju.edu.cn>

36th Conference on Neural Information Processing Systems (NeurIPS 2022).

to the non-stationarity of online environments and the lack of supervision. As such, we investigate a simplified case with a focus on the specific change on the *label distribution*. We formalize the *online label shift* (OLaS) problem,[3] which consists of two stages, including offline initialization and online adaptation. Specifically, the learner collects labeled samples drawn independently from the initial distribution $\mathcal{D}_0(\mathbf{x}, y)$ where $\mathbf{x}$ and $y$ denote the feature vector and its associated label, and trains an initial model following standard supervised learning methods. Subsequently, she needs to adapt it to an unknown non-stationary environment where the underlying label distributions change over time. Specifically, at time $t$, she receives a few *unlabeled* samples drawn from the current distribution $\mathcal{D}_t(\mathbf{x})$ and uses them to update the model. In OLaS problems, the essential environment change happens on the label distribution $\mathcal{D}_t(y)$ with the conditional $\mathcal{D}(\mathbf{x} \mid y)$ always remaining the same.

The label shift problem has been widely studied in the offline setting [8, 9, 10, 11, 12, 13, 14], but this is less explored in the more challenging online setup. One natural impulse is to handle OLaS by online learning techniques [15], but it is generally non-trivial due to the lack of supervision in the adaptation stage and also the non-stationarity issue. Wu et al. [16] made the first such attempt for OLaS, where they constructed an unbiased risk estimator with the unlabeled data for model assessment and used online gradient descent for model updating. Let $T$ denote the number of rounds. They proved an $\mathcal{O}(\sqrt{T})$ regret bound, which measures the gap between the learner's decision and the best *fixed* decision in hindsight. However, in non-stationary environments, a single decision can hardly perform well all the time, which makes the guarantee less attractive for OLaS problems. Another technical caveat is that their theory relies on a vital assumption of the convexity of risk functions, which was not verified strictly. In fact, this assumption can hardly be satisfied as an operation to take the argument of the maximum is involved in its formulation of the risk estimator.

In this paper, we aim to develop an algorithm for adapting to online label shift with *provable* guarantees. To this end, we first reframe the construction of the unbiased risk estimator via risk rewriting techniques and prove that the estimator still enjoys benign theoretical properties, albeit with a potential *non-convex* behavior. Second, to handle the non-stationarity of the online stream, instead of using traditional regret as the performance measure, we employ *dynamic regret* to guide the algorithm design, which ensures the online algorithm is competitive with a clairvoyant who knows the online functions in hindsight and hence chooses the best decision of each round. To optimize such a strengthened measure, we propose a novel online ensemble algorithm building upon the risk estimator, consisting of a meta-algorithm running over a group of base-learners, each associated with a customized configuration. Our algorithm enjoys an $\mathcal{O}(V_T^{1/3}T^{2/3})$ dynamic regret, where $V_T = \sum_{t=2}^{T} \|\boldsymbol{\mu}_{y_t} - \boldsymbol{\mu}_{y_{t-1}}\|_1$ measures the variation of label distributions, with $\boldsymbol{\mu}_{y_t}$ denoting the vector consisting of the class-prior probabilities at time $t$. Notably, the regret guarantee achieved by our algorithm is *minimax optimal* in terms of both number of rounds and non-stationarity measure, and importantly, our algorithm does not require the unknown class-prior variation $V_T$ as the input.

Furthermore, for many situations where online label shift contains some patterns such as periodicity or gradual change behavior, we present an improved algorithm to exploit such structures and achieve provably better guarantees. The key idea is to leverage historical information to serve as a hint for online updates. We prove an $\mathcal{O}(V_T^{1/3}G_T^{1/3}T^{1/3})$ dynamic regret, where $G_T$ measures the reusability of historical information that is at most $\mathcal{O}(T)$ while could be much smaller in benign environments. As a benefit, the improved algorithm safeguards the $\mathcal{O}(V_T^{1/3}T^{2/3})$ worst-case bound and meanwhile achieves great improvement in easier environments. Extensive experiments are conducted to evaluate our approach, which show the usefulness of meta-base structure in tackling non-stationarity and validate the effectiveness of other adaptive components.

**Technical Contribution.** Our method is not a direct application of existing non-stationary online learning methods [17, 18, 19], but rather requires in-depth technical innovations. First, the potentially *non-convex* risk estimator makes it hard to apply existing techniques of online convex optimization (OCO) [15], but fortunately, we prove the convexity of its expectation such that the OCO framework can still be used (see Remark 2). Second, to optimize the dynamic regret, we employ a meta-base structure to hedge the uncertainty of the *unknown* minimizer of the expected risk function at each round and convert the variation of expected risk minimizers to the intensity of label distribution drifts, a natural non-stationarity measure for the OLaS problem (see Remark 3). Third, earlier study showed adaptive dynamic regret bounds for convex and smooth functions [19], while the smoothness

---

[3]We use OLaS instead of OLS, since OLS often refers to "ordinary least squares".

assumption can hardly be satisfied in our case. We remove such a constraint by introducing an *implicit update*, which could be of independent interest for general OCO purposes (see Remark 4).

## 2    Problem Formulation

We focus on OLaS of multi-class classification with feature space $\mathcal{X} \subseteq \mathbb{R}^d$ and label space $\mathcal{Y} = [K] \triangleq \{1, \ldots, K\}$, where $d$ is the dimension and $K$ is the number of classes. Below we formulate setups of two stages of OLaS (offline initialization and online adaptation).

**Problem Setup.**   In the *offline initialization stage*, the learner collects a number of labeled samples denoted by $S_0 = \{(\mathbf{x}_n, y_n)\}_{n=1}^{N_0}$ drawn from the distribution $\mathcal{D}_0(\mathbf{x}, y)$ and then obtains a well-performed initial model $f_0 : \mathcal{X} \mapsto \mathcal{Y}$. In the *online adaptation stage*, data come in the streaming form *without* labels. The current model is deployed to predict the labels of online data and also to evolve adaptively. Specifically, at each round $t \in [T]$, the learner receives a small number of *unlabeled* data $S_t = \{\mathbf{x}_n\}_{n=1}^{N_t}$ drawn from the distribution $\mathcal{D}_t(\mathbf{x})$. The non-stationary nature indicates $\mathcal{D}_t \neq \mathcal{D}_{t'}$ in general for different $t, t' \in [T]$. OLaS considers the simplified case that essential changes come from label distributions and there are no new classes, formally described in the following condition.

**Assumption 1** (Online Label Shift)**.**  In the online label shift problem, the label distribution $\mathcal{D}_t(y)$ changes over time while the class-conditional distribution $\mathcal{D}_t(\mathbf{x} \,|\, y)$ is identical throughout the process for $t \in \{0, 1, \ldots, T\}$. Moreover, it holds that $\mathcal{D}_0(y) > 0$ for any $y \in \mathcal{Y}$.

**Performance Measure.**   At round $t \in [T]$, the learner uses the information observed so far to make the prediction and also update the model $\mathbf{w}_t \in \mathcal{W}$, where $\mathcal{W}$ is a convex decision set with diameter $\Gamma \triangleq \sup_{\mathbf{w}, \mathbf{w}' \in \mathcal{W}} \|\mathbf{w} - \mathbf{w}'\|_2$. The goal is to ensure that the $t$-round model $\mathbf{w}_t$ generalizes well on the underlying distribution $\mathcal{D}_t$. Thus, the model's quality is evaluated by its risk defined as $R_t(\mathbf{w}) = \mathbb{E}_{(\mathbf{x}, y) \sim \mathcal{D}_t}[\ell(f(\mathbf{w}, \mathbf{x}), y)]$, where $f : \mathcal{W} \times \mathcal{X} \mapsto \mathbb{R}^K$ is the predictive function and $\ell : \mathbb{R}^K \times \mathcal{Y} \mapsto \mathbb{R}$ is any convex surrogate loss for classification such that $\ell(f(\mathbf{w}, \mathbf{x}), y)$ is convex in $\mathbf{w}$. We introduce two constants, $G \triangleq \sup_{(\mathbf{x}, y) \in \mathcal{X} \times \mathcal{Y}, \mathbf{w} \in \mathcal{W}} \|\nabla_\mathbf{w} \ell(f(\mathbf{w}, \mathbf{x}), y)\|_2$ and $B \triangleq \sup_{(\mathbf{x}, y) \in \mathcal{X} \times \mathcal{Y}, \mathbf{w} \in \mathcal{W}} |\ell(f(\mathbf{w}, \mathbf{x}), y)|$, as upper bounds of gradient norm and loss function value.

We use regret to examine the performance of online algorithms. In particular, *dynamic regret* [20, 17] is employed to compete the algorithm's performance with the best response at each round, defined as

$$\mathbf{Reg}_T^\mathbf{d} \triangleq \sum_{t=1}^T R_t(\mathbf{w}_t) - \sum_{t=1}^T \min_{\mathbf{w} \in \mathcal{W}} R_t(\mathbf{w}) = \sum_{t=1}^T R_t(\mathbf{w}_t) - \sum_{t=1}^T R_t(\mathbf{w}_t^*), \tag{1}$$

where $\mathbf{w}_t^* \in \arg\min_{\mathbf{w} \in \mathcal{W}} R_t(\mathbf{w})$ is the model (or one of the models) with the best generalization ability on the distribution $\mathcal{D}_t$. Notably, it is known that a sublinear dynamic regret is impossible in the worst case [17], so an upper bound of dynamic regret is desired to scale with a certain non-stationarity measure. A natural measure for OLaS would be the variation intensity of label distributions.

**Remark 1** (Static regret vs. dynamic regret)**.**  The classic measure for online learning is *static* regret, defined as $\mathbf{Reg}_T^\mathbf{s} = \sum_{t=1}^T R_t(\mathbf{w}_t) - \sum_{t=1}^T R_t(\mathbf{w}^*)$, where $\mathbf{w}^* \in \arg\min_{\mathbf{w} \in \mathcal{W}} \sum_{t=1}^T R_t(\mathbf{w})$ is the best *fixed* model in hindsight. The measure was adopted in the prior work of OLaS [16]. However, the measure is not suitable for OLaS, because it is too optimistic to expect a single fixed model to behave well over the whole process in changing environments. By contrast, minimizing dynamic regret facilitates the online algorithm with more adaptivity and robustness to non-stationary environments.

## 3    Proposed Approach

This section presents our approach for the OLaS problem, including the algorithms and theoretical guarantees. In the following, we respectively address the two central challenges of OLaS: the lack of supervision and the non-stationarity of online environments.

### 3.1    Unbiased Risk Estimator for Online Convex Optimization

OCO is a powerful and versatile framework for online learning problems, which enjoys both practical and theoretical appeals [21, 15]. Online Gradient Descent (OGD) [20] is one of the most fundamental

and powerful algorithms due to its light computational cost and sound regret guarantees. In the OLaS problem, recall that the learner's goal is to obtain a model sequence $\{\mathbf{w}_t\}_{t=1}^T$ enjoying low cumulative risk $\sum_{t=1}^T R_t(\mathbf{w}_t)$. Thus, suppose the model's risk $R_t(\mathbf{w}_t)$ is known at each round; then OGD simply updates the model by $\mathbf{w}_{t+1} = \Pi_{\mathcal{W}}[\mathbf{w}_t - \eta \nabla R_t(\mathbf{w}_t)]$, where $\Pi_{\mathcal{W}}[\cdot]$ denotes the projection onto $\mathcal{W}$ and $\eta > 0$ is the step size. It is well-known that OGD guarantees the regret bound $\sum_{t=1}^T R_t(\mathbf{w}_t) - \min_{\mathbf{w} \in \mathcal{W}} \sum_{t=1}^T R_t(\mathbf{w}) \leq \mathcal{O}(\sqrt{T})$ when risk function $R_t(\mathbf{w})$ is convex and step size is set as $\eta = \Theta(T^{-1/2})$ [15] (see Appendix B.1 for more details).

However, the expected risk function $R_t(\mathbf{w})$ is unknown in the current OLaS setup as it is defined over the underlying joint distribution $\mathcal{D}_t(\mathbf{x}, y)$. More severely, the online environments in the OLaS problem are fully *unlabeled*, which poses great challenges to apply the OCO framework. Indeed, the lack of supervision makes it hard to empirically assess the expected risk, not to mention ensuring the convexity. In the following, we establish an unbiased estimator $\widehat{R}_t(\mathbf{w})$ with unlabeled data $S_t$, which exhibits nice properties such that the OCO framework is still applicable for our purpose.

**Unbiased Estimator under Label Shift.** Inspired by the progress in offline label shift [9, 12, 13], we establish an unbiased risk estimator in OLaS for $R_t(\mathbf{w})$ with unlabeled data $S_t$ and offline data $S_0$ by the risk rewriting technique. To this end, let $\boldsymbol{\mu}_{y_t} \in \Delta_K$ denote the label distribution vector with the $k$-th entry $[\boldsymbol{\mu}_{y_t}]_k \triangleq \mathcal{D}_t(y = k)$, then we have the following decomposition for the true risk:

$$R_t(\mathbf{w}) \triangleq \mathbb{E}_{(\mathbf{x},y)\sim\mathcal{D}_t}[\ell(f(\mathbf{w},\mathbf{x}),y)] = \sum_{k=1}^K [\boldsymbol{\mu}_{y_t}]_k \cdot R_t^k(\mathbf{w}) = \sum_{k=1}^K [\boldsymbol{\mu}_{y_t}]_k \cdot R_0^k(\mathbf{w}), \qquad (2)$$

where $R_t^k(\mathbf{w}) \triangleq \mathbb{E}_{\mathbf{x}\sim\mathcal{D}_t(\mathbf{x}\,|\,y=k)}[\ell(f(\mathbf{w},\mathbf{x}),k)]$ is the risk of the model over the $k$-th label at round $t$. The second equality holds due to the law of total probability, and the third equality is by the label shift assumption that $\mathcal{D}_t(\mathbf{x}\,|\,y) = \mathcal{D}_0(\mathbf{x}\,|\,y)$ for any $t \in [T]$. Since the labeled offline data $S_0$ is always available, one can approximate $R_0^k(\mathbf{w})$ by its empirical version $\widehat{R}_0^k(\mathbf{w})$ with offline data $S_0$, where $\widehat{R}_0^k(\mathbf{w}) \triangleq \frac{1}{|S_0^k|} \sum_{\mathbf{x}_n \in S_0^k} \ell(f(\mathbf{w},\mathbf{x}_n),k)$ and $S_0^k$ is a subset of $S_0$ containing all samples with label $k$.

Therefore, the task is now to estimate the label distribution vector $\boldsymbol{\mu}_{y_t}$. To this end, we employ the Black Box Shift Estimation (BBSE) method [12] to construct an estimator via only offline data $S_0$ and unlabeled data $S_t$. Specifically, we first use the initial offline model $f_0$ to predict over the unlabeled data $S_t$ and get predictive labels $\widehat{y}_t$, and next estimate the label distribution via solving the crucial equation $\boldsymbol{\mu}_{y_t} = C_{f_0}^{-1} \boldsymbol{\mu}_{\widehat{y}_t}$, where $\boldsymbol{\mu}_{\widehat{y}_t} \in \Delta_K$ is the distribution vector of the predictive labels $\widehat{y}_t$ and $C_{f_0} \in \mathbb{R}^{K \times K}$ is the confusion matrix with $[C_{f_0}]_{ij} = \mathbb{E}_{\mathbf{x}\sim\mathcal{D}_0(\mathbf{x}\,|\,y=j)}[\mathbb{1}\{f_0(\mathbf{x}) = i\}]$ being the classification rate that the initial model $f_0$ predicts samples from class $i$ as class $j$. We defer more details to Appendix B.2. As a result, through risk rewriting and prior estimation, we can construct the following estimator for the true risk $R_t(\mathbf{w})$:

$$\widehat{R}_t(\mathbf{w}) = \sum_{k=1}^K [\widehat{C}_{f_0}^{-1} \widehat{\boldsymbol{\mu}}_{\widehat{y}_t}]_k \cdot \widehat{R}_0^k(\mathbf{w}), \qquad (3)$$

where $\widehat{C}_{f_0}$ and $\widehat{\boldsymbol{\mu}}_{\widehat{y}_t}$ are empirical estimators of the confusion matrix and predictive label distribution vector using offline data $S_0$ and unlabeled data $S_t$ only. Our constructed risk estimator enjoys the unbiasedness property, which plays a crucial role in the later algorithm design and theoretical analysis.

**Lemma 1.** *The estimator $\widehat{R}_t(\mathbf{w})$ in Eq. (3) is unbiased to $R_t(\mathbf{w}) = \mathbb{E}_{(\mathbf{x},y)\sim\mathcal{D}_t}[\ell(f(\mathbf{w},\mathbf{x}),y)]$, i.e., $\mathbb{E}_{S_t\sim\mathcal{D}_t}[\widehat{R}_t(\mathbf{w})] = R_t(\mathbf{w})$, for any $\mathbf{w} \in \mathcal{W}$ independent of the dataset $S_t$, provided $C_{f_0}$ is invertible and the offline dataset $S_0$ has sufficient samples such that $\widehat{C}_{f_0} = C_{f_0}$ and $\widehat{R}_0^k(\mathbf{w}) = R_0^k(\mathbf{w})$, $\forall k \in \mathcal{Y}$.*

The proof of Lemma 1 is in Appendix D.1. Note that the sufficient sample assumption is introduced on offline data $S_0$ to simplify the presentation. Indeed, we can further show $|\mathbb{E}_{S_t\sim\mathcal{D}_t}[\widehat{R}_t(\mathbf{w})] - R_t(\mathbf{w})| \leq \mathcal{O}(\sqrt{1/|S_0|})$ with high probability (details in Appendix D.2). Such an additional dependence on $S_0$ also appears in the classical offline label shift [12, 13] and is negligible when a large number of offline data is collected at the initial stage. The requirement is easy to realize and will not trivialize the online adaptation. Another caveat is that storing all the offline data can be burdensome in resource-constrained learning scenarios, then one may use data sketching techniques like corsets [22, 23] or reduced kernel mean embedding [24, 25, 26] to further improve the storage complexity.

**Remark 2** (Non-convexity issue). Our risk estimator $\widehat{R}_t(\mathbf{w})$ can be *non-convex* as the estimated label distribution $[\widehat{C}_{f_0}^{-1}\widehat{\boldsymbol{\mu}}_{\widehat{y}_t}]_k$ might be negative in order to ensure the unbiasedness. Such a non-convex behavior introduces a great challenge for applying the OCO framework. Fortunately, owing to its unbiasedness and the fact that the *expected* risk $R_t(\mathbf{w})$ is indeed convex, we can continue the following algorithm design and theoretical analysis building upon the constructed unbiased estimator.

**OGD with Unbiased Estimator.** Building upon the risk estimator in Eq. (3), we then deploy OGD and obtain our UOGD algorithm (abbreviated for "OGD with unbiased risk estimator"), namely,

$$\mathbf{w}_{t+1} = \Pi_{\mathcal{W}}[\mathbf{w}_t - \eta\nabla\widehat{R}_t(\mathbf{w}_t)]. \tag{4}$$

Despite the potential non-convexity of the risk estimator itself, we can still establish solid regret guarantees via the OCO framework due to the benign property that the risk estimator is unbiased and the expected risk is indeed convex. For example, UOGD provably enjoys an $\mathcal{O}(\sqrt{T})$ static regret. See the formal statement in Appendix D.3. We remark that our attained static regret already achieves the state-of-the-art theoretical understanding of OLaS, in the sense that previously the same bound can be only achieved with an additional unrealistic convexity assumption imposed over the algorithm [16], which UOGD does not require. Concretely, Wu et al. [16] assumed that *the risk estimator is convex* (in expectation), which is hard to theoretically verify since the estimator approximates the $0/1$-loss and involves an indicator function and an argmax operation due to the employed reweighting mechanism. By contrast, our estimator directly approximates the surrogate loss without reweighting and thus does not suffer from such limitations. Even if modifying their estimator to optimize a surrogate loss, the reweighting mechanism makes their method still hardly suitable for the OCO framework. More details are in Appendix C.3. In a nutshell, our constructed risk estimator enjoys nice properties, which are indispensable for the algorithm design and theoretical analysis.

## 3.2   Adapting to Non-stationarity of Online Label Shift

So far, an $\mathcal{O}(\sqrt{T})$ static regret has been established for OLaS; however, the guarantee is not appealing because static regret is not suitable for non-stationary online problems as discussed in Remark 1. We now introduce our method adapting to the non-stationarity with provable dynamic regret guarantees.

First, benefiting from the unbiasedness and expected convexity of our risk estimator, we prove that UOGD achieves a dynamic regret scaling with the label distribution drift $V_T$.

**Theorem 1.** *Under the same assumptions as Lemma 1, UOGD in Eq. (4) with step size $\eta$ satisfies*

$$\mathbb{E}\big[\,\mathbf{Reg}_T^{\mathsf{d}}\,\big] \leq 2(KG^2/\sigma^2 + B^2)\eta T + \Gamma^2/\eta + 4(\Gamma+1)\sqrt{BV_TT/\eta} = \mathcal{O}\Big(\eta T + 1/\eta + \sqrt{(V_TT)/\eta}\Big), \tag{5}$$

*where the constant $\sigma > 0$ denotes the minimum singular value of the invertible confusion matrix $C_{f_0}$. Moreover, $V_T = \sum_{t=2}^{T}\|\boldsymbol{\mu}_{y_t} - \boldsymbol{\mu}_{y_{t-1}}\|_1$ measures the intensity of the label distribution shift.*

The proof of Theorem 1 can be found in Appendix E.1. The dynamic regret guarantee is obtained in a non-trivial way, and below we expand the technical innovations.

**Remark 3** (Non-stationarity measure for OLaS). For readers who are familiar with the literature, our result is reminiscent of the existing dynamic regret bound in the OCO studies [17, 27] on the surface; however, our result exhibits fundamental differences. The key caveat is that in our case the comparator $\mathbf{w}_t^*$ at each round is *not* the minimizer of the online function. Specifically, as the expected risk $R_t$ is inaccessible, one has to work on the unbiased risk estimator $\widehat{R}_t$ and requires optimizing the *empirical* dynamic regret $\sum_{t=1}^{T}\widehat{R}_t(\mathbf{w}_t) - \sum_{t=1}^{T}\widehat{R}_t(\mathbf{w}_t^*)$. Importantly, $\mathbf{w}_t^* \in \arg\min_{\mathbf{w}\in\mathcal{W}} R_t(\mathbf{w})$ but $\mathbf{w}_t^* \notin \arg\min_{\mathbf{w}\in\mathcal{W}}\widehat{R}_t(\mathbf{w})$ in general. Although the empirical dynamic regret can be trivially bounded by $\sum_{t=1}^{T}\widehat{R}_t(\mathbf{w}_t) - \sum_{t=1}^{T}\min_{\mathbf{w}\in\mathcal{W}}\widehat{R}_t(\mathbf{w})$, the bound will then be loose and related to temporal variation of risk estimators [17, 27], making it hard to establish relationship to the natural non-stationarity measure of OLaS: $V_T$.[4] On the other hand, there exist studies benchmarking dynamic regret with other choices of comparators [20, 18], but the bounds scale with the consecutive variation

---

[4]Besbes et al. [17] also considered a more general setting with noisy function value feedback. In such cases, the comparator is not the exact minimizer of the online function at each round, and their algorithm will require a *periodical restart* to deal with the non-stationarity. By contrast, ours does not require the restart in the algorithm.

| **Algorithm 1** ATLAS: base-algorithm | **Algorithm 2** ATLAS: meta-algorithm |
|---|---|
| **Input:** step size $\eta_i \in \mathcal{H}$ | **Input:** step size pool $\mathcal{H}$; learning rate $\varepsilon$ |
| 1: let $\mathbf{w}_{1,i}$ be any point in $\mathcal{W}$ | 1: initialization: $\forall i \in [N], p_{1,i} = 1/N$ |
| 2: **for** $t = 2$ **to** $T$ **do** | 2: **for** $t = 2$ **to** $T$ **do** |
| 3:    construct the risk estimator $\widehat{R}_{t-1}$ as (3) | 3:    receive $\{\mathbf{w}_{t,i}\}_{i=1}^N$ from base-learners |
| 4:    update the model of base-learner by | 4:    update weight $\mathbf{p}_t \in \Delta_N$ according to |
|      $\mathbf{w}_{t,i} = \Pi_{\mathcal{W}}[\mathbf{w}_{t-1,i} - \eta_i \nabla \widehat{R}_{t-1}(\mathbf{w}_{t-1,i})]$ |      $p_{t,i} \propto \exp(-\varepsilon \sum_{s=1}^{t-1} \widehat{R}_s(\mathbf{w}_{s,i})), i \in [N]$ |
| 5:    send $\mathbf{w}_{t,i}$ to the meta-algorithm | 5:    predict final output $\mathbf{w}_t = \sum_{i=1}^N p_{t,i} \mathbf{w}_{t,i}$ |
| 6: **end for** | 6: **end for** |

of comparators $\sum_{t=2}^T \|\mathbf{w}_t^* - \mathbf{w}_{t-1}^*\|_2$, whose relation to $V_T$ is also unclear. To address the difficulty, drawing inspiration from [28], we decompose the expected dynamic regret into two parts: (i) the analysis of the first part relies on an in-depth analysis of the empirical dynamic regret of UOGD to track a specific sequence of piecewise-stationary comparators to avoid undesired variation of comparators; and (ii) the analysis of the second part is directly conducted on the *original* risk functions to attain a non-stationarity measure only related to the underlying label distribution shift.

From the upper bound in Eq. (5), we can observe that a proper step size tuning is crucial. Specifically, it can be verified that when the environment is near-stationary (more precisely, $V_T \le \Theta(T^{-\frac{1}{2}})$), simply choosing $\eta = \Theta(1/\sqrt{T})$ ensures an $\mathcal{O}(\sqrt{T})$ dynamic regret, which is known to be minimax optimal even for the weaker measure of static regret [29]. Thus, in the following, we focus on the non-degenerated OLaS situation where $V_T \ge \Theta(T^{-\frac{1}{2}})$, and then the dynamic regret upper bound can be further simplified as $\mathcal{O}(\eta T + 1/\eta + \sqrt{(V_T T)/\eta})$. As a result, UOGD can attain an $\mathcal{O}(V_T^{\frac{1}{3}} T^{\frac{2}{3}})$ dynamic regret by setting the step size optimally as $\eta = \Theta(T^{-\frac{1}{3}} V_T^{\frac{1}{3}})$. According to the lower bound results of Besbes et al. [17], we know that UOGD with an optimal step size tuning ensures a minimax optimal dynamic regret guarantee, see more discussions on the minimax optimality in Appendix E.3.

However, the optimal step size $\eta^* = \Theta(T^{-\frac{1}{3}} V_T^{\frac{1}{3}})$ crucially depends on the label distribution drift $V_T = \sum_{t=2}^T \|\boldsymbol{\mu}_{y_t} - \boldsymbol{\mu}_{y_{t-1}}\|_1$, which measures the non-stationarity and is unfortunately *unknown* to the learner. It is worth emphasizing that the problem cannot be addressed by standard adaptive step size tuning mechanisms in online learning literature, such as the doubling trick [30] or self-confident tuning [31], in that the variation quantity $V_T$ cannot be empirically evaluated as it is defined over the underlying label distribution $\mathcal{D}_t(y)$ inaccessible to the learner. Even diving into the analysis of Theorem 1, the adaptive tuning requires the knowledge of *original* risk functions $\{R_t\}_{t=1}^T$, which are also unavailable. Intuitively, the hardness of such optimal step size tuning essentially comes from the uncertainty of the non-stationary label shifts.

To overcome the difficulty, inspired by recent advances in non-stationary online learning [18, 19], we propose an *online ensemble* algorithm for the OLaS problem called ATLAS (Adapting To LAbel Shift). Specifically, to cope with the uncertainty in the optimal step size tuning, we first design a pool of candidate step sizes denoted by $\mathcal{H} = \{\eta_1, \ldots, \eta_N\}$ such that $\eta_*$ can be well approximated by at least one of those candidates, which will be configured later and here $N$ is the number of candidate step sizes. Then, ATLAS deploys a two-layer structure by maintaining a group of base-learners, each associated with a candidate step size from the pool $\mathcal{H}$ and then employs a meta-algorithm to track the best base-learner. The main procedures are presented in Algorithm 1 (base-algorithm) and Algorithm 2 (meta-algorithm), and we describe the details below.

**Base-algorithm.** We parallelly run a group of instances of UOGD, each one is associated with a candidate step size in the step size pool $\mathcal{H}$. Formally, the $i$-th base-learner yields a sequence of base models $\{\mathbf{w}_{t,i}\}_{t=1}^T$, which is updated by $\mathbf{w}_{t,i} = \Pi_{\mathcal{W}}[\mathbf{w}_{t-1,i} - \eta_i \nabla \widehat{R}_{t-1}(\mathbf{w}_{t-1,i})]$ with $\eta_i \in \mathcal{H}$.

**Meta-algorithm.** The meta-algorithm aims to combine all the base-learners' decisions such that the final output is competitive with the decisions returned by the (unknown) base-learner associated with the best step size. To achieve this, we employ a weighted combination mechanism with the final model as $\mathbf{w}_t = \sum_{i=1}^N p_{t,i} \mathbf{w}_{t,i}$, where $\mathbf{p}_t \in \Delta_N$ is the weight vector with $p_{t,i}$ denoting the weight of the $i$-th base-learner. We use the classic Hedge algorithm [32] to update the weight vector, namely, $p_{t,i} \propto \exp\left(-\varepsilon \sum_{s=1}^{t-1} \widehat{R}_s(\mathbf{w}_{s,i})\right)$, where $\varepsilon > 0$ is the learning rate of the meta-algorithm

that can be simply set as $\Theta(\sqrt{(\ln N)/T})$ without dependence on label shift quantity. Intuitively, the meta-algorithm puts larger weights on base-learners with a smaller cumulative estimated risk so that the overall models $\{\mathbf{w}_t\}_{t=1}^T$ can be competitive to the base-learner with the best performance.

ATLAS enjoys the following dynamic regret guarantee with the proof in Appendix E.2.

**Theorem 2.** *Set the step size pool as* $\mathcal{H} = \{\eta_i = \frac{\Gamma\sigma}{2G\sqrt{KT}} \cdot 2^{i-1} \mid i \in [N]\}$, *where* $N = 1 + \lceil \frac{1}{2} \log_2(1 + 2T) \rceil$ *is the number of base-learners.* ATLAS *ensures that* $\mathbb{E}[\mathbf{Reg}_T^{\mathbf{d}}] \leq \mathcal{O}(\max\{V_T^{\frac{1}{3}} T^{\frac{2}{3}}, \sqrt{T}\})$, *or simplified as* $\mathcal{O}(V_T^{\frac{1}{3}} T^{\frac{2}{3}})$ *for non-degenerated cases of* $V_T \geq \Theta(T^{-\frac{1}{2}})$.

Theorem 2 shows that ATLAS enjoys the same dynamic regret as UOGD with the (unknown) optimal step size, but unknown non-stationarity $V_T$ is no more required in advance. Algorithmically, our online method maintains $N = \Theta(\log T)$ base-learners to achieve an optimal dynamic regret, which would be computationally acceptable given a logarithmic dependence on $T$.

### 3.3 More Adaptive Algorithm by Exploiting Label Shift Structures

ATLAS is equipped with a minimax optimal dynamic regret, which indicates that it can safeguard the optimal theoretical property in the worst case. Worst-case optimality serves as the "stress-testing" for the robustness to non-stationarity environments [33]. At the same time, more adaptive results beyond the worst-case analysis are also urgently desired, as improved performance is naturally expected in many easier situations when the shift admits specific patterns such as periodicity or gradual evolution.

To this end, we propose an improved algorithm called ATLAS-ADA with provably more adaptive guarantees. The key idea is to exploit the label shift patterns and reuse historical information to help the current online update [34]. We build on the framework of *optimistic online learning* [35, 36] by introducing a *hint* function $H_t : \mathcal{W} \mapsto \mathbb{R}$ to encode shift patterns from historical data, which serves as an estimation of the expected risk $R_t(\mathbf{w}_t)$. Below, we start with a given hint function and describe the usage, and finally elaborate on how to design $H_t(\cdot)$ guided by the theory.

Similar to ATLAS, the improved ATLAS-ADA also deploys a two-layer meta-base structure. The key difference lies in the usage of the hint function at both the base-level and meta-level.

**Base-algorithm.** Besides the gradient descent step as did in ATLAS, another update step related to the hint function $H_t(\cdot)$ is performed. Concretely, the $i$-th base-learner updates the parameters by

$$\widehat{\mathbf{w}}_{t,i} = \Pi_{\mathcal{W}}[\widehat{\mathbf{w}}_{t-1,i} - \eta_i \nabla \widehat{R}_{t-1}(\mathbf{w}_{t-1,i})], \quad \mathbf{w}_{t,i} = \arg\min_{\mathbf{w} \in \mathcal{W}} \eta_i H_t(\mathbf{w}) + {}^{1}/{}_2 \cdot \|\mathbf{w} - \widehat{\mathbf{w}}_{t,i}\|_2^2, \quad (6)$$

where $\widehat{\mathbf{w}}_{t,i}$ is an intermediate output and $\mathbf{w}_{t,i}$ is the final returned model. When $H_t(\mathbf{w}) = 0$ (i.e., without a hint function), the above two-step update simply degenerates to the same UOGD update in the base-learner of ATLAS by noting that now $\mathbf{w}_{t,i} = \widehat{\mathbf{w}}_{t,i}$. In the general $H_t(\cdot)$ case, the second step in (6) is crucial and can be regarded as another descent towards the direction specified by the hint function. As a result, this will reduce the regret whenever the hint function is set appropriately to approximate well the next-round risk function, which will be clear in the regret bound presented later.

**Meta-algorithm.** The meta-algorithm is used to track the best base-learner, and the hint function $H_t(\cdot)$ is also necessary to be considered in the update to achieve the adaptivity. To this end, we inject the hint function as the loss evaluation of the meta-algorithm, and then the weight is updated by $p_{t,i} \propto \exp\left(-\varepsilon\left(\sum_{s=1}^{t-1} \widehat{R}_s(\mathbf{w}_{s,i}) + H_t(\mathbf{w}_{t,i})\right)\right)$ for all $i \in [N]$, where $\varepsilon$ is the learning rate that can be set properly without dependence on $V_T$. The key distinction to the meta-algorithm of ATLAS is the additional loss $H_t(\mathbf{w}_{t,i})$ evaluated over the current local models $\{\mathbf{w}_{t,i}\}_{i=1}^N$ by the hint function.

The main procedures are presented in Algorithm 4 (base-algorithm) and Algorithm 5 (meta-algorithm). We have the following dynamic regret bound for ATLAS-ADA (proof in Appendix F.2).

**Theorem 3.** *Suppose the hint function* $H_t : \mathcal{W} \mapsto \mathbb{R}$ *is convex, satisfies* $\max_{\mathbf{w} \in \mathcal{W}} \|\nabla H_t(\mathbf{w})\|_2 \leq \max_{\mathbf{w} \in \mathcal{W}} \|\nabla \widehat{R}_t(\mathbf{w})\|_2$, *and is independent of current data* $S_t$. *Set the step size pool as* $\mathcal{H} = \{\eta_i = \frac{\Gamma\sigma}{\sqrt{\sigma^2 + 4G^2 KT}} \cdot 2^{i-1} \mid i \in [N]\}$ *with* $N = 2 + \lceil \frac{1}{2} \log_2(3T(1 + 4G^2 KT/\sigma)) \rceil$. ATLAS-ADA *ensures* $\mathbb{E}[\mathbf{Reg}_T^{\mathbf{d}}] \leq \mathcal{O}(V_T^{1/3} G_T^{1/3} T^{1/3})$, *where* $G_T = \sum_{t=1}^T \mathbb{E}[\sup_{\mathbf{w} \in \mathcal{W}} \|\nabla \widehat{R}_t(\mathbf{w}) - \nabla H_t(\mathbf{w})\|_2^2]$ *measures the reusability of historical information, depending on label shift patterns and hint function designs.*

In the worst case, $G_T$ is at most $\mathcal{O}(T)$ given a bounded gradient norm $\|\nabla H_t(\mathbf{w})\|_2$, and thus the bound presented in Theorem 3 safeguards the same $\mathcal{O}(V_T^{1/3} T^{2/3})$ bound as ATLAS. More

importantly, when the hint function $H_t(\cdot)$ encodes beneficial information and is close to the risk function, the obtained bound can be substantially better than the minimax rate.

**Remark 4** (Implicit update). Problem-dependent dynamic regret was first presented in [19] for convex and smooth functions. However, their result critically relied on the *smoothness* condition, which is not satisfied in our OLaS case. Our key technical innovation is the *implicit update* in the second step of (6). The previous method required the gradient-descent type update $\mathbf{w}_{t,i} = \Pi_{\mathbf{w} \in \mathcal{W}}[\widehat{\mathbf{w}}_{t,i} - \eta_i \nabla H_t(\mathbf{w}_{t-1,i})]$, which can be deemed as an approximated optimization over the linearized loss $\langle \nabla H_t(\mathbf{w}_{t-1,i}), \mathbf{w} \rangle$. By contrast, we directly updates over the *original function* $H_t(\mathbf{w})$, hence called the "implicit" update [37, 38]. Albeit with slightly larger computational complexity (which will not be a barrier given a proper design of hint functions), our method enjoys the same dynamic regret *without* smoothness, which could be of independent interest for general OCO purposes.

**Design of Hint Functions.** The hint functions should minimize the reusability measure $G_T$ to sharpen the dynamic regret as suggested by Theorem 3. Recall that $\widehat{R}_t(\mathbf{w}) = \sum_{k=1}^{K} [\widehat{\boldsymbol{\mu}}_{y_t}]_k \cdot \widehat{R}_0^k(\mathbf{w})$. Thus, a natural construction is $H_t(\mathbf{w}) = \sum_{k=1}^{K} [\boldsymbol{h}_{y_t}]_k \cdot \widehat{R}_0^k(\mathbf{w})$ parametrized by *hint priors* $\boldsymbol{h}_{y_t} \in \mathbb{R}^K$, which is used to estimate the class prior based on the past observed data $\{S_\tau\}_{\tau=1}^{t-1}$. Then, $G_T$ satisfies the *bias-variance decomposition* (with bias term $\mathbb{E}[\|\boldsymbol{h}_{y_t} - \boldsymbol{\mu}_{y_t}\|_2^2]$ and variance term $\mathbb{E}[\|\boldsymbol{\mu}_{y_t} - \widehat{\boldsymbol{\mu}}_{y_t}\|_2^2]$):

$$G_T \leq KG^2 \sum_{t=1}^{T} \mathbb{E}\left[\|\boldsymbol{h}_{y_t} - \widehat{\boldsymbol{\mu}}_{y_t}\|_2^2\right] \leq 2KG^2 \sum_{t=1}^{T} \left(\mathbb{E}\left[\|\boldsymbol{h}_{y_t} - \boldsymbol{\mu}_{y_t}\|_2^2\right] + \mathbb{E}\left[\|\boldsymbol{\mu}_{y_t} - \widehat{\boldsymbol{\mu}}_{y_t}\|_2^2\right]\right).$$

Setting $\boldsymbol{h}_{y_t} = \boldsymbol{\mu}_{y_t}$ will make the upper bound tightest possible, though the underlying class prior $\boldsymbol{\mu}_{y_t}$ is not accessible in practice. So the design of the hint function is actually a task of approximating it with different parts of previous data guided by prior patterns. In the experiments, we design four hint functions by encoding different knowledge, including Forward Hint (*Fwd*), Window Hint (*Win*), Periodic Hint (*Peri*), and Online KMeans Hint (*OKM*). More details are in Appendix F.1.

## 4 Experiments

In this section, we conduct extensive experiments to validate the effectiveness of the proposed methods (ATLAS and ATLAS-ADA) and justify the theoretical findings. We begin this section with a brief introduction to the experimental setups (more details are deferred to Appendix A) and then present empirical results on the synthetic and real-world data, respectively.

**Experiments Setup.** We compare seven algorithms in various experimental configurations. The contenders include a baseline that predicts with the initial model directly (*FIX*), three OLaS algorithms proposed by the previous work [16] (*ROGD*, *FTH*, and *FTFWH*), and our proposals (*UOGD*, ATLAS, and ATLAS-ADA) with the logistic regression model. Besides, we simulate four types of label shift on synthetic and benchmark data to capture different distribution change patterns. Two of them, including Sine Shift (〰〰) and Square Shift (⊓⊔⊓⊔) change in a periodic pattern. The other two have no periodic structure but are introduced to capture different shift intensities. The underlying distribution changes slowly in the Linear Shift (⌣) while changes fast in the Bernoulli Shift (⊓⊔⊓). We repeat all experiments for five times and evaluate the contenders by the average error $\frac{1}{T} \sum_{t=1}^{T} \frac{1}{|S_t|} \sum_{n=1}^{|S_t|} \mathbb{1}[f(\mathbf{w}, \mathbf{x}_n) \neq y_n]$ over $T = 10,000$ rounds.

### 4.1 Illustrations on Synthetic Data

This subsection first compares all contenders on the synthetic data. Then, we further illustrate the effectiveness of our proposal by a closer look at the two key components, including a meta-base structure for step size search and a hint function for historical information reuse.

**Overall Performance Comparison.** Table 1 compares ATLAS with other methods when $N_t = 100$ samples are received at every iteration. Basically, *FIX* is inferior to the online methods, which shows the necessity of designing online algorithms for the OLaS problem. Besides, *UOGD* outperforms *ROGD*, the OGD algorithm running with the risk estimator proposed by [16]. The comparison demonstrates the empirical superiority of our estimator besides its benign theoretical properties. Moreover, the ATLAS surpasses almost all other methods in the four shift patterns. In particular, it achieves a significant advantage over *UOGD* (with step size $\eta = \Theta(T^{-1/2})$) when the environments change relatively fast (Squ, Sin, and Ber). The results justify our theoretical finding that the small

**Table 1:** Average error (%) for different algorithms under various simulated shifts for the synthetic data. The best algorithms are emphasized in bold (paired $t$-test at a 5% confidence level).

|       | Lin       | Squ       | Sin       | Ber       |
|-------|-----------|-----------|-----------|-----------|
| FIX   | 7.87±0.03 | 7.87±0.02 | 7.34±0.03 | 7.79±0.02 |
| FTH   | **4.70±0.02** | 6.50±0.01 | 6.36±0.03 | 6.60±0.01 |
| FTFWH | 5.27±0.02 | 6.52±0.01 | 6.36±0.02 | 6.60±0.01 |
| ROGD  | 6.08±0.01 | 7.11±0.01 | 6.87±0.02 | 6.40±0.01 |
| UOGD  | 5.35±0.02 | 6.17±0.01 | 6.37±0.01 | 5.46±0.05 |
| ATLAS | 5.44±0.02 | **4.27±0.02** | 5.75±0.01 | **4.04±0.07** |

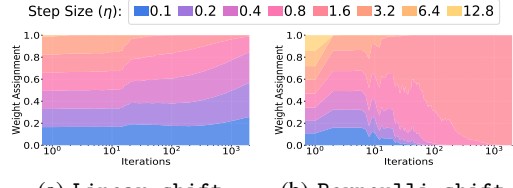

Step Size ($\eta$): 0.1  0.2  0.4  0.8  1.6  3.2  6.4  12.8

(a) Linear shift  (b) Bernoulli shift

**Figure 1:** Weight assigned of the ATLAS algorithm for each step size along the learning process. Different colors are used to indicate different step sizes.

**Table 2:** Average error (%) for ATLAS-ADA with four hint functions under different sample sizes. The best one is emphasized in bold. Besides, ● indicates a better result than ATLAS without hint (*None*).

| Shift Type | Sample Size: 1 | | | | | Sample Size: 10 | | | | | Sample Size: 100 | | | | |
|------------|------|------|------|------|------|------|------|------|------|------|------|------|------|------|------|
|            | *None* | *Win* | *Peri* | *Fwd* | *OKM* | *None* | *Win* | *Peri* | *Fwd* | *OKM* | *None* | *Win* | *Peri* | *Fwd* | *OKM* |
| Lin | 6.28 | ●5.89 | ●5.99 | ●6.01 | ●**5.35** | 5.61 | ●5.47 | ●5.43 | ●5.53 | ●5.42 | 5.44 | 5.44 | ●**5.38** | ●5.40 | 5.45 |
|     | ±0.21 | ±0.26 | ±0.29 | ±0.31 | ±0.31 | ±0.04 | ±0.04 | ±0.03 | ±0.05 | ±0.05 | ±0.02 | ±0.03 | ±0.02 | ±0.02 | ±0.03 |
| Squ | 6.03 | ●5.83 | ●5.27 | 5.88 | ●**5.07** | 4.59 | 4.69 | ●3.85 | ●**3.72** | ●3.91 | 4.27 | 4.68 | ●3.39 | ●**3.33** | ●3.46 |
|     | ±0.23 | ±0.24 | ±0.20 | ±0.23 | ±0.35 | ±0.02 | ±0.02 | ±0.04 | ±0.02 | ±0.03 | ±0.02 | ±0.02 | ±0.03 | ±0.03 | ±0.04 |
| Sin | 6.90 | ●6.58 | ●6.59 | ●6.43 | ●**5.25** | 6.12 | ●5.99 | ●5.83 | ●**5.78** | ●5.86 | 5.75 | 5.78 | ●5.53 | ●**5.48** | ●5.58 |
|     | ±0.22 | ±0.22 | ±0.25 | ±0.26 | ±0.22 | ±0.07 | ±0.06 | ±0.05 | ±0.05 | ±0.04 | ±0.01 | ±0.01 | ±0.00 | ±0.01 | ±0.00 |
| Ber | 5.55 | ●5.42 | ●5.43 | 5.63 | ●**4.69** | 4.39 | 4.45 | 4.43 | ●**3.66** | ●3.73 | 4.04 | 4.29 | 4.26 | ●**3.19** | ●3.45 |
|     | ±0.09 | ±0.11 | ±0.09 | ±0.16 | ±0.17 | ±0.10 | ±0.08 | ±0.10 | ±0.10 | ±0.06 | ±0.07 | ±0.06 | ±0.06 | ±0.07 | ±0.11 |

step size $\Theta(T^{-1/2})$ suggested by the static regret analysis is unsuitable for the dynamic environments. Our method can better adapt to the changing environments by enjoying the dynamic regret guarantees.

**Effectiveness of Meta-Base Structure.** One key component of our method is the meta-base structure to address the non-stationarity. To better illustrate its effectiveness, we visualize the weights $p_{t,i}$ assigned for each base-learner of ATLAS. As shown in Figure 5, the meta-algorithm can quickly assign larger weights to appropriate base-learners along the learning process. Specifically, Figure 1(a) illustrates the case of slowly changing environments, where more weights are assigned to the base-learners with small step sizes. In the fast-changing case, see Figure 1(b), larger step sizes are preferred. The results show that our algorithm can adaptively track the suitable step sizes according to the shift intensity of environments. Additional results for other shifts can be found in Figure 5.

**Effectiveness of Using Hint Functions.** Table 2 reports the performance of ATLAS-ADA with four different hint functions under sample sizes are $N_t = 1, 10$, and 100. All hint functions improve over vanilla ATLAS in most cases. When $N_t$ is reasonably large, the Fwd hint, performing transductive learning with current unlabeled data, achieves the best performance. While, the OKM hint, which learns previous patterns by online k-means, is the best choice for a small sample size case ($N_t = 1$).

To illustrate how the hint function works, we further vary the buffer size of the Periodic Hint (*Peri*) on the Squ environment ( ), which shifts in a periodic length $L = 40$. As shown in Figure 2, when the buffer size matches the multiples of length $L$, *Peri* can significantly improve the vanilla ATLAS. Besides the improvement, our method is shown to achieve comparable performance with ATLAS even if the buffer size is misspecified. The result validates our theoretical guarantee of the safety of using hint functions (the readers can refer to the paragraph above Remark 4).

### 4.2 Comparisons on Real-world Data

We conduct experiments on real-world data, including six benchmark datasets (ArXiv, EuroSAT, MNIST, Fashion, CIFAR10, and CINIC10) and the SHL dataset [39] for the real-life locomotion detection task. Table 3 reports the averaged error of different algorithms, which shows a similar tendency as the results in the synthetic experiments. When the distribution changes rapidly (Ber), ATLAS and ATLAS-ADA outperform other contenders. Similar results are also observed in the Sine and Square shift, see Appendix A.1 for details. Even in a relatively stationary environment (Lin), our algorithms are comparable with the best algorithm (UOGD), which is specifically designed for stationary cases. The above results validate the adaptivity of the proposed algorithms.

Further, we highlight the results on the locomotion detection task. The task aims to distinguish six types of locomotion with sensor data from mobile phones. Figure 3(a) reports the averaged error of all contenders, which shows the superiority of our proposals ATLAS and ATLAS-ADA (with the *OKM* hint) over the entire time horizons. In addition, to further validate the adaptivity of our algorithm

**Table 3:** Average error (%) of different algorithms on various real-world datasets (`Lin` and `Ber`). We report the mean and standard deviation over five runs. The best algorithms are emphasized in bold. "●" indicates the algorithms that are significantly inferior to ATLAS-ADA by the paired $t$-test at a 5% significance level. Here AT-ADA represents ATLAS-ADA (with *OKM*). The online sample size is set as $N_t = 10$.

| | Lin | | | | | | | Ber | | | | | | |
|---|---|---|---|---|---|---|---|---|---|---|---|---|---|---|
| | FIX | FTH | FTFWH | ROGD | UOGD | ATLAS | AT-ADA | FIX | FTH | FTFWH | ROGD | UOGD | ATLAS | AT-ADA |
| **ArXiv** | ●30.28 ±0.07 | ●28.18 ±0.28 | ●25.74 ±0.21 | ●23.09 ±0.20 | **21.04** ±0.11 | ●22.10 ±0.09 | 21.28 ±0.09 | ●30.63 ±0.20 | ●27.69 ±0.13 | ●28.50 ±0.19 | ●24.82 ±0.11 | ●21.53 ±0.68 | ●21.11 ±0.70 | **20.58** ±0.69 |
| **EuroSAT** | ●14.06 ±0.09 | ●11.16 ±0.11 | ●9.78 ±0.12 | ●12.56 ±3.16 | **7.04** ±0.11 | ●7.19 ±0.10 | 7.13 ±0.11 | ●14.12 ±0.13 | ●10.48 ±0.09 | ●10.50 ±0.08 | ●9.06 ±0.05 | ●7.28 ±0.04 | ●6.99 ±0.03 | **6.91** ±0.05 |
| **MNIST** | ●1.79 ±0.02 | ●1.38 ±0.03 | ●1.20 ±0.02 | ●1.25 ±0.02 | **1.06** ±0.02 | **1.06** ±0.02 | 1.06 ±0.02 | ●1.81 ±0.05 | ●1.29 ±0.03 | ●1.34 ±0.03 | ●1.33 ±0.03 | ●1.12 ±0.02 | **1.03** ±0.02 | 1.03 ±0.02 |
| **Fashion** | ●11.86 ±0.04 | ●8.47 ±0.07 | **7.84** ±0.06 | 8.18 ±0.07 | 7.95 ±0.08 | ●8.36 ±0.07 | 8.04 ±0.08 | ●11.85 ±0.09 | ●8.48 ±0.11 | ●8.69 ±0.10 | ●8.72 ±0.08 | ●8.23 ±0.12 | ●7.91 ±0.12 | **7.69** ±0.12 |
| **CIFAR10** | ●20.77 ±0.12 | ●17.36 ±0.14 | 15.77 ±0.12 | ●18.45 ±0.47 | **15.54** ±0.15 | ●15.77 ±0.11 | 15.62 ±0.14 | ●20.82 ±0.12 | ●17.06 ±0.14 | ●16.96 ±0.15 | ●17.66 ±0.13 | ●15.93 ±0.29 | ●14.98 ±0.30 | **14.80** ±0.29 |
| **CINIC10** | ●33.98 ±0.22 | ●28.85 ±0.10 | ●26.87 ±0.13 | ●32.54 ±2.59 | **26.21** ±0.15 | ●26.66 ±0.19 | 26.38 ±0.16 | ●34.11 ±0.35 | ●28.48 ±0.17 | ●28.44 ±0.19 | ●28.90 ±0.19 | ●26.63 ±0.55 | ●25.85 ±0.58 | **25.63** ±0.60 |

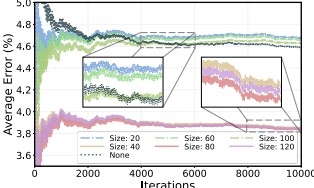

**Figure 2:** Average error of *Peri* hint with different buffer sizes. ATLAS-ADA enjoys a significant improvement with proper buffer sizes while still safeguarding similar performance as ATLAS even the size is misspecified.

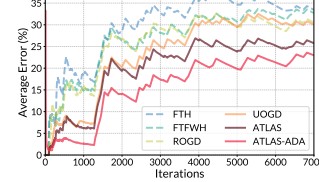

(a) overall performance

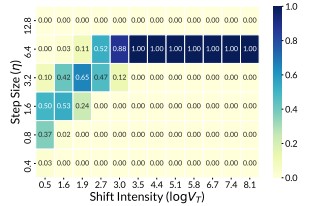

(b) weight heatmap

**Figure 3:** (a) Overall performance comparison on the locomotion detection task. (b) The heatmap of the final round weights for each step size. A larger $V_T$ implies a larger intensity of distribution shift. The darker color in the heatmap indicates the larger weight.

to the underlying environment regardless of the fast or slow changes, we simulate various shift intensities by sampling the original data with different frequencies. Figure 3(b) shows the weight assignment of ATLAS-ADA for each step size in the final round. Our method automatically selects a larger step size for larger $V_T$ while tracking a small step size in a relatively static scenario.

## 5 Conclusion

This paper proposed algorithms for online label shift with provable guarantees. We constructed an unbiased risk estimator without using any supervision at test time. Then, we designed novel online ensemble algorithms that automatically adapt to the non-stationary online label shift and enjoy problem-dependent dynamic regret. Our proposed ATLAS algorithm employed a meta-base structure to handle the non-stationarity and obtained an $\mathcal{O}(V_T^{1/3}T^{2/3})$ guarantee, and ATLAS-ADA further introduced hint functions to exploit the shift structure and obtained an improved $\mathcal{O}(V_T^{1/3}G_T^{1/3}T^{1/3})$ guarantee. Extensive experiments validated the effectiveness of the proposed algorithms.

Our study serves as a preliminary attempt to bridge the distribution change problem and online learning techniques by focusing on the label shift case. Considering a more general distribution change setting is an important future direction. Besides, our current algorithm is designed for the most challenging unlabeled scenario, and it is interesting to consider relaxed real-world demand where a few labels could be available in the learning process. Moreover, our obtained regret guarantees hold in expectation, and we will take high-probability bounds as the future work.

## Acknowledgments

Yong Bai, Peng Zhao, and Zhi-Hua Zhou were supported by NSFC (61921006, 62206125), JiangsuSF (BK20220776), and Collaborative Innovation Center of Novel Software Technology and Industrialization. Yu-Jie Zhang was supported by Todai Fellowship and the Institute for AI and Beyond, UTokyo. Masashi Sugiyama was supported by JST AIP Acceleration Research Grant Number JPMJCR20U3 and the Institute for AI and Beyond, UTokyo. Peng Zhao thanks Chen-Yu Wei for discussions on the conversation between function variation and gradient variation. The authors also thank Ruihan Wu, Yu-Yang Qian, and Dheeraj Baby for their helpful discussions.

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
