## A    Omitted Details for Experiments

In this section, we mainly supplement the omitted details in Section 4. We first present the omitted numerical results on benchmark datasets in Appendix A.1. Then we list the experiment setups in Appendix A.2, followed by additional experimental results in Appendix A.3.

### A.1    More Numerical Results for Section 4.2

Table 4 presents the numerical results omitted in Section 4.2, which reports the average error of different algorithms on various real-world datasets. The results show that the ATLAS and ATLAS-ADA methods outperform other contenders on all tasks. The advantage is particularly significant when the underlying class prior changes fast. The empirical results validate that our method can effectively adapt to the non-stationary label shift.

### A.2    Experiment Setup

This subsection describes details of experimental setups, including contenders, simulated shifts, and datasets. All our experiments are run on a machine with 2 CPUs (24 cores for each).

**Table 4:** Average error (%) of different algorithms on various real-world datasets (Sin and Squ). We report the mean and standard deviation over five runs. The best algorithms are emphasized in bold. "•" indicates the algorithms that are significantly inferior to ATLAS-ADA by the paired $t$-test at a 5% significance level. Here AT-ADA represents ATLAS-ADA (with *OKM*). The online sample size is set as $N_t = 10$.

| | Sin | | | | | | | Squ | | | | | | |
|---|---|---|---|---|---|---|---|---|---|---|---|---|---|---|
| | **FIX** | **FTH** | **FTFWH** | **ROGD** | **UOGD** | **ATLAS** | **AT-ADA** | **FIX** | **FTH** | **FTFWH** | **ROGD** | **UOGD** | **ATLAS** | **AT-ADA** |
| **ArXiv** | •31.58 ±0.10 | •30.63 ±0.24 | •31.90 ±0.22 | •28.35 ±0.32 | •25.64 ±0.18 | •26.03 ±0.16 | **25.08** ±0.12 | •30.35 ±0.06 | •26.72 ±0.39 | •28.05 ±0.20 | •24.44 ±0.17 | •21.96 ±0.07 | •21.36 ±0.06 | **20.80** ±0.06 |
| **EuroSAT** | •13.62 ±0.13 | •10.90 ±0.03 | •10.96 ±0.02 | •9.68 ±0.08 | •8.03 ±0.06 | •8.03 ±0.06 | **7.97** ±0.08 | •14.15 ±0.11 | •10.22 ±0.08 | •10.26 ±0.06 | •8.91 ±0.05 | •7.30 ±0.07 | •6.97 ±0.08 | **6.81** ±0.06 |
| **MNIST** | •1.81 ±0.02 | •1.46 ±0.03 | •1.47 ±0.03 | •1.46 ±0.03 | •1.30 ±0.04 | 1.28 ±0.03 | **1.27** ±0.03 | •1.79 ±0.04 | •1.26 ±0.03 | •1.28 ±0.04 | •1.32 ±0.04 | •1.13 ±0.03 | •1.04 ±0.02 | **1.01** ±0.04 |
| **Fashion** | •11.77 ±0.11 | 9.37 ±0.15 | 9.39 ±0.14 | •9.75 ±0.12 | 9.36 ±0.07 | •9.44 ±0.04 | **9.32** ±0.04 | •11.92 ±0.09 | •8.24 ±0.09 | •8.35 ±0.07 | •8.63 ±0.07 | •8.42 ±0.04 | •8.05 ±0.07 | **7.73** ±0.05 |
| **CIFAR10** | •21.40 ±0.09 | •18.57 ±0.07 | •18.62 ±0.08 | •19.16 ±0.12 | •18.17 ±0.07 | •18.01 ±0.07 | **17.89** ±0.05 | •20.77 ±0.08 | •16.67 ±0.12 | •16.72 ±0.12 | •17.40 ±0.12 | •16.29 ±0.11 | •15.18 ±0.07 | **14.84** ±0.05 |
| **CINIC10** | •35.29 ±0.19 | •31.17 ±0.12 | •31.20 ±0.12 | •31.46 ±0.14 | •30.22 ±0.10 | •30.15 ±0.11 | **30.06** ±0.15 | •33.99 ±0.16 | •27.99 ±0.09 | •28.08 ±0.08 | •28.58 ±0.09 | •27.00 ±0.14 | •25.94 ±0.13 | **25.56** ±0.12 |

**Contenders.**  In the experiments, we mainly compare seven online label shift algorithms, including:

- *FIX* predicts with the fixed initialized classifier without any online updates.
- *ROGD* is a variant of OGD algorithm proposed by Wu et al. [16]. The algorithm constructs the risk estimator with the $0/1$ loss and uses a re-weighting classifier for the model update. (Detailed description and the comparison with our risk estimator can be found in Appendix C.3).
- *FTH* is short for Following The History algorithm proposed by Wu et al. [16]. The method takes the class prior for every iteration $t$ as the average of all previously estimated priors.
- *FTFWH* is a variant of FTH proposed as a compared method in experiments of Wu et al. [16]. The method takes an average across previously estimated priors within a sliding window. In all experiments, the length of the sliding window is set as 100.
- *UOGD*/ATLAS/ATLAS-ADA: The OLaS algorithms proposed in Section 3.

Following our theoretical results, the step size of *ROGD* and *UOGD* is set to be $\Gamma/(G\sqrt{T})$, where $G$ can be estimated during training the initial model and $\Gamma$ is given by the decision set domain. In the experiments, we choose the decision domain as a ball with a fixed diameter for each dataset. For a fair comparison, all contenders use the same decision domain $\mathcal{W}$ with the diameter $\Gamma$ set according to the parameter norm of the initial offline model $f_0$. The settings of step size pool of ATLAS and ATLAS-ADA are guided by Theorem 2 and Theorem 3, respectively. We employ a multinomial logistic regression classifier for *UOGD*, ATLAS, and ATLAS-ADA.

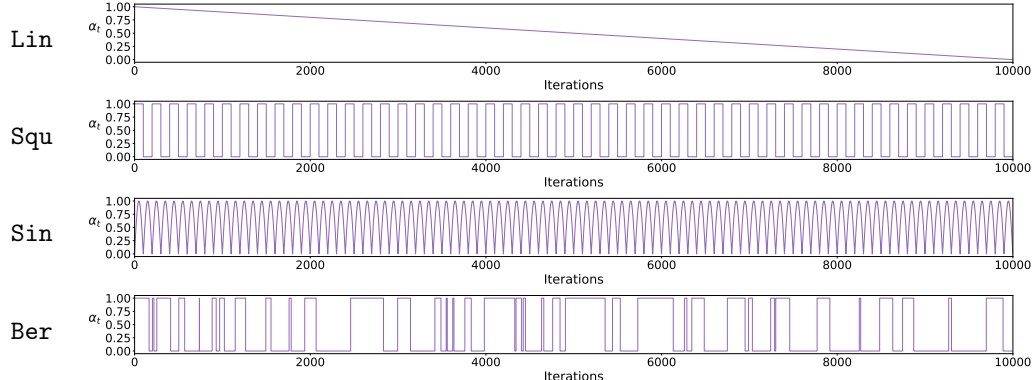

**Figure 4:** From top to bottom: `Linear Shift`, `Square Shift`, `Sine Shift`, `Bernoulli Shift`.

**Simulated Shifts.**  We simulate four kinds of label shift patterns to capture different kinds of non-stationary environments. For each shift, the priors are a mixture of two different constant priors $\boldsymbol{\mu}_1, \boldsymbol{\mu}_2 \in \Delta_K$ with a time-varying coefficient $\alpha_t$: $\boldsymbol{\mu}_{y_t} = (1 - \alpha_t)\boldsymbol{\mu}_1 + \alpha_t\boldsymbol{\mu}_2$, where $\boldsymbol{\mu}_{y_t}$ denotes the label distribution at round $t$ and $\alpha_t$ controls the shift non-stationarity and patterns. We list the details in the following.

- `Linear Shift` (Lin): the parameter $\alpha_t = \frac{t}{T}$, which represents the gradually changed environments.
- `Square Shift` (Squ): the parameter $\alpha_t$ switches between 1 and 0 every $L/2$ rounds, where $L$ is the periodic length. In the experiments, we set $L = \Theta(\sqrt{T})$ by default, which implies that the fluctuation of the class prior $V_T = \sum_{t=2}^{T}\|\boldsymbol{\mu}_{y_t} - \boldsymbol{\mu}_{y_{t-1}}\|_1$ is $\Theta(\sqrt{T})$. The `Square Shift` simulates a fast-changing environment with periodic patterns.
- `Sine Shift` (Sin): $\alpha_t = \sin\frac{i\pi}{L}$, where $i = t \bmod L$ and $L$ is a given periodic length. In the experiments, we set $L = \Theta(\sqrt{T})$ by default. The `Sine Shift` also simulates a fast-changing environment with periodic patterns.
- `Bernoulli Shift` (Ber): At every iteration, we keep the $\alpha_t = \alpha_{t-1}$ with probability $p \in [0,1]$ and otherwise set $\alpha_t = 1 - \alpha_{t-1}$. In the experiments, the parameter is set as $p = 1/\sqrt{T}$ by default, which implies $V_t = \Theta(\sqrt{T})$. The `Bernoulli Shift` simulates a fast-changing environment without periodic patterns.

Figure 4 demonstrates how $\alpha_t$ changes over time. We can observe that `Square Shift` and `Sine Shift` change in a periodic pattern while the others do not. Moreover, it can be seen that `Linear Shift` has simulated a much more tender class prior change than the other three shifts.

**Datasets.** We conduct experiments on synthetic data, six real-world datasets, and a real-life application of locomotion detection.

*Synthetic data.* There are three classes in the synthetic data, where the feature distribution of each class follows a Gaussian distribution. More specifically, for each instance $(\mathbf{x}_i, y_i)$ in the dataset $S_t$ at round $t$, the label is generated from the discrete distribution defined by the given priors $\mathcal{D}_t(y)$, and the feature $\mathbf{x} \in \mathbb{R}^{12}$ is generated from the corresponding multivariate normal distributions $\mathcal{N}(\mu_y, \Sigma)$. We set $\boldsymbol{\mu}_1 = [1/3, 1/3, 1/3]$ and $\boldsymbol{\mu}_2 = [1, 0, 0]$ for simulated shifts.

*Real-world datasets.* We conduct experiments on six real-world datasets. For each dataset, we set $\boldsymbol{\mu}_1 = [1/K, 1/K, ..., 1/K]$ and $\boldsymbol{\mu}_2 = [1, 0, ..., 0]$ to generate the simulated shifts. Specifically, we include the following datasets.

- **ArXiv** [5]: A paper classification dataset which contains metadata of the scholarly articles. We select 296, 708 papers from the computer science domain, which cover 23 classes, including cs.AI, cs.CC, cs.CL, cs.CR, cs.CV, cs.CY, cs.DB, cs.DC, cs.DM, cs.DS, cs.GT, cs.HC, cs.IR, cs.IT, cs.LG, cs.LO, cs.NE, cs.NI, cs.PL, cs.RO, cs.SE, cs.SI, and cs.SY. [6] We use a finetuned DistilBERT [40] to extract features from authors, titles and abstracts. The papers selected additionally to finetune the DistilBERT do not overlap with the offline and online datasets.
- **EuroSAT** [41]: A land cover classification dataset, which includes satellite images with the purpose of identifying the visible land use or land cover class. The dataset consists of 27, 000 satellite images from over 30 different European countries. It contains ten different classes, including industrial, residential, annual crop, permanent crop, river, sea and lake, herbaceous vegetation, highway, pasture, and forest. We use a finetuned ResNet [42] to extract features from the images of EuroSAT and the following four datasets. The images selected to train the ResNet also do not overlap with both the offline or online datasets.
- **MNIST** [43]: A widely-used image dataset of handwritten digits, which consists of 70, 000 grayscale images with 10 different classes.
- **Fashion** [44]: A dataset of 70, 000 grayscale fashion images, consisting of 10 different classes: T-shirt, trouser, shirt and sneaker, pullover, dress, coat, sandal, bag, and ankle boot.
- **CIFAR10** [45]: A dataset consists of 60, 000 color images in 10 classes, including airplane, automobile, ship, truck, bird, cat, deer, dog, frog, and horse.
- **CINIC10** [46]: A tiny ImageNet [47] dataset, which consists of images from CIFAR10 and ImageNet, and has the same ten classes as CIFAR10.

*Real-life application.* The real-life application is to distinguish human locomotion through the sensor data collected by the carry-on mobile phones[7]. The tabular data covers the sensor data (e.g., acceleration, gyroscope, magnetometer, orientation, gravity, pressure, altitude, and temperature) and

---

[5] www.kaggle.com/datasets/Cornell-University/arxiv
[6] See www.arxiv.org/archive/cs for the full name.
[7] www.shl-dataset.org

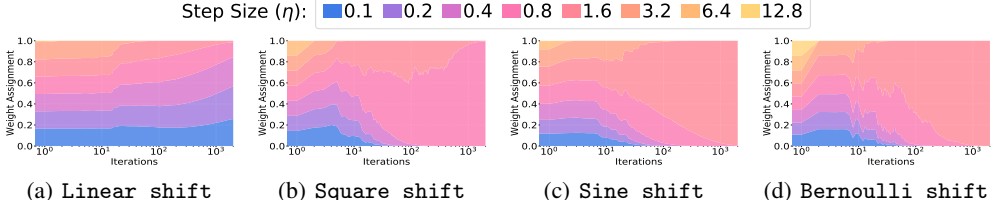

(a) Linear shift  (b) Square shift  (c) Sine shift  (d) Bernoulli shift

**Figure 5:** Weight assigned of the ATLAS algorithm for each step size along the learning process. Different colors are used to indicate different step sizes.

the corresponding human motion and timestamp. We sample 30, 000 offline data and 77, 000 online data from 11 days, covering six classes, including still, walking, run, bike, car, and bus. During the online update, the online samples arrive in real chronological order based on the timestamp.

### A.3  Additional Experimental Results

This part further reports the additional experimental results omitted in the main paper. Specifically, we show a supplement for the assigned weight visualization of ATLAS and a skyline to illustrate how the meta-algorithm tracks the proper base-algorithms.

**Weight Assignment of ATLAS.** In Figure 1, we show that slower Linear Shift is assigned relatively small step sizes, while faster Bernoulli Shift is assigned relatively large step sizes. Here, we replenish the experimental results for Square Shift and Sine Shift. As these two simulated shifts also change relatively fast like Bernoulli Shift, the assigned weights are also expected to be relatively large, which is in line with the experimental results in Figure 5(b) and 5(c).

**Skyline of ATLAS-ADA.** The purpose of this part is to answer the question of whether our proposed ATLAS-ADA can cope with online label shifts, regardless of how fast or slow the change is. To see this, we use the SHL dataset to simulate shifts with different non-stationarities, increasing from left to right side. We plot the final average errors of the major base-learners and ATLAS-ADA in Figure 6. Note that base-learners with much smaller or larger step sizes lead to worse results for all situations and are omitted for clarity. As shown in Figure 6, small step sizes achieve the best average errors for tender non-stationarity cases (left side) but fail for dramatic cases (right side), which is consistent with the theoretical results that the optimal step size scales with the non-stationarity measure $V_T$. Moreover, we find that the

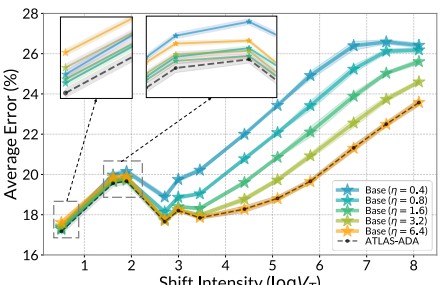

**Figure 6:** Skyline on the SHL dataset. The intensity of the label distribution shift increases from left to right. The black dashed line represents the average error of ATLAS-ADA, while the other lines represent that of different base-learners.

proposed ATLAS-ADA can always track the best base-learners, no matter tender cases or dramatic cases, which implies that we can safely employ it without knowing the change of the unknown environment in advance.

## B  Preliminaries

In this section, we first provide a brief introduction to the online convex optimization framework in Appendix B.1, and then in Appendix B.2 we present a detailed description of the BBSE method that is designed for the offline label shift problem, as well as its usage in constructing our risk estimator for the online label shift problem.

### B.1  Brief Introduction to Online Convex Optimization

This section briefly introduces the online convex optimization framework. In many real-world tasks, data are often accumulated in a sequential way, which requires a learning paradigm to update the model in an online fashion. Taking the locomotion detection task as an example, each time, we

---

**Algorithm 3** Online Gradient Descent (OGD)

---

**Input:** step size $\eta > 0$
1: Initialization: let $\mathbf{w}_{1,i}$ be any point in $\mathcal{W}$.
2: **for** $t = 1$ **to** $T$ **do**
3:    the learner plays $\mathbf{w}_t$ and observes loss $R_t(\mathbf{w}_t)$ and gradient $\nabla R_t(\mathbf{w}_t)$;
4:    gradient-descent update:
$$\mathbf{w}_{t+1} = \Pi_{\mathcal{W}} \left[ \mathbf{w}_t - \eta \nabla R_t(\mathbf{w}_t) \right].$$

5: **end for**

---

receive a few data collected by smartphone sensors and need to predict the motion type immediately. The sequential nature of the data makes it challenging to train a model offline using the standard machine learning methods. These circumstances can be modeled and handled with the Online Convex Optimization (OCO) framework [20, 15, 48].

The OCO framework can be viewed as a structured repeated game between a learner and the environment. During the learning process, the learner iteratively chooses decisions from a fixed convex decision set $\mathcal{W}$ based on the feedback from the environments. Let $T$ denote the total number of game iterations, and the protocol of OCO is given as follows.

---

ONLINE CONVEX OPTIMIZATION FRAMEWORK

1: **for** $t = 1$ **to** $T$ **do**
2:    the learner chooses a decision $\mathbf{w}_t \in \mathcal{W}$, and the environments simultaneously choose a convex loss function $R_t : \mathcal{W} \mapsto \mathbb{R}$;
3:    the learner suffers cost $R_t(\mathbf{w}_t)$ and observes certain information about $R_t$.
4: **end for**

---

The classic performance measure for the OCO algorithms is the (static) regret defined as

$$\mathbf{Reg}_T^{\mathsf{s}} = \sum_{t=1}^{T} R_t(\mathbf{w}_t) - \sum_{t=1}^{T} R_t(\mathbf{w}_*),$$

which compares the learner's prediction with the best single decision in hindsight $\mathbf{w}_* \in \arg\min_{\mathbf{w} \in \mathcal{W}} \sum_{t=1}^{T} R_t(\mathbf{w})$. The static regret is a reasonable measure for stationary environments. But the performance measure could be too optimistic in non-stationary environments, where the underlying distribution changes over time, and the single best model could perform badly. Under such a case, a more suitable performance is the *dynamic regret* that competes with the online learner's performance with the best decisions for each round. A more detailed discussion and literature review for the dynamic regret can be found in Appendix C.2.

The OCO framework has received extensive studies over the decades. Among them, one of the most prominent algorithms is the Online Gradient Descent (OGD) algorithm [20], which takes a step in the opposite direction to the gradient of the previous risk at each iteration (See Algorithm 3 for the procedures). Despite its simplicity, OGD algorithm is powerful enough to handle a large family of online problems. Specifically, it enjoys a static regret of $\mathcal{O}(\sqrt{T})$ by setting the step size to be $\Theta(1/\sqrt{T})$, which was proven to be minimax optimal [49] for convex functions. Latter, tighter static bounds for loss functions with stronger curvature [50] and that can adapt to benign environments [51, 35, 36] were proposed. For more detailed reviews of the OCO framework, we refer the readers to the seminal books [48, 15].

## B.2 BBSE Method for Online Label Shift

The Black Box Shift Estimation (BBSE) method [12] is a family of label shift algorithms that uses the confusion matrix of a given black-box classifier to estimate the label distributions from unlabeled samples. This part illustrates how to employ BBSE to the online label distribution estimation.

Recall that we have the labeled offline data $S_0$ and the unlabeled data $S_t$, and we need to estimate the label distribution $\boldsymbol{\mu}_{y_t}$. Note that $\boldsymbol{\mu}_{y_t}$ cannot be estimated directly due to labels of $S_t$ being unavailable. Since $S_0$ is labeled, a possible way is to employ the labels of $S_0$. To this end, we introduce an initial

black-box model $f_0$ and have the following equation:

$$\mathbb{E}_{\mathbf{x} \sim \mathcal{D}_t(\mathbf{x})} \left[ \mathbb{1}\{f_0(\mathbf{x}) = i\} \right] = \sum_{j=1}^{K} [\boldsymbol{\mu}_{y_t}]_j \cdot \mathbb{E}_{\mathbf{x} \sim \mathcal{D}_t(\mathbf{x} \mid y=j)} \left[ \mathbb{1}\{f_0(\mathbf{x}) = i\} \right]$$

$$= \sum_{j=1}^{K} [\boldsymbol{\mu}_{y_t}]_j \cdot \mathbb{E}_{\mathbf{x} \sim \mathcal{D}_0(\mathbf{x} \mid y=j)} \left[ \mathbb{1}\{f_0(\mathbf{x}) = i\} \right], \tag{7}$$

which holds for all $i \in [K]$. The first equality holds due to the law of total probability and the second equality is based on Assumption 1 that $\mathcal{D}_t(\mathbf{x} \mid y) = \mathcal{D}_0(\mathbf{x} \mid y)$ for any $t \in [T]$. (7) can be equivalently expressed in matrix notation, which is given by:

$$\boldsymbol{\mu}_{\widehat{y}_t} = C_{f_0} \boldsymbol{\mu}_{y_t}, \tag{8}$$

where $\boldsymbol{\mu}_{\widehat{y}_t} \in \Delta_K$ is the distribution vector of the black-box model's prediction under the distribution $\mathcal{D}_t(\mathbf{x})$, with the $i$-th entry $[\boldsymbol{\mu}_{\widehat{y}_t}]_i = \mathbb{E}_{\mathbf{x} \sim \mathcal{D}_t(\mathbf{x})} \left[ \mathbb{1}\{f_0(\mathbf{x}) = i\} \right]$. $C_{f_0} \in \mathbb{R}^{K \times K}$ is the confusion matrix, whose $(i, j)$-th entry $[C_{f_0}]_{ij} = \mathbb{E}_{\mathbf{x} \sim \mathcal{D}_0(\mathbf{x} \mid y=j)}[\mathbb{1}\{f_0(\mathbf{x}) = i\}]$ is the classification rate that the initial model $f_0$ predicts samples from class $i$ as class $j$. Here we assume the confusion matrix $C_{f_0}$ is invertible. By solving (8), we have

$$\boldsymbol{\mu}_{y_t} = C_{f_0}^{-1} \boldsymbol{\mu}_{\widehat{y}_t}. \tag{9}$$

The LHS of (9) $\boldsymbol{\mu}_{y_t}$ is the label distribution we want. In the RHS, we note that the distribution $\mathcal{D}_t(\mathbf{x})$ and $\mathcal{D}_0(\mathbf{x} \mid y)$ is empirically accessible via the unlabeled data $S_t$ and the labeled data $S_0$, so both $C_{f_0}$ and $\boldsymbol{\mu}_{\widehat{y}_t}$ can be unbiasedly estimated with its empirical version. Specifically, $C_{f_0}$ can be estimated with the offline labeled data $S_0$:

$$[\widehat{C}_{f_0}]_{ij} = \sum_{(\mathbf{x}, y) \in S_0} \frac{\mathbb{1}\{f_0(\mathbf{x}) = i \text{ and } y = j\}}{\mathbb{1}\{y = j\}}. \tag{10}$$

And $\boldsymbol{\mu}_{\widehat{y}_t}$ in the RHS can be estimated with the online data $S_t$, which is given by

$$[\widehat{\boldsymbol{\mu}}_{\widehat{y}_t}]_j = \frac{1}{|S_t|} \sum_{\mathbf{x} \in S_t} \mathbb{1}\{f_0(\mathbf{x}) = j\}. \tag{11}$$

With the above estimation, we construct the final estimator for the label distribution vector:

$$\widehat{\boldsymbol{\mu}}_{y_t} = \widehat{C}_{f_0}^{-1} \widehat{\boldsymbol{\mu}}_{\widehat{y}_t}.$$

Note that although BBSE has benign unbiasedness, it has limitations arising from the inverse operation, which might amplify slight errors in the confusion matrix. Other powerful unbiased estimators can also be applied to our OLaS method directly.

## C  Related Work and Discussion

This section introduces related works to our paper, including supervised learning with label shift in Appendix C.1, and online learning measures for non-stationary environments in Appendix C.2. Furthermore, in Appendix C.3, we clarify the main difference between Wu et al. [16] and our work.

### C.1  Supervised Learning with Label Shift

Label shift is a typical scenario of learning with distribution change problem [1]. Most existing works have focused on the offline setting, where the label distribution varies from the source to target stages, but the label-conditional density remains the same. The main challenge of offline label shift problem lies in the estimation of the target label distribution. With such knowledge, the learner can train classifiers guaranteed to perform well over the target distribution. Saerens et al. [8] first introduced two kinds of solutions for label distribution estimation, including the confusion matrix-based method and the maximum likelihood estimation. Lipton et al. [12] provided a convergence analysis of the matrix-based method with black box models and Azizzadenesheli et al. [13] enhanced the matrix-based method by regularization. Another line of studies investigates the label distribution estimation problem via distribution matching [9, 10, 11, 14]. Garg et al. [14] further showed that the

matrix-based method can be interpreted from the distribution matching view. Beyond the label shift, the label distribution estimation with labeled data from the source domain and unlabeled data from the target domain is also known as the mixture proportion estimation problem [52, 53, 54], which has been wildly applied in learning with noisy label [55], positive and unlabeled learning [56, 57], and learning with unknown classes [58], etc.

The above methods only considered the scenario where the label distribution changes once, and sufficient unlabeled samples from the target distribution can be collected in advance. In many real-world applications, data are accumulated with time, and thus it is important to consider the online version of the label shift problem. Wu et al. [16] first introduced online label shift and considered the scenario where test samples come in sequence and label distribution changes with time. To address the OLaS problem, Wu et al. [16] proposed a risk estimator and used the vanilla OGD algorithm to learn over the online streams with label shift. They derived a static regret of order $\mathcal{O}(\sqrt{T})$, but the analysis crucially relied on the assumption of the convexity of the (expected) risk function, which is hard to be theoretically verified due to the use of 0/1-loss and the argmax operation. In contrast, by constructing an unbiased risk estimator with nice properties, our methods can adapt to dynamic environments with provable grantees. We provide more detailed discussions on the differences between the risk estimators in Appendix C.3.

## C.2 Online Learning in Non-stationary Environments

This section first reviews existing results for non-stationary online convex optimization, including the worst-case dynamic regret and the universal dynamic regret. Then, we discuss the difference between our dynamic regret bounds and those in the previous works. We also present a brief introduction to readers unfamiliar with the OCO paradigm in Appendix B.1.

**Worst-case Dynamic Regret.** The rationale behind the static regret is that the best single decision could perform well over the time horizon. However, the underlying environments could change over the online learning process, making the static regret unsuitable. A better measure for the non-stationary environments is the worst-case dynamic regret $\mathbf{Reg}_T^{\mathsf{d}} = \sum_{t=1}^{T} R_t(\mathbf{w}_t) - \sum_{t=1}^{T} R(\mathbf{w}_t^*)$, which compares the learner's decision with the best decision at each iteration $\mathbf{w}_t^* \in \arg\min_{\mathbf{w} \in \mathcal{W}} R_t(\mathbf{w})$ and has draw growing attention recently [20, 17, 27, 59, 60, 61, 62, 63, 28, 64].

It is well-known that a sublinear regret bound is impossible for the dynamic regret in the worst case unless imposing certain regularities on the non-stationarity of the environments. There are two commonly studied non-stationarity measures. The first one is the *path length* $P_T^* = \sum_{t=2}^{T} \|\mathbf{w}_t^* - \mathbf{w}_{t-1}^*\|_2$, which reflects the fluctuation of the comparator sequence. When the loss functions are convex, Yang et al. [59] showed that OGD algorithm attains an $\mathcal{O}(\sqrt{T(1 + P_T^*)})$ dynamic regret. The bound can be further improved to $\mathcal{O}(P_T^*)$ when the functions are convex and smooth, and the minimizer $\mathbf{w}_t^*$ lies in the interior of the feasible domain $\mathcal{W}$ [59], or the loss functions are strongly convex and smooth [65]. Another commonly studied non-stationarity measure is the *temporal variability* $V_T^f = \sum_{t=2}^{T} \sup_{\mathbf{w} \in \mathcal{W}} |f_t(\mathbf{w}) - f_{t-1}(\mathbf{w})|$, which quantifies the variation of the online functions. When the online functions are convex, Besbes et al. [17] showed an $\mathcal{O}(T^{2/3} V_T^{f\,1/3})$ dynamic regret by the restarted OGD algorithm, and the bound can be further improved to $\mathcal{O}(T^{1/2} V_T^{f\,1/2})$ for strongly convex functions. Then, the temporal variability bound was generalized to high-order deviations [62] and the $p, q$-norm [63]. There are also studies on the best-of-both-worlds bounds [27, 28, 64], where the path-length bound and temporal variability bound are achieved simultaneously by a single algorithm. For strongly convex and smooth functions, the best known result is $\mathcal{O}(\min\{P_T^*, S_T^*, V_T^f\})$ due to Zhao and Zhang [64], where $S_T^* = \sum_{t=2}^{T} \|\mathbf{w}_t^* - \mathbf{w}_{t-1}^*\|_2^2$ is the squared path length.

**Universal Dynamic Regret.** The worst-case dynamic regret will sometimes be too pessimistic to guide the algorithm design as the function minimizer $\mathbf{w}_t^*$ can easily overfit to the noise [18]. To this end, the universal dynamic regret is proposed and defined as $\mathbf{Reg}_T^{\mathsf{d}}(\mathbf{u}_1, \dots, \mathbf{u}_T) = \sum_{t=1}^{T} R_t(\mathbf{w}_t) - \sum_{t=1}^{T} R_t(\mathbf{u}_t)$, which supports the comparison to any feasible comparator sequence $\{\mathbf{u}_t\}_{t=1}^{T}$ with $\mathbf{u}_t \in \mathcal{W}$, and the measure has gained more and more attention [20, 18, 19, 66, 67, 68, 69, 70, 71, 72, 73, 74]. Zinkevich [20] showed that OGD can achieve an $\mathcal{O}((1 + P_T)\sqrt{T})$ universal dynamic regret, where $P_T = \sum_{t=2}^{T} \|\mathbf{u}_t - \mathbf{u}_{t-1}\|_2$ is the path length of the comparator sequence $\{\mathbf{u}_t\}_{t=1}^{T}$. Nevertheless,

the guarantee is far from the $\Omega(\sqrt{T(1+P_T)})$ minimax lower bound for convex functions as shown by Zhang et al. [18], who further developed an algorithm with an $\mathcal{O}(\sqrt{T(1+P_T)})$ universal dynamic regret and hence closed the gap [18]. Later, many studies are devoted to developing algorithms with tighter guarantees that can benefit from stronger curvature of loss functions [70], or can adapt to the benign environments [19, 68], or to the more challenging setting with bandit feedback [69].

**Discussion on Our Dynamic Regret Bounds.** Although there are many studies on non-stationary online learning, it is hard to apply existing methods to OLaS, and our regret bound is novel. First of all, although our performance measure shares seemingly similarity with the worst-case dynamic regret, the algorithm with such bounds is not suitable for the OLaS problem. The main reason is that existing methods require access to the loss function $R_t$ for the model update. However, in the OLaS problem, the true loss function $R_t$ is established on the inaccessible data distribution $\mathcal{D}_t$ and the learner can only approximate it with $\widehat{R}_t$ based on the empirical data. A direct application of algorithms with a worst-case dynamic regret can only lead to regret bounds defined over $\widehat{R}_t$ such that the learned model would suffer from severe overfitting on the sample noise. A few previous works [17, 59, 62] have attempted to develop methods to learn with the noisy loss function, but nevertheless the methods in [17, 59] require knowing the non-stationarity of the underlying distribution $\mathcal{D}_t$ ahead of time, which is generally unavailable.

Given that the comparators in our regret measure are *not* the optimizers of the loss functions, our dynamic regret measure is essentially a kind of *universal* dynamic regret rather than the worst-case dynamic regret. Our results differ from the existing universal dynamic regret bounds appearing in the literature by carefully exploiting the structure of online label shift. In particular, existing bounds always scale with the path length of compactors $P_T = \sum_{t=2}^{T} \|\mathbf{u}_t - \mathbf{u}_{t-1}\|_2$, whereas our dynamic regret can adapt to the variation of label distribution $V_T = \sum_{t=2}^{T} \|\boldsymbol{\mu}_{y_t} - \boldsymbol{\mu}_{y_{t-1}}\|_1$ directly, which is a more interpretable and suitable measure for OLaS.

From the technical side, the most related works to us are [18, 19] as all algorithms use the meta-base structure to achieve the dynamic regret bound. Besides the difference in the non-stationarity measure, our method improves the regret bound of Zhang et al. [18] for convex functions by replacing the dependence on $T$ with the problem-dependent quantity $G_T$. The $G_T$ quantity can be much smaller than $T$ in benign environments, which implies a better adaptivity of our algorithm to underlying environments. The first dynamic regret bound with such adaptivity was achieved by Zhao et al. [19], but their work requires an additional smoothness assumption on the loss functions. We remove such a constraint via refined analyses for the meta-algorithm, which updates the weights with function values and an improved algorithm design for base-algorithm, which trains the model following an implicit update rule. The new finding might be of independent interest to the general online convex optimization. We note that a similar implicit update rule also appeared in a concurrent work [75, Section 5] for achieving problem-dependent *static* regret. Our dynamic regret is much more challenging, as we not only need an implicit update in base-algorithms but also need a refined analysis of the meta-algorithm, which is unique in online ensemble analysis of dynamic regret minimization.

## C.3   Detailed Comparison with Wu et al. [16]

As mentioned in the main paper the work of Wu et al. [16] is the closest one to our proposal, who also proposed a risk estimator to cope with OLaS. In this part, we first review their result by restating their proposed risk estimator and then elaborate on the salient differences between their method and ours. The comparison demonstrates both theoretical and practical appeals of our proposed risk estimator in that it makes the OCO framework still applicable to OLaS.

**Estimation of $0/1$ risk.** The performance measure of Wu et al. [16] is slightly different from ours, where the authors aim to minimize the regret defined over the $0/1$ risk

$$\mathbf{Reg}_T^{01} = \sum_{t=1}^{T} R_t^{01}(g_t) - \min_{g \in \mathcal{G}} \sum_{t=1}^{T} R_t^{01}(g).$$

In above, the $0/1$ risk is defined as $R_t^{01}(g) = \mathbb{E}_{(\mathbf{x},y) \sim \mathcal{D}_t}[\mathbb{1}\{g(\mathbf{x}) \neq y\}]$ and $g : \mathcal{X} \mapsto \mathcal{Y}$ is the classifier whose definition will be clear latter. Denoting by $\boldsymbol{\mu}_{y_t}$ the class prior vector with

$[\boldsymbol{\mu}_{y_t}]_k = \mathcal{D}_t(y = k)$, the expected $0/1$ risk can be rewritten as

$$R_t^{01}(g) = \sum_{k=1}^{K} \mathbb{E}_{\mathbf{x} \sim \mathcal{D}_t(\mathbf{x} \mid y=k)} \left[ \mathbb{1}\{g(\mathbf{x}) \neq k\} \right] \cdot [\boldsymbol{\mu}_{y_t}]_k$$

$$= \sum_{k=1}^{K} \mathbb{E}_{\mathbf{x} \sim \mathcal{D}_0(\mathbf{x} \mid y=k)} \left[ \mathbb{1}\{g(\mathbf{x}) \neq y\} \right] \cdot [\boldsymbol{\mu}_{y_t}]_k. \tag{12}$$

In decomposition (12), the first term $\mathbb{E}_{\mathbf{x} \sim \mathcal{D}_0(\mathbf{x} \mid y=k)}[\mathbb{1}\{g(\mathbf{x}) \neq y\}]$ is established on the class-conditional distribution $\mathcal{D}_0(\mathbf{x} \mid y)$, which can be empirically approximated with the offline data $S_0$. The second term is the class prior at each iteration. One can estimate it with the BBSE estimator introduced in Section 3. As a consequence, Wu et al. [16] empirically approximated $R_t^{01}(g)$ by

$$\widehat{R}_t^{01}(g) = \sum_{k=1}^{K} [\widehat{\boldsymbol{\mu}}_{y_t}]_k \cdot \widehat{R}_{t,k}^{01}(g),$$

where $\widehat{\boldsymbol{\mu}}_{y_t} = C_{f_0}^{-1} \cdot \widehat{\boldsymbol{\mu}}_{\widehat{y}_t}$ is the estimated class prior returned by the BBSE model and $\widehat{R}_{t,k}^{01} = \frac{1}{|S_0^k|} \sum_{\mathbf{x}_n \in S_0^k} \mathbb{1}\{g(\mathbf{x}_n) \neq k\}$ is the model's empirical risk over $S_0^k$.

**Reweighting classifier.** To directly optimize the $0/1$ loss, Wu et al. [16] further restricted the model's structure by focusing on the reweighting algorithm. The basic idea is, the optimal classifier that minimizes $R_t^{01}(g)$ is $g_*(\mathbf{x}) = \arg\max_{k \in [K]} \mathcal{D}_t(y = k \mid \mathbf{x})$. Under the class shift condition, they rewrote $\mathcal{D}_t(y = k \mid \mathbf{x})$ as

$$\mathcal{D}_t(y = k \mid \mathbf{x}) = \frac{\mathcal{D}_t(y=k)\mathcal{D}_t(\mathbf{x} \mid y=k)}{\mathcal{D}_t(\mathbf{x})} = \frac{\mathcal{D}_t(y=k)\mathcal{D}_0(\mathbf{x} \mid y=k)}{\mathcal{D}_t(\mathbf{x})}$$

$$= \frac{\mathcal{D}_t(y=k)}{\mathcal{D}_0(y=k)} \frac{\mathcal{D}_0(\mathbf{x})}{\mathcal{D}_t(\mathbf{x})} \mathcal{D}_0(y = k \mid \mathbf{x}) \propto \frac{\mathcal{D}_t(y=k)}{\mathcal{D}_0(y=k)} \mathcal{D}_0(y = k \mid \mathbf{x}). \tag{13}$$

Since the $\mathcal{D}_0$ can be approximate by the offline data $S_0$, the leaner can train a probability model $g_0 : \mathcal{X} \mapsto \Delta_K$ on $S_0$ to approximate $\mathcal{D}_0(y \mid \mathbf{x})$. Then, by the relationship (13), one can construct a classifier $g_{\mathbf{p}} : \mathcal{X} \mapsto [K]$ parameterized by the reweighting parameter $\mathbf{p} \in \Delta_K$ to approximate the optimal classifier $g_*$ by

$$g_{\mathbf{p}}(\mathbf{x}) = \arg\max_{y \in [K]} \frac{1}{Z(\mathbf{x})} \frac{[\mathbf{p}]_k}{\mathcal{D}_0(y=k)} \cdot [g_0(\mathbf{x})]_k, \tag{14}$$

where $Z(\mathbf{x}) = \sum_{k=1}^{K} \frac{[\mathbf{p}]_k}{\mathcal{D}_0(y=k)} \cdot [g_0(\mathbf{x})]_k$ is the normalization factor and the initial label distribution $\mathcal{D}_0(y = k)$ can be estimated with the offline data $S_0$. When $g_0$ provides a sufficiently accurate approximation of $\mathcal{D}_t(y \mid \mathbf{x})$, one can show that $g_* = \arg\min_{\mathbf{p} \in \Delta_K} R_t^{01}(g_{\mathbf{p}})$.

**Comparisons.** In summary, the differences between the risk estimator of Wu et al. [16] and ours are mainly in the following two aspects:

   (i) **Loss function.** As shown in (12), Wu et al. [16] used the non-convex $0/1$ loss $\mathbb{1}\{g(\mathbf{x}) \neq y\}$ for evaluation. Theoretically, the non-convexity of $0/1$ loss makes it hard to confirm the convexity of the $0/1$ risk, which is the critical condition for the regret bound. Practically, the gradient of $0/1$ loss is always zero, which brings troubles to the optimization. By contrast, we use a convex surrogate loss in our risk estimator to address the above problems.

   (ii) **Classifier.** To match the 0/1 loss function, Wu et al. [16] designed a classifier based on the reweighting algorithm, which directly updates the reweighting parameter $\mathbf{p}$ in a simplex. Theoretically, as shown in (14), there is a troublesome argmax operation in the formulation of the classifier, which makes it non-convex and then hurts the convexity of the entire risk. Practically, only adjusting the outputs of the initial classifier $f_0^{\mathbf{p}}$ by reweighting might limit the learning capability of the model, especially when the initial classifier does not perform well enough. By comparison, we employ a strictly convex classifier and update the whole parameters in a convex parameter space, which confirms the convexity of the risk and the flexibility to better adapt to the environmental changes. The classifier in (14) is hard to be convex even if we replace the 0/1 loss with convex surrogate loss (details elaborated below).

We finally remark that even if we modify the risk estimator of Wu et al. [16] by optimizing the convex surrogate loss rather than the $0/1$ loss, it remains unclear how to apply the OCO framework to this modified estimator. The critical challenge is that even if the argmax operation can be avoided, another renormalization process is required due to the reweighting mechanism, making it hard to confirm the convexity of online functions. By contrast, our risk estimator well fits the OCO framework and thus can achieve favorable dynamic regret guarantees with suitable online update rules.

## D  Omitted Details for Section 3.1

This section presents the omitted details for Section 3.1.

### D.1  Proof of Lemma 1

*Proof of Lemma 1.*  Recall the definition of the expected risk (2) as

$$R_t(\mathbf{w}) = \sum_{k=1}^{K} [\boldsymbol{\mu}_{y_t}]_k \cdot R_0^k(\mathbf{w}),$$

where $\boldsymbol{\mu}_{y_t}$ is the true class prior at iteration $t$ with $[\boldsymbol{\mu}_{y_t}]_k = \mathcal{D}(y = k)$. Besides, the risk estimator (3) is defined by

$$\widehat{R}_t(\mathbf{w}) = \sum_{k=1}^{K} [\widehat{C}_{f_0}^{-1} \widehat{\boldsymbol{\mu}}_{\widehat{y}_t}]_k \cdot \widehat{R}_0^k(\mathbf{w}),$$

where $\widehat{\boldsymbol{\mu}}_{\widehat{y}_t} \in \Delta_K$ with $[\widehat{\boldsymbol{\mu}}_{\widehat{y}_t}]_k = (1/|S_t|) \sum_{\mathbf{x} \in S_t} \mathbb{1}\{f_0(\mathbf{x}) = k\}$ is the estimated class prior of the prediction $f_0(\mathbf{x})$.

Given that the initial offline data has sufficient samples such that $\widehat{C}_{f_0} = C_{f_0}$ and $\widehat{R}_0^k(\mathbf{w}) = R_0^k(\mathbf{w})$ and the model $\mathbf{w} \in \mathcal{W}$ is independent of $S_t$, we have

$$
\begin{aligned}
\mathbb{E}_{S_t \sim \mathcal{D}_t}[\widehat{R}_t(\mathbf{w})] &= \mathbb{E}_{S_t \sim \mathcal{D}_t} \left[ \sum_{k=1}^{K} [\widehat{C}_{f_0}^{-1} \widehat{\boldsymbol{\mu}}_{\widehat{y}_t}]_k \cdot \widehat{R}_0^k(\mathbf{w}) \right] \\
&= \sum_{k=1}^{K} \left[ \widehat{C}_{f_0}^{-1} \mathbb{E}_{S_t \sim \mathcal{D}_t} [\widehat{\boldsymbol{\mu}}_{\widehat{y}_t}] \right]_k \cdot \widehat{R}_0^k(\mathbf{w}) \\
&= \sum_{k=1}^{K} \left[ \widehat{C}_{f_0}^{-1} \boldsymbol{\mu}_{\widehat{y}_t} \right]_k \cdot \widehat{R}_0^k(\mathbf{w}),
\end{aligned}
$$

where the first equality is due to the definition of $\widehat{R}_t(\mathbf{w})$. The second inequality comes from the fact that $\widehat{C}_{f_0}$ and $\widehat{R}_0^k$ are independent of $S_t$. The last equality is a consequence of the unbiasedness of $\widehat{\boldsymbol{\mu}}_{\widehat{y}_t}$, i.e., $\mathbb{E}_{S_t \sim \mathcal{D}_t}[\widehat{\boldsymbol{\mu}}_{\widehat{y}_t}] = \boldsymbol{\mu}_{\widehat{y}_t}$, where $[\boldsymbol{\mu}_{\widehat{y}_t}]_k = \mathbb{E}_{\mathbf{x} \sim \mathcal{D}_t(\mathbf{x})}[\mathbb{1}\{f_0(\mathbf{x}) = k\}]$ is the expected class prior of the prediction $f_0(\mathbf{x})$.

Then, under the condition that $\widehat{C}_{f_0} = C_{f_0}$ and $\widehat{R}_0^k(\mathbf{w}) = R_0^k(\mathbf{w})$, we can further obtain

$$\mathbb{E}_{S_t \sim \mathcal{D}_t}[\widehat{R}_t(\mathbf{w})] = \sum_{k=1}^{K} \left[ C_{f_0}^{-1} \boldsymbol{\mu}_{\widehat{y}_t} \right]_k \cdot R_0^k(\mathbf{w}) = \sum_{k=1}^{K} [\boldsymbol{\mu}_{y_t}]_k \cdot R_0^k(\mathbf{w}) = R_t(\mathbf{w}),$$

where the second equality is due to the relationship $\boldsymbol{\mu}_{y_t} = C_{f_0}^{-1} \boldsymbol{\mu}_{\widehat{y}_t}$ as introduced by (8). The last inequality comes from the definition of $R_t(\mathbf{w})$, which completes the proof. $\qquad \square$

### D.2  Concentration of Risk Estimator

In this section, we show that the risk estimator converges to the expected risk at the rate of $\mathcal{O}(\sqrt{1/|S_0|})$ with high probability.

**Lemma 2.** *Let $\delta \in (0, 1/4]$. For any $\mathbf{w} \in \mathcal{W}$ independent of the dataset $S_t$, with probability at least $1 - (K+3)\delta$, the risk estimator $\widehat{R}_t(\mathbf{w})$ in Eq. (3) satisfies*

$$|\mathbb{E}_{S_t \sim \mathcal{D}_t}[\widehat{R}_t(\mathbf{w})] - R_t(\mathbf{w})| \leq \frac{4\sqrt{K}B}{\kappa^2\sigma^2} \left( \frac{\log(2K/\delta)}{|S_0|} + \sqrt{\frac{2\log(2K/\delta)}{|S_0|}} \right),$$

*given that $|S_0| \geq (25\log(2K/\delta))/(\kappa\sigma)^2$, where $\kappa = \min_{k \in [K]} \mathcal{D}_0(y = k)$ and $\sigma$ is the minimum singular value of $C_{f_0}$.*

*Proof of Lemma 2.* For notation convenience, we denote by $\boldsymbol{R}_0 : \mathcal{W} \mapsto \mathbb{R}^K$ the vector risk function with its $k$-th entry $[\boldsymbol{R}_0(\mathbf{w})]_k = R_0^k(\mathbf{w})$ and its empirical version $\widehat{\boldsymbol{R}}_0 : \mathcal{W} \mapsto \mathbb{R}^K$ with $[\widehat{\boldsymbol{R}}_0(\mathbf{w})]_k = \widehat{R}_0^k(\mathbf{w})$. Under such a case, we can rewrite the risk estimator as $\widehat{R}_t(\mathbf{w}) = \widehat{\boldsymbol{\mu}}_{\widehat{y}_t}^\top \widehat{C}_{f_0}^{-1} \widehat{\boldsymbol{R}}_0(\mathbf{w})$ and the expected risk as $R_t(w) = \boldsymbol{\mu}_{\widehat{y}_t}^\top C_{f_0}^{-1} \boldsymbol{R}_0(\mathbf{w})$. Then, since $S_t$ is independent of $S_0$, we have

$$\mathbb{E}_{S_t \sim \mathcal{D}_t}[\widehat{R}_t(\mathbf{w})] = \mathbb{E}_{S_t \sim \mathcal{D}_t}[\widehat{\boldsymbol{\mu}}_{\widehat{y}_t}^\top \widehat{C}_{f_0}^{-1} \widehat{\boldsymbol{R}}_0(\mathbf{w})] = \mathbb{E}_{S_t \sim \mathcal{D}_t}[\widehat{\boldsymbol{\mu}}_{\widehat{y}_t}^\top]\widehat{C}_{f_0}^{-1} \widehat{\boldsymbol{R}}_0(\mathbf{w}) = \boldsymbol{\mu}_{\widehat{y}_t}^\top \widehat{C}_{f_0}^{-1} \widehat{\boldsymbol{R}}_0(\mathbf{w}).$$

Let $W_{f_0} \in \mathbb{R}^K$ be the diagonal matrix with $[W_{f_0}]_{ii} = \mathcal{D}_0(y = i)$ and $[J_{f_0}]_{ij} = \mathbb{E}_{(\mathbf{x}, y) \sim \mathcal{D}_0}[\mathbb{1}\{f_0(\mathbf{x}) = i \text{ and } y = j\}]$ be the confusion matrix defined over the joint distribution of $f_0(\mathbf{x})$ and $y$. Besides, denote by $[\widehat{W}_{f_0}]_{ii} = (1/|S_0|)\sum_{(\mathbf{x},y) \in S_0} \mathbb{1}\{y = i\}$ and $\widehat{J}_{f_0} = (1/|S_0|)\sum_{(\mathbf{x},y) \in S_0} \mathbb{1}\{f_0(\mathbf{x}) = i \text{ and } y = j\}$ the empirical versions. Definition (10) indicates that

$$J_{f_0} = W_{f_0} \cdot C_{f_0} \quad \text{and} \quad \widehat{J}_{f_0} = \widehat{W}_{f_0} \cdot \widehat{C}_{f_0}. \tag{15}$$

Under such a case, the empirical and expected risk can be further rewritten as

$$\mathbb{E}_{S_t \sim \mathcal{D}_t}[\widehat{R}_t(\mathbf{w})] = \boldsymbol{\mu}_{\widehat{y}_t}^\top \widehat{J}_{f_0}^{-1} \widehat{W}_{f_0} \widehat{\boldsymbol{R}}_0(\mathbf{w}) \quad \text{and} \quad R_t(\mathbf{w}) = \boldsymbol{\mu}_{\widehat{y}_t}^\top J_{f_0}^{-1} W_{f_0} \boldsymbol{R}_0(\mathbf{w}).$$

So, we have

$$\mathbb{E}_{S_t \sim \mathcal{D}_t}\left[\widehat{R}_t(\mathbf{w})\right] - R_t(\mathbf{w})$$
$$\leq \boldsymbol{\mu}_{\widehat{y}_t}^\top \widehat{J}_{f_0}^{-1} \widehat{W}_{f_0} \widehat{\boldsymbol{R}}_0(\mathbf{w}) - \boldsymbol{\mu}_{\widehat{y}_t}^\top J_{f_0}^{-1} W_{f_0} \boldsymbol{R}_0(\mathbf{w})$$
$$\leq \underbrace{\boldsymbol{\mu}_{\widehat{y}_t}^\top \widehat{J}_{f_0}^{-1} \widehat{W}_{f_0} \widehat{\boldsymbol{R}}_0(\mathbf{w}) - \boldsymbol{\mu}_{\widehat{y}_t}^\top J_{f_0}^{-1} \widehat{W}_{f_0} \widehat{\boldsymbol{R}}_0(\mathbf{w})}_{\texttt{term (a)}} + \underbrace{\boldsymbol{\mu}_{\widehat{y}_t}^\top J_{f_0}^{-1} \widehat{W}_{f_0} \widehat{\boldsymbol{R}}_0(\mathbf{w}) - \boldsymbol{\mu}_{\widehat{y}_t}^\top J_{f_0}^{-1} W_{f_0} \boldsymbol{R}_0(\mathbf{w})}_{\texttt{term (b)}}.$$

For term (a), with probability at least $1 - \delta$, we have

$$\texttt{term (a)} = \boldsymbol{\mu}_{\widehat{y}_t}^\top (J_{f_0}^{-1} - \widehat{J}_{f_0}^{-1})\widehat{W}_{f_0} \widehat{\boldsymbol{R}}_0(\mathbf{w})$$
$$\leq \|\boldsymbol{\mu}_{\widehat{y}_t}\|_2 \cdot \|\widehat{W}_{f_0} \widehat{\boldsymbol{R}}_0(\mathbf{w})\|_2 \cdot \|\widehat{J}_{f_0}^{-1} - J_{f_0}^{-1}\|_2$$
$$\leq \sqrt{K}B\|\widehat{J}_{f_0}^{-1} - J_{f_0}^{-1}\|_2$$
$$\leq \frac{2\sqrt{K}B}{\kappa^2\sigma^2} \left( \frac{\log(2K/\delta)}{|S_0|} + \sqrt{\frac{2\log(2K/\delta)}{|S_0|}} \right),$$

where the second inequality is due to the Cauchy-Schwarz inequality and the property of the operator norm. The third inequality holds since $\|\boldsymbol{\mu}_{\widehat{y}_t}\|_2 \leq 1$ and $\|\widehat{W}_{f_0} \widehat{\boldsymbol{R}}_0(\mathbf{w})\|_2 \leq \|\widehat{W}_{f_0}\|_2 \cdot \|\widehat{\boldsymbol{R}}_0(\mathbf{w})\|_2 \leq \sqrt{K}B$. The last inequality is due to Lemma 4.

Next, we analyze the term (b).

$$\texttt{term (b)} = \boldsymbol{\mu}_{\widehat{y}_t}^\top J_{f_0}^{-1}(\widehat{W}_{f_0} \widehat{\boldsymbol{R}}_0(\mathbf{w}) - W_{f_0} \boldsymbol{R}_0(\mathbf{w}))$$
$$\leq \|\boldsymbol{\mu}_{\widehat{y}_t}\|_2 \cdot \|J_{f_0}^{-1}\|_2 \cdot \|\widehat{W}_{f_0} \widehat{\boldsymbol{R}}_0(\mathbf{w}) - W_{f_0} \boldsymbol{R}_0(\mathbf{w})\|_2$$
$$\leq \frac{1}{\kappa\sigma}\|\widehat{W}_{f_0} \widehat{\boldsymbol{R}}_0(\mathbf{w}) - W_{f_0} \boldsymbol{R}_0(\mathbf{w})\|_2,$$

where the third inequality is due to the fact that $\|\boldsymbol{\mu}_{\widehat{y}_t}\|_2 \leq 1$ and $\|J_{f_0}^{-1}\|_2 \leq \|W_{f_0}^{-1}C_{f_0}^{-1}\|_2 \leq 1/(\kappa\sigma)$. The term $\|\widehat{W}_{f_0}\widehat{\boldsymbol{R}}_0(\mathbf{w}) - W_{f_0}\boldsymbol{R}_0(\mathbf{w})\|_2$ can be further decomposed as

$$
\begin{aligned}
\|\widehat{W}_{f_0}\widehat{\boldsymbol{R}}_0(\mathbf{w}) - W_{f_0}\boldsymbol{R}_0(\mathbf{w})\|_2 &= \|\widehat{W}_{f_0}\widehat{\boldsymbol{R}}_0(\mathbf{w}) - W_{f_0}\widehat{\boldsymbol{R}}_0(\mathbf{w})\|_2 + \|W_{f_0}\widehat{\boldsymbol{R}}_0(\mathbf{w}) - W_{f_0}\boldsymbol{R}_0(\mathbf{w})\|_2 \\
&\leq \|\widehat{\boldsymbol{R}}_0(\mathbf{w})\|_2 \cdot \|\widehat{W}_{f_0} - W_{f_0}\|_2 + \|W_{f_0}\|_2 \cdot \|\widehat{\boldsymbol{R}}_0(\mathbf{w}) - \boldsymbol{R}_0(\mathbf{w})\|_2 \\
&\leq B\|\widehat{W}_{f_0} - W_{f_0}\|_2 + \|\widehat{\boldsymbol{R}}_0(\mathbf{w}) - \boldsymbol{R}_0(\mathbf{w})\|_2,
\end{aligned}
$$

where the first inequality is due to the property of the operator norm. The second inequality is a consequence of the fact that $\|\widehat{\boldsymbol{R}}(\mathbf{w})\|_2 \leq B$ for any $\mathbf{w} \in \mathcal{W}$ and $\|W_{f_0}\|_2 \leq 1$.

Then, by the matrix Bernstein inequality in Tropp [76, Theorem 1.4], we have

$$
\|W_{f_0} - \widehat{W}_{f_0}\|_2 \leq \frac{2\log(2K/\delta)}{3|S_0|} + \sqrt{\frac{2\log(2K/\delta)}{|S_0|}} \tag{16}
$$

with probability at least $1 - \delta$.

Now, we turn to bound the term $\|\widehat{\boldsymbol{R}}_0(\mathbf{w}) - \boldsymbol{R}_0(\mathbf{w})\|_2$. Let $N_{\min} = \min\{|S_0^1|, \ldots, |S_0^K|\}$. According to Lemma 5, we can first show that $N_{\min} \geq (\kappa|S_0|)/2$ with probability at least $1 - \delta$ for any $\delta \in (0, 1/4]$. Then, we can bound the last term by

$$
\|\widehat{\boldsymbol{R}}_0(\mathbf{w}) - \boldsymbol{R}_0(\mathbf{w})\|_2 \leq \sqrt{K} \max_{k \in [K]}\{|R_0^k(\mathbf{w}) - \widehat{R}_0^k(\mathbf{w})|\}.
$$

According to the Hoeffding's lemma, for each $k \in [K]$, we can bound

$$
|R_0^k(\mathbf{w}) - \widehat{R}_0^k(\mathbf{w})| \leq \sqrt{\frac{B^2\log(1/\delta)}{2|S_0^k|}} \leq \sqrt{\frac{B^2\log(1/\delta)}{2N_{\min}}}
$$

with probability at least $1 - \delta$. Combining the $K$ events, which holds with probability $1 - \delta$, and the event that $N_{\min} \geq (\kappa|S_0|)/2$, we can obtain that

$$
\|\widehat{\boldsymbol{R}}_0(\mathbf{w}) - \boldsymbol{R}_0(\mathbf{w})\|_2 \leq \sqrt{\frac{KB^2\log(1/\delta)}{\kappa|S_0|}} \tag{17}
$$

with probability at least $1 - (K+1)\delta$.

Putting (16) and (17) together, we have

$$
\texttt{term (b)} \leq \frac{B}{\kappa\sigma}\left(\frac{2\log(2K/\delta)}{3|S_0|} + \sqrt{\frac{2\log(2K/\delta)}{|S_0|}} + \sqrt{\frac{K\log(1/\delta)}{\kappa|S_0|}}\right)
$$

with probability at least $1 - (K+2)\delta$. Combining the upper bounds for term (a) and term (b), we can complete the proof by

$$
|\mathbb{E}_{S_t \sim \mathcal{D}_t}[\widehat{R}_t(\mathbf{w})] - R_t(\mathbf{w})| \leq \frac{4\sqrt{K}B}{\kappa^2\sigma^2}\left(\frac{\log(2K/\delta)}{|S_0|} + \sqrt{\frac{2\log(2K/\delta)}{|S_0|}}\right)
$$

with probability at least $1 - (K+3)\delta$.

$\square$

### D.3 Static Regret of UOGD Algorithm

The following lemma presents the static regret guarantees of the UOGD algorithm when the step size is set as $\eta = \Theta(T^{-1/2})$.

**Lemma 3.** *Under the same assumptions of Lemma 1, UOGD in Eq. (4) with a step size $\eta = \Theta(T^{-1/2})$ satisfies*

$$
\mathbb{E}_{1:T}\left[\mathbf{Reg}_T^{\mathsf{s}}\right] \leq \frac{2G\Gamma}{\sigma}\sqrt{KT} = \mathcal{O}(\sqrt{T}),
$$

*where $\mathbb{E}_{1:T}[\cdot]$ denotes the expectation taken over the random draw of dataset $\{S_t\}_{t=1}^T$. The constant $\sigma > 0$ denotes the minimum singular value of confusion matrix $C_{f_0}$.*

## D.4 Useful Lemmas

This section presents several useful lemmas for the proofs in Appendix D.

**Lemma 4.** *Let $J_{f_0}$ and $\widehat{J}_{f_0}$ be the matrix defined as (15). Then, for any $\delta \in [0.1)$, with probability at least $1 - \delta$ we have*

$$\|\widehat{J}_{f_0}^{-1} - J_{f_0}^{-1}\|_2 \leq \frac{2\log(2K/\delta)}{\kappa^2\sigma^2|S_0|} + \frac{2}{\kappa^2\sigma^2}\sqrt{\frac{2\log(2K/\delta)}{|S_0|}},$$

*given that $|S_0| \geq (25\log(2K/\delta))/(\kappa\sigma)^2$, where $\kappa = \min_{k\in[K]}\mathcal{D}_0(y = k)$ and $\sigma$ is the minimum singular value of $C_{f_0}$.*

*Proof of Lemma 4.* Denoting by $\Delta_J = \widehat{J}_{f_0} - J_{f_0}$, according to Azizzadenesheli et al. [13, Lemma 2], we have

$$\|\Delta_J\|_2 \leq \frac{2\log(2K/\delta)}{3|S_0|} + \sqrt{\frac{2\log(2K/\delta)}{|S_0|}} \leq \frac{\kappa\sigma}{2}$$

with probability at least $1 - \delta$, where the last inequality is due to the assumption that $|S_0| \geq (\log(2K/\delta))/(\sigma\kappa)^2$. We can check the condition for Lemma 10

$$\|J_{f_0}^{-1}\Delta_J\|_2 \leq \|C_{f_0}^{-1}\|_2 \cdot \|W_{f_0}^{-1}\|_2 \cdot \|\Delta_J\|_2 \leq \frac{1}{\kappa\sigma} \cdot \frac{\kappa\sigma}{2},$$

where the last inequality holds since $\sigma$ is the minimum singular value of $C_{f_0}$ and $\kappa$ is that of $W_{f_0}$.

Then, according to Lemma 10, we can complete the proof by

$$\|\widehat{J}_{f_0}^{-1} - J_{f_0}^{-1}\|_2 \leq 2\|J_{f_0}^{-1}\|_2^2\|\Delta\|_2 \leq \frac{2\log(2K/\delta)}{\kappa^2\sigma^2|S_0|} + \frac{2}{\kappa^2\sigma^2}\sqrt{\frac{2\log(2K/\delta)}{|S_0|}}.$$

$\square$

**Lemma 5.** *Let $|S_0| \geq (16\log(1/\delta))/\kappa^2$. Then, for any $\delta \in (0, 1/4]$, with probability at least $1 - \delta$ the following holds:*

$$N_{\min} \geq \frac{\kappa|S_0|}{2},$$

*where $N_{\min} = \min\{|S_0^1|, \ldots, |S_0^K|\}$ and $\kappa = \min_{k\in[K]}\mathcal{D}_0(y = k)$.*

*Proof of Lemma 5.* For notation simplicity, we denote by $\widetilde{\mu}_0$ the empirical class prior at $t = 0$, i.e. $[\widetilde{\mu}_0]_k = |S_0^k|/|S_0|$ and $\mu_0$ the expected class prior with $[\mu_0]_k = \mathcal{D}_0(y = k)$. Then, according to Hsu et al. [77, Proposition 19], we have

$$\|\widetilde{\mu}_0 - \mu_0\|_2 \leq \frac{1}{\sqrt{|S_0|}} + \sqrt{\frac{\log(1/\delta)}{|S_0|}}$$

with probability at least $1 - \delta$, which implies that

$$[\widetilde{\mu}_0]_k \geq [\mu_0]_k - \frac{1}{\sqrt{|S_0|}} - \sqrt{\frac{\log(1/\delta)}{|S_0|}}, \quad \forall k \in [K]$$

holds with probability at least $1 - \delta$. We complete the proof by noticing that $[\mu_0]_k \geq \kappa$ holds for any $k \in [K]$ and further noting the assumption that $|S_0| \geq \frac{16\log(1/\delta)}{\kappa^2}$. $\square$

# E  Omitted Proofs for Section 3.2

This section provides the proof omitted in section 3.2. We will first present the regret bound for the UOGD algorithm without step size tuning (Theorem 1), and then show the overall dynamic regret bound of the ATLAS algorithm (Theorem 2). At the end of this section, we list several useful lemmas.

Before presenting the proofs, we highlight the main challenge of the regret analysis. As noted by Section 3.2 and Appendix C.2, our performance measure (1) is different from the conventional notion of (worst-case) dynamic regret [27], where the performance of the learned model is measured by the observed loss function. However, in the OLaS problem, the model is desired to perform well over the underlying distribution, and its quality is evaluated over the expected risk function $R_t(\cdot)$. The main challenge here is that the expected risk function $R_t(\cdot)$ is unavailable in the learning process, which makes it hard to apply the analysis of (worst-case) dynamic regret for our case.

## E.1  Proof of Theorem 1

This part presents the proof of Theorem 1. The key idea is to decompose the overall dynamic regret into two parts by introducing a reference sequence, which changes in a piecewise-stationary manner. In our analysis, the first part measures the regret of our algorithm when compared with the reference sequence, and the second part reflects the quality of the reference sequence. Since the reference sequence appears only in the analysis, we can make a tight regret bound by choosing a proper one to balance the first and second parts.

*Proof of Theorem 1.* We can decompose the regret bound into two parts by introducing a reference sequence $\{\mathbf{u}_t\}_{t=1}^T$ that only changes every $\Delta$ iteration. More specifically, denoting by $\mathcal{I}_m = [(m-1)\Delta + 1, m\Delta]$ the $m$-th time interval, any comparator $\mathbf{u}_t$ that falls into $\mathcal{I}_m$ is taken as the single best decision over the interval, i.e. $\mathbf{u}_t = \mathbf{w}_{\mathcal{I}_m}^* \in \arg\min_{\mathbf{w}\in\mathcal{W}} \sum_{t\in\mathcal{I}_m} R_t(\mathbf{w})$ for any $t \in \mathcal{I}_m$. We have

$$
\mathbb{E}_{1:T}\left[\sum_{t=1}^T R_t(\mathbf{w}_t)\right] - \sum_{t=1}^T R_t(\mathbf{w}_t^*)
$$

$$
= \mathbb{E}_{1:T}\left[\sum_{t=1}^T R_t(\mathbf{w}_t)\right] - \sum_{t=1}^T R_t(\mathbf{u}_t) + \sum_{t=1}^T R_t(\mathbf{u}_t) - \sum_{t=1}^T R_t(\mathbf{w}_t^*)
$$

$$
= \underbrace{\mathbb{E}_{1:T}\left[\sum_{t=1}^T R_t(\mathbf{w}_t)\right] - \sum_{m=1}^M \sum_{t\in\mathcal{I}_m} R_t(\mathbf{w}_{\mathcal{I}_m}^*)}_{\texttt{term (a)}} + \underbrace{\sum_{m=1}^M \sum_{t\in\mathcal{I}_m} R_t(\mathbf{w}_{\mathcal{I}_m}^*) - \sum_{t=1}^T R_t(\mathbf{w}_t^*)}_{\texttt{term (b)}},
$$

where $M = \lceil \frac{T}{\Delta} \rceil \leq T/\Delta + 1$ is the number of the intervals. Then, we turn to analyze term (a) and term (b), respectively.

**Analysis for term (a).**  Term (a) measures the regret of our algorithm when compared with a piecewise-stationary sequence. We first show that the regret defined over the expected risk $R_t(\cdot)$ can be related to the empirical risk estimator $\widehat{R}_t(\cdot)$ due to its unbiased property.

```
term (a)
```

$$
= \mathbb{E}_{1:T}\left[\sum_{t=1}^T R_t(\mathbf{w}_t)\right] - \sum_{t=1}^T R_t(\mathbf{u}_t)
$$

$$
\leq \mathbb{E}_{1:T}\left[\sum_{t=1}^T \langle \nabla R_t(\mathbf{w}_t), \mathbf{w}_t - \mathbf{u}_t \rangle\right]
$$

$$
= \underbrace{\mathbb{E}_{1:T}\left[\sum_{t=1}^T \langle \nabla R_t(\mathbf{w}_t) - \nabla \widehat{R}_t(\mathbf{w}_t), \mathbf{w}_t - \mathbf{u}_t \rangle\right]}_{\texttt{term (a-1)}} + \underbrace{\mathbb{E}_{1:T}\left[\sum_{t=1}^T [\langle \nabla \widehat{R}_t(\mathbf{w}_t), \mathbf{w}_t - \mathbf{u}_t \rangle]\right]}_{\texttt{term (a-2)}}, \quad (18)
$$

where the first inequality is due to the convexity of the risk function $R_t(\cdot)$. For term (a-1), we have

$$\texttt{term (a-1)} = \mathbb{E}_{1:T}\left[\langle \nabla R_t(\mathbf{w}_t) - \nabla \widehat{R}_t(\mathbf{w}_t), \mathbf{w}_t - \mathbf{u}_t\rangle\right]$$

$$= \mathbb{E}_{1:t-1}\left[\langle \nabla R_t(\mathbf{w}_t) - \mathbb{E}_t\left[\nabla \widehat{R}_t(\mathbf{w}_t)\,\middle|\, 1:t-1\right], \mathbf{w}_t - \mathbf{u}_t\rangle\right] = 0,$$

where the last equality is due to the unbiasedness of the risk estimator $\widehat{R}_t$ such that $\nabla R_t(\mathbf{w}_t) = \mathbb{E}_t\left[\nabla \widehat{R}_t(\mathbf{w}_t)\,\middle|\, 1:t-1\right]$. Thus, it is sufficient to analyze term (a-2) to provide an upper bound for term (a). For the model sequence $\{\mathbf{w}_t\}_{t=1}^T$ generated by the UOGD algorithm in Eq. (4), we have the following lemma, whose proof is deferred to Appendix E.4.

**Lemma 6.** *Under same assumptions of Theorem 1, UOGD in Eq. (4) with a step size $\eta > 0$ satisfies*

$$\sum_{t=1}^T \langle \nabla \widehat{R}_t(\mathbf{w}_t), \mathbf{w}_t - \mathbf{u}_t\rangle \leq \frac{2\eta K G^2 T}{\sigma^2} + \frac{2\Gamma P_T + \Gamma^2}{2\eta}$$

*for any comparator sequence $\{\mathbf{u}_t\}_{t=1}^T$ with $\mathbf{u}_t \in \mathcal{W}$. Moreover, $P_T = \sum_{t=2}^T \|\mathbf{u}_t - \mathbf{u}_{t-1}\|_2$ measures the variation of the comparator sequence.*

Since the comparator sequence in term (a) only changes $M - 1$ times, its variation is bounded by $P_T \leq \Gamma(M - 1) \leq (\Gamma T)/\Delta$. By Lemma 6 and taking an expectation over both sides, we have

$$\texttt{term (a-2)} \leq \frac{2\eta K G^2 T}{\sigma^2} + \frac{2\Gamma^2 T/\Delta + \Gamma^2}{2\eta}.$$

Combining the upper bounds of term(a-1) and term (a-2) yields

$$\texttt{term (a)} \leq \texttt{term (a-1)} + \texttt{term (a-2)} \leq \frac{2\eta K G^2 T}{\sigma^2} + \frac{2\Gamma^2 T/\Delta + \Gamma^2}{2\eta}. \tag{19}$$

**Analysis for term (b).** For term (b), we can follow the reasoning in Besbes et al. [17] and subsequently simplified analysis in Zhang et al. [78, Lemma 1] to show that

$$\texttt{term (b)} = \sum_{m=1}^M \sum_{t\in\mathcal{I}_m} \left(R_t(\mathbf{w}_{\mathcal{I}_m}^*) - R_t(\mathbf{w}_t^*)\right)$$

$$\leq \sum_{m=1}^M \sum_{t\in\mathcal{I}_m} \left(R_t(\mathbf{w}_{s_m}^*) - R_t(\mathbf{w}_t^*)\right)$$

$$= \sum_{m=1}^M \sum_{t\in\mathcal{I}_m} \left(R_t(\mathbf{w}_{s_m}^*) - R_{s_m}(\mathbf{w}_{s_m}^*) + R_{s_m}(\mathbf{w}_{s_m}^*) - R_t(\mathbf{w}_t^*)\right)$$

$$\leq \sum_{m=1}^M \sum_{t\in\mathcal{I}_m} \left(R_t(\mathbf{w}_{s_m}^*) - R_{s_m}(\mathbf{w}_{s_m}^*) + R_{s_m}(\mathbf{w}_t^*) - R_t(\mathbf{w}_t^*)\right)$$

$$\leq 2\Delta \sum_{m=1}^M \sum_{t\in\mathcal{I}_m} \sup_{\mathbf{w}\in\mathcal{W}} |R_t(\mathbf{w}) - R_{t-1}(\mathbf{w})|$$

$$= 2\Delta \sum_{t=2}^T \sup_{\mathbf{w}\in\mathcal{W}} |R_t(\mathbf{w}) - R_{t-1}(\mathbf{w})|$$

$$\triangleq 2B\Delta V_T^R, \tag{20}$$

where $s_m = (m-1)\Delta + 1$ is the first time step at interval $\mathcal{I}_m$. In the above, the first inequality is due to the optimality of $\mathbf{w}_{\mathcal{I}_m}^*$ over the interval $\mathcal{I}_m$. The second inequality holds since $\mathbf{w}_{s_m}^* \in \arg\min_{\mathbf{w}\in\mathcal{W}} R_{s_m}(\mathbf{w})$.

Combining term (a) and term (b), we have

$$\mathbb{E}_{1:T}\left[\sum_{t=1}^{T}R_t(\mathbf{w}_t)\right] - \sum_{t=1}^{T}R_t(\mathbf{w}_t^*) \leq \frac{2\eta KG^2 T}{\sigma^2} + \frac{2\Gamma^2 T/\Delta + \Gamma^2}{2\eta} + 2B\Delta V_T^R$$

$$\leq \frac{2\eta KG^2 T}{\sigma^2} + \frac{\Gamma^2}{2\eta} + 4\Gamma\sqrt{\frac{BTV_T^R}{\eta}} + 2BV_T^R$$

$$\leq \frac{2\eta KG^2 T}{\sigma^2} + \frac{\Gamma^2}{2\eta} + 4\Gamma\sqrt{\frac{BTV_T^R}{\eta}} + 4B\sqrt[3]{T^2 V_T^R}$$

$$\leq \left(\frac{2KG^2}{\sigma^2} + 2B^2\right)\eta T + \frac{\Gamma^2}{\eta} + 4(\Gamma+1)\sqrt{\frac{BTV_T^R}{\eta}}, \quad (21)$$

where the second inequality is due to the parameter setting $\Delta = \lceil\sqrt{\Gamma^2 T/(\eta BV_T^R)}\rceil$. The third inequality is due to the fact that $V_T^R \leq 2T$. The last inequality holds by the AM-GM inequality $3B\sqrt[3]{T^2 V_T^R} \leq 2\sqrt{\frac{BTV_T^R}{\eta}} + B^2\eta T$.

Inspired by Zhou et al. [79, Proposition 1], we know that the bounded distance between probability distributions can lead to the temporal variability condition. In the OLaS problem, with the label shift condition, we can further bound the variation of the loss function by the variation of the class prior $V_T = \sum_{t=2}^{T}\|\boldsymbol{\mu}_{y_t} - \boldsymbol{\mu}_{y_{t-1}}\|_1$:

$$V_T^R \triangleq \sum_{t=2}^{T}\sup_{\mathbf{w}\in\mathcal{W}}|R_t(\mathbf{w}) - R_{t-1}(\mathbf{w})|$$

$$= \sum_{t=2}^{T}\sup_{\mathbf{w}\in\mathcal{W}}\left|\sum_{k=1}^{K}([\boldsymbol{\mu}_{y_t}]_k - [\boldsymbol{\mu}_{y_{t-1}}]_k)R_0^k(\mathbf{w})\right|$$

$$\leq \sum_{t=2}^{T}B\sum_{k=1}^{K}\left|[\boldsymbol{\mu}_{y_t}]_k - [\boldsymbol{\mu}_{y_{t-1}}]_k\right|$$

$$= B\sum_{t=2}^{T}\|\boldsymbol{\mu}_{y_t} - \boldsymbol{\mu}_{y_{t-1}}\|_1 \quad (22)$$

$$\triangleq BV_T. \quad (23)$$

Plug (22) into (26) of Lemma 8, we have

$$\mathbb{E}\left[\mathbf{Reg}_T^{\mathbf{d}}\right] \leq \left(\frac{2KG^2}{\sigma^2} + 2B^2\right)\eta T + \frac{\Gamma^2}{\eta} + 4(\Gamma+1)\sqrt{\frac{BTV_T}{\eta}},$$

which completes the proof.

$\square$

### E.2 Proof of Theorem 2

*Proof of Theorem 2.* We can decompose the dynamic regret into two parts with the piecewise-stationary sequence $\{\mathbf{u}_t\}_{t=1}^{T}$ introduced in the proof of Theorem 1.

$$\mathbb{E}_{1:T}\left[\sum_{t=1}^{T}R_t(\mathbf{w}_t)\right] - \sum_{t=1}^{T}R_t(\mathbf{w}_t^*)$$

$$= \mathbb{E}_{1:T}\left[\sum_{t=1}^{T}R_t(\mathbf{w}_t)\right] - \sum_{t=1}^{T}R_t(\mathbf{u}_t) + \sum_{t=1}^{T}R_t(\mathbf{u}_t) - \sum_{t=1}^{T}R_t(\mathbf{w}_t^*)$$

$$= \mathbb{E}_{1:T}\underbrace{\left[\sum_{t=1}^{T}R_t(\mathbf{w}_t)\right] - \sum_{m=1}^{M}\sum_{t\in\mathcal{I}_m}R_t(\mathbf{w}_{\mathcal{I}_m}^*)}_{\texttt{term (a)}} + \underbrace{\sum_{m=1}^{M}\sum_{t\in\mathcal{I}_m}R_t(\mathbf{w}_{\mathcal{I}_m}^*) - \sum_{t=1}^{T}R_t(\mathbf{w}_t^*)}_{\texttt{term (b)}},$$

where term (b) can be bounded by

$$\texttt{term (b)} \leq 2B\Delta V_T$$

following the same arguments in deriving (20) and (22). So, we only need to focus on the analysis of term (a), the regret of the model sequence returned by ATLAS algorithm to the piecewise-stationary compactors.

For any $i$-th base UOGD algorithm, we can further decompose term (a) into the meta-regret and the base-regret

$$\mathbb{E}_{1:T}\left[\sum_{t=1}^{T}R_t(\mathbf{w}_t)\right] - \sum_{t=1}^{T}R_t(\mathbf{u}_t)$$

$$= \underbrace{\mathbb{E}_{1:T}\left[\sum_{t=1}^{T}R_t(\mathbf{w}_t) - \sum_{t=1}^{T}R_t(\mathbf{w}_{t,i})\right]}_{\texttt{meta-regret}} + \underbrace{\mathbb{E}_{1:T}\left[\sum_{t=1}^{T}R_t(\mathbf{w}_{t,i})\right] - \sum_{t=1}^{T}R_t(\mathbf{u}_t)}_{\texttt{base-regret}}, \qquad (24)$$

where the meta-regret measures the performance gap between the model returned by ATLAS algorithm and that by the $i$-th base UOGD algorithm, and the base-regret is the regret of the $i$-th base UOGD algorithm running with step size $\eta_i > 0$.

**Analysis for meta-regret.** For the meta-regret, we have

$$\mathbb{E}_{1:T}\left[\sum_{t=1}^{T}R_t(\mathbf{w}_t) - \sum_{t=1}^{T}R_t(\mathbf{w}_{t,i})\right]$$

$$\leq \mathbb{E}_{1:T}\left[\sum_{t=1}^{T}\sum_{j=1}^{N}p_{t,j}R_t(\mathbf{w}_{t,j}) - \sum_{t=1}^{T}R_t(\mathbf{w}_{t,i})\right]$$

$$= \mathbb{E}_{1:T}\left[\sum_{t=1}^{T}\sum_{j=1}^{N}p_{t,j}\widehat{R}_t(\mathbf{w}_{t,j}) - \sum_{t=1}^{T}\widehat{R}_t(\mathbf{w}_{t,i})\right]$$

$$+ \mathbb{E}_{1:T}\left[\sum_{t=1}^{T}\sum_{j=1}^{N}p_{t,j}(R_t(\mathbf{w}_{t,j}) - \widehat{R}_t(\mathbf{w}_{t,j})) + \sum_{t=1}^{T}(R_t(\mathbf{w}_{t,i}) - \widehat{R}_t(\mathbf{w}_{t,i}))\right]$$

$$= \mathbb{E}_{1:T}\left[\sum_{t=1}^{T}\sum_{j=1}^{N}p_{t,j}\widehat{R}_t(\mathbf{w}_{t,j}) - \sum_{t=1}^{T}\widehat{R}_t(\mathbf{w}_{t,i})\right], \qquad (25)$$

where the first inequality is due to the Jensen's inequality and the last equality is due to the unbiasedness of the risk estimator such that $\mathbb{E}_t[\widehat{R}_t(\mathbf{w}_{t,i})] = R_t(\mathbf{w}_{t,i})$ for any $i \in [N]$. Then, we can upper bound the meta-regret by the following lemma, whose proof is deferred to Appendix E.4.

**Lemma 7.** *By setting the learning rate* $\varepsilon = \frac{\sigma}{B}\sqrt{\frac{\ln N+2}{KT}}$, *the meta-algorithm of* ATLAS *(Algorithm 2) satisfies*

$$\sum_{t=1}^{T}\sum_{j=1}^{N}p_{t,j}\widehat{R}_t(\mathbf{w}_{t,j}) - \sum_{t=1}^{T}\widehat{R}_t(\mathbf{w}_{t,i}) \leq \frac{2B}{\sigma}\sqrt{(\ln N + 2)KT}$$

*for any* $i \in [N]$, *where* $B \triangleq \sup_{(\mathbf{x},y)\in\mathcal{X}\times\mathcal{Y},\mathbf{w}\in\mathcal{W}}|\ell(f(\mathbf{w},\mathbf{x}),y)|$ *is defined as the upper bound of the loss function.*

As a consequence of Lemma 7, we can upper bound the meta-regret as

$$\texttt{meta-regret} \leq \frac{2B}{\sigma}\sqrt{(\ln N + 2)KT}.$$

**Analysis for base-regret.** Since the base-algorithm of ATLAS algorithm (Algorithm 1) is taken as the UOGD algorithm, following a similar argument for obtaining Lemma 6, we have

$$\mathbb{E}_{1:T}\left[\sum_{t=1}^{T} R_t(\mathbf{w}_{t,i})\right] - \sum_{t=1}^{T} R_t(\mathbf{u}_t) \leq \frac{2\eta_i K G^2 T}{\sigma^2} + \frac{2\Gamma P_T + \Gamma^2}{2\eta_i}$$

for any base-algorithm with the index $i \in [N]$, where $P_T$ is the variation of the comparator sequence $P_T = \sum_{t=2}^{T}\|\mathbf{u}_t - \mathbf{u}_{t-1}\|_2$. Then, by combining the meta-regret and the base-regret, we can obtain that

$$\texttt{term (a)} = \mathbb{E}_{1:T}\left[\sum_{t=1}^{T}(\mathbf{w}_t)\right] - \sum_{t=1}^{T} R_t(\mathbf{u}_t)$$

$$\leq \frac{2B}{\sigma}\sqrt{(\ln N + 2)KT} + \frac{2\eta_i K G^2 T}{\sigma^2} + \frac{2\Gamma P_T + \Gamma^2}{2\eta_i}$$

for any base-algorithm $i \in [N]$.

It remains to identify the optimal step size $\eta_{i_*}$ to make the bound tight. Due to the construction of the step size pool

$$\mathcal{H} = \left\{\frac{\Gamma\sigma}{2G\sqrt{KT}} \cdot 2^{i-1} \mid i \in [N]\right\}$$

with $N = 1 + \lceil\frac{1}{2}\log_2(1 + 2T)\rceil$, we can make sure that the optimal step size $\eta^* = \frac{\sigma}{2G}\sqrt{\frac{\Gamma^2 + 2\Gamma P_T}{KT}}$ is covered by the step size pool. Even better, due to the logarithmic construction of the pool, we claim that there must exist an $i_* \in [N]$ satisfies that $\eta_{i_*}/2 \leq \eta^* \leq \eta_{i_*}$. So, we further bound term (a) as

$$\texttt{term (a)} \leq \frac{2B}{\sigma}\sqrt{(\ln N + 2)KT} + \frac{3G}{\sigma}\sqrt{KT(2\Gamma P_T + \Gamma^2)}$$

by comparing with the $i_*$-th base-algorithm in the decomposition (24).

**Overall dynamic regret bound.** Combining term (a) and term (b) and noticing that $\{\mathbf{u}_t\}_{t=1}^{T}$ is the piecewise stationary comparator sequences with $P_T = \sum_{t=2}^{T}\|\mathbf{u}_t - \mathbf{u}_{t-1}\|_{t=1}^{T} \leq \Gamma T/\Delta$, we can bound the overall dynamic regret as

$$\mathbf{Reg}_T^{\mathbf{d}} = \mathbb{E}_{1:T}\left[\sum_{t=1}^{T} R_t(\mathbf{w}_t)\right] - \sum_{t=1}^{T} R_t(\mathbf{w}_t^*)$$

$$\leq 2B\Delta V_T + \frac{3G\Gamma}{\sigma}\sqrt{KT\left(1 + \frac{2T}{\Delta}\right)} + \frac{2B}{\sigma}\sqrt{(\ln N + 2)KT}$$

$$\leq 2B\Delta V_T + \frac{3G\Gamma T}{\sigma}\sqrt{\frac{2K}{\Delta}} + \frac{3G\Gamma}{\sigma}\sqrt{KT} + \frac{2B}{\sigma}\sqrt{(\ln N + 2)KT},$$

where the last inequality is due to the fact that $\sqrt{a + b} \leq \sqrt{a} + \sqrt{b}$ for any $a, b \geq 0$. We can complete the proof by choosing $\Delta = \lceil(\frac{3G\Gamma T\sqrt{2K}}{4\sigma B V_T})^{\frac{2}{3}}\rceil$ and obtain the bound as

$$\mathbf{Reg}_T^{\mathbf{d}} \leq 3\left(\frac{9KBG^2\Gamma^2}{\sigma^2}\right)^{\frac{1}{3}} \cdot V_T^{1/3}T^{2/3} + \frac{3\sqrt{K}(G\sqrt{\Gamma} + B\sqrt{\ln N + 2})}{\sigma}\sqrt{T} + 2BV_T$$

$$\leq 3\left(\frac{9KBG^2\Gamma^2}{\sigma^2}\right)^{\frac{1}{3}} \cdot V_T^{1/3}T^{2/3} + \frac{3\sqrt{K}(G\sqrt{\Gamma} + B\sqrt{\ln N + 2})}{\sigma}\sqrt{T} + 4BV_T^{1/3}T^{2/3},$$

where the last inequality is due to the fact that $V_T \leq 2T$. $\qquad\square$

### E.3 Discussion on Minimax Optimality

For online convex optimization with a general convex function $R_t : \mathcal{W} \to \mathbb{R}$, Besbes et al. [17, Theorem 2] established an $\Omega\left((V_T^R)^{1/3}T^{2/3}\right)$ lower bound for the dynamic regret $\sum_{t=1}^{T}\mathbb{E}[R_t(\mathbf{w}_t)] -$

$\sum_{t=1}^{T} \inf_{\mathbf{w} \in \mathcal{W}} R_t(\mathbf{w})$ when the learner can only observe a noisy estimate of the full information function $R_t$. In the above, the quantity $V_T^R = \sum_{t=2}^{T} \sup_{\mathbf{w} \in \mathcal{W}} |R_t(\mathbf{w}) - R_{t-1}(\mathbf{w})|$ measures the non-stationarity of the environments by the variation of the loss functions. In addition to the $\mathcal{O}(V_T^{1/3} T^{2/3})$ upper bound stated in Theorem 1, we can show that the UOGD algorithm can actually achieve an $\mathcal{O}\big((V_T^R)^{1/3} T^{2/3}\big)$ dynamic regret bound based on the following lemmas.

**Lemma 8.** *Under the same assumptions as Lemma 1, UOGD in Eq. (4) with step size $\eta$ satisfies*

$$\mathbb{E}\big[\mathbf{Reg}_T^{\mathbf{d}}\big] \leq 2\big(\frac{KG^2}{\sigma^2} + B^2\big)\eta T + \frac{\Gamma^2}{\eta} + 4(\Gamma + 1)\sqrt{\frac{BV_T^R T}{\eta}}$$

$$= \mathcal{O}\Big(\eta T + 1/\eta + \sqrt{(V_T^R T)/\eta}\Big), \tag{26}$$

*where the constant $\sigma > 0$ denotes the minimum singular value of the invertible confusion matrix $C_{f_0}$. Moreover, $V_T^R \triangleq \sum_{t=2}^{T} \sup_{\mathbf{w} \in \mathcal{W}} |R_t(\mathbf{w}) - R_{t-1}(\mathbf{w})|$ measures the variation of the loss function, usually known as the temporal variability or function variation in the online learning literature.*

*Proof of Lemma 8.* This lemma is actually an intermediate result (21) in the proof of Theorem 1. □

By setting the step size in (26) optimally as $\eta = \Theta\big(T^{-\frac{1}{3}}(V_T^R)^{\frac{1}{3}}\big)$, UOGD enjoys an $\mathcal{O}\big((V_T^R)^{\frac{1}{3}} T^{\frac{2}{3}}\big)$ dynamic regret. Such a rate matches the $\Omega\big((V_T^R)^{1/3} T^{2/3}\big)$ lower bound [17] and thus demonstrates the minimax optimality of the UOGD algorithm in the non-degenerated cases with the noisy feedback function. Moreover, we can use a similar argument to show that our ATLAS also enjoys an $\mathcal{O}(\max\{(V_T^R)^{\frac{1}{3}} T^{\frac{2}{3}}, \sqrt{T}\})$ dynamic regret guarantee in addition to the $\mathcal{O}(\max\{V_T^{\frac{1}{3}} T^{\frac{2}{3}}, \sqrt{T}\})$ bound exhibited in Theorem 2. Finally, we mention that although our algorithms are minimax optimal in terms of $T$ and $V_T^R$, while it remains unclear about the tightness of the dynamic regret bounds scaling with the variation of class priors $V_T$, which is left as future work to investigate.

We also make another remark on the situation when the online function observed by the learner is noiseless, an even better dynamic regret upper bound of order $\mathcal{O}(V_T)$ is achievable in the OCO setting [80], see more discussions on the worst-case dynamic regret in Appendix C.2. Note that such a rate is also attainable for our problem under the assumption that the distribution $\mathcal{D}_t(\mathbf{x})$ is *available* at every iteration, however, this assumption can hardly be true in practice since the data samples at the online adaptation stage are usually very few. Thus, we focus on the non-degenerated case where the noise exists due to the sampling from the unknown data distribution.

### E.4 Useful Lemmas in the Analysis of Theorem 1 and Theorem 2

This section presents several useful lemmas that are omitted in the proof of Theorem 1 and Theorem 2.

**Lemma 6.** *Under same assumptions of Theorem 1, UOGD in Eq. (4) with a step size $\eta > 0$ satisfies*

$$\sum_{t=1}^{T} \langle \nabla \widehat{R}_t(\mathbf{w}_t), \mathbf{w}_t - \mathbf{u}_t \rangle \leq \frac{2\eta KG^2 T}{\sigma^2} + \frac{2\Gamma P_T + \Gamma^2}{2\eta}$$

*for any comparator sequence $\{\mathbf{u}_t\}_{t=1}^{T}$ with $\mathbf{u}_t \in \mathcal{W}$. Moreover, $P_T = \sum_{t=2}^{T} \|\mathbf{u}_t - \mathbf{u}_{t-1}\|_2$ measures the variation of the comparator sequence.*

*Proof of Lemma 6.* We can decompose the L.H.S. of the inequality in Lemma 6 as

$$\langle \nabla \widehat{R}_t(\mathbf{w}_t), \mathbf{w}_t - \mathbf{u}_t \rangle = \underbrace{\langle \nabla \widehat{R}_t(\mathbf{w}_t), \mathbf{w}_t - \mathbf{w}_{t+1} \rangle}_{\texttt{term (a)}} + \underbrace{\langle \nabla \widehat{R}_t(\mathbf{w}_t), \mathbf{w}_{t+1} - \mathbf{u}_t \rangle}_{\texttt{term (b)}}.$$

For term (a), we have

$$\texttt{term (a)} = \langle \nabla \widehat{R}_t(\mathbf{w}_t), \mathbf{w}_t - \mathbf{w}_{t+1} \rangle$$

$$\leq \|\nabla \widehat{R}_t(\mathbf{w}_t)\|_2 \|\mathbf{w}_t - \mathbf{w}_{t+1}\|_2$$

$$\leq 2\eta \|\nabla \widehat{R}_t(\mathbf{w}_t)\|_2^2 + \frac{1}{2\eta} \|\mathbf{w}_t - \mathbf{w}_{t+1}\|_2^2,$$

where the first inequality is due to the Cauchy-Schwarz inequality and the second inequality comes from the AM-GM inequality. Then, we bound term (b) by Zhao et al. [19, Lemma 7],

$$\texttt{term (b)} \leq \frac{1}{2\eta} \left( \|\mathbf{w}_t - \mathbf{u}_t\|_2^2 - \|\mathbf{w}_{t+1} - \mathbf{u}_t\|_2^2 - \|\mathbf{w}_t - \mathbf{w}_{t+1}\|_2^2 \right).$$

Combining the two terms and taking the summation over $T$ iterations, we have

$$
\begin{aligned}
&\sum_{t=1}^{T} \langle \nabla \widehat{R}_t(\mathbf{w}_t), \mathbf{w}_t - \mathbf{u}_t \rangle \\
&\leq 2\eta \sum_{t=1}^{T} \|\nabla \widehat{R}_t(\mathbf{w}_t)\|_2^2 + \frac{1}{2\eta} \sum_{t=1}^{T} \left( \|\mathbf{w}_t - \mathbf{u}_t\|_2^2 - \|\mathbf{w}_{t+1} - \mathbf{u}_t\|_2^2 \right) \\
&\leq 2\eta \sum_{t=1}^{T} \|\nabla \widehat{R}_t(\mathbf{w}_t)\|_2^2 + \frac{1}{2\eta} \sum_{t=2}^{T} \left( \|\mathbf{w}_t - \mathbf{u}_t\|_2^2 - \|\mathbf{w}_t - \mathbf{u}_{t-1}\|_2^2 \right) + \frac{\Gamma^2}{2\eta} \\
&\leq 2\eta \sum_{t=1}^{T} \|\nabla \widehat{R}_t(\mathbf{w}_t)\|_2^2 + \frac{1}{2\eta} \sum_{t=2}^{T} (\|\mathbf{w}_t - \mathbf{u}_t\|_2 + \|\mathbf{w}_t - \mathbf{u}_{t-1}\|_2) \cdot \|\mathbf{u}_t - \mathbf{u}_{t-1}\|_2 + \frac{\Gamma^2}{2\eta} \\
&\leq 2\eta \sum_{t=1}^{T} \|\nabla \widehat{R}_t(\mathbf{w}_t)\|_2^2 + \frac{\Gamma}{\eta} \sum_{t=2}^{T} \|\mathbf{u}_t - \mathbf{u}_{t-1}\|_2 + \frac{\Gamma^2}{2\eta} \\
&= 2\eta \sum_{t=1}^{T} \|\nabla \widehat{R}_t(\mathbf{w}_t)\|_2^2 + \frac{2\Gamma P_T + \Gamma^2}{2\eta},
\end{aligned}
\tag{27}
$$

where we denote by $P_t = \sum_{t=2}^{T} \|\mathbf{u}_t - \mathbf{u}_{t-1}\|_2$ the variation of the comparator sequence $\{\mathbf{u}_t\}_{t=1}^{T}$. The third inequality is due to the fact $\|a\|_2^2 - \|b\|_2^2 \leq (\|a\|_2 + \|b\|_2) \cdot \|a - b\|_2$ for any $a, b \in \mathbb{R}^d$.

We can further bound the first term of (27) by

$$
\begin{aligned}
2\eta \sum_{t=1}^{T} \|\nabla \widehat{R}_t(\mathbf{w}_t)\|_2^2 &\overset{(3)}{=} 2\eta \sum_{t=1}^{T} \left\| \sum_{k=1}^{K} [C_{f_0}^{-1} \widehat{\boldsymbol{\mu}}_{\widehat{y}_t}]_k \cdot \nabla R_0^k(\mathbf{w}_t) \right\|_2^2 \\
&\leq 2\eta \sum_{t=1}^{T} \left( \sum_{k=1}^{K} \left| [C_{f_0}^{-1} \widehat{\boldsymbol{\mu}}_{\widehat{y}_t}]_k \right| \|\nabla R_0^k(\mathbf{w}_t)\|_2 \right)^2 \\
&\leq 2\eta G^2 \sum_{t=1}^{T} \left\| C_{f_0}^{-1} \widehat{\boldsymbol{\mu}}_{\widehat{y}_t} \right\|_1^2 \\
&\leq 2\eta K G^2 \sum_{t=1}^{T} \left\| C_{f_0}^{-1} \widehat{\boldsymbol{\mu}}_{\widehat{y}_t} \right\|_2^2 \\
&\leq 2\eta K G^2 \sum_{t=1}^{T} \left\| C_{f_0}^{-1} \right\|_2^2 \cdot \|\widehat{\boldsymbol{\mu}}_{\widehat{y}_t}\|_2^2 \\
&\leq 2\eta K G^2 \sum_{t=1}^{T} \|C_{f_0}^{-1}\|_2^2 \leq \frac{2\eta K G^2 T}{\sigma^2},
\end{aligned}
\tag{28}
$$

where $\sigma$ is the minimum singular value of the matrix $C_{f_0}$. In above, the second inequality is due to the definition $R_0^k(\mathbf{w}) = \mathbb{E}_{\mathbf{x} \sim \mathcal{D}_0(\mathbf{x} \mid y=k)}[\ell(f(\mathbf{w}, \mathbf{x}), k)]$ and the norm of gradient of the loss function $\|\nabla_{\mathbf{w}} \ell(f(\mathbf{w}, \mathbf{x}), y)\|_2$ is bounded by $G$ for any $\mathbf{x} \in \mathcal{X}$ and $y \in [K]$. The third inequality is a consequence of the fact that $\|\mathbf{x}\|_1 \leq \sqrt{\mathbf{x}_2}$ for any $\mathbf{x} \in \mathbb{R}^K$. The last second inequality hold since $\widehat{\boldsymbol{\mu}}_{\widehat{y}_t} \in \Delta_{K-1}$ and then we have $\|\widehat{\boldsymbol{\mu}}_{\widehat{y}_t}\|_2^2 \leq 1$. We complete the proof by plugging (28) into (27). $\square$

Then, we provide the proof of Lemma 7, which provide an upper bound for the meta-algorithm of the ATLAS algorithm.

**Lemma 7.** *By setting the learning rate* $\varepsilon = \frac{\sigma}{B}\sqrt{\frac{\ln N + 2}{KT}}$, *the meta-algorithm of* ATLAS *(Algorithm 2) satisfies*

$$\sum_{t=1}^{T}\sum_{j=1}^{N} p_{t,j}\widehat{R}_t(\mathbf{w}_{t,j}) - \sum_{t=1}^{T}\widehat{R}_t(\mathbf{w}_{t,i}) \leq \frac{2B}{\sigma}\sqrt{(\ln N + 2)KT}$$

*for any* $i \in [N]$, *where* $B \triangleq \sup_{(\mathbf{x},y)\in\mathcal{X}\times\mathcal{Y},\mathbf{w}\in\mathcal{W}} |\ell(f(\mathbf{w},\mathbf{x}),y)|$ *is defined as the upper bound of the loss function.*

*Proof.* Since ATLAS takes the Hedge algorithm as the meta-algorithm, we can exploit the standard analysis to upper bound the meta-regret. According to the regret guarantee of the Hedge algorithm (Lemma 11 in Appendix G) and setting $\mathbf{m}_t = 0$, we can bound the meta-regret bound by

$$\sum_{t=1}^{T}\sum_{j=1}^{N} p_{t,j}\widehat{R}_t(\mathbf{w}_{t,j}) - \sum_{t=1}^{T}\widehat{R}_t(\mathbf{w}_{t,i}) \leq \frac{\ln N + 2}{\varepsilon} + \varepsilon\sum_{t=1}^{T}|\max_{i\in[N]}\{\widehat{R}(\mathbf{w}_{t,i})\}|^2, \qquad (29)$$

where the last term can be further bounded by

$$|\widehat{R}_t(\mathbf{w}_{t,i})| = \left|\sum_{k=1}^{K}[C_{f_0}^{-1}\widehat{\boldsymbol{\mu}}_{\widehat{y}_t}]_k \cdot R_0^k(\mathbf{w}_{t,i})\right|$$

$$\leq B\|C_{f_0}^{-1}\widehat{\boldsymbol{\mu}}_{\widehat{y}_t}\|_1 \leq B\sqrt{K}\|C_{f_0}^{-1}\widehat{\boldsymbol{\mu}}_{\widehat{y}_t}\|_2$$

$$\leq B\sqrt{K}\|C_{f_0}^{-1}\|_2 \cdot \|\widehat{\boldsymbol{\mu}}_{\widehat{y}_t}\|_2 \leq \frac{B\sqrt{K}}{\sigma}, \qquad (30)$$

where the third inequality is due to the Cauchy–Schwarz inequality. The last inequality comes from the fact that $\|C_{f_0}^{-1}\|_2 \leq \sigma^{-1}$ and $\|\widehat{\boldsymbol{\mu}}_{\widehat{y}_t}\|_2 \leq 1$. Plugging (30) into (29) yields

$$\sum_{t=1}^{T}\sum_{j=1}^{N} p_{t,j}\widehat{R}_t(\mathbf{w}_{t,j}) - \sum_{t=1}^{T}\widehat{R}_t(\mathbf{w}_{t,i}) \leq \frac{\ln N + 2}{\varepsilon} + \varepsilon\sum_{t=1}^{T}\frac{B^2 KT}{\sigma^2}.$$

We complete the proof by setting $\varepsilon = \frac{\sigma}{B}\sqrt{\frac{(\ln N + 2)}{KT}}$. $\qquad\square$

## F   Omitted Details for Section 3.3

In this section, we first introduce the omitted algorithm details for the ATLAS-ADA algorithm in Appendix F.1. Then, we present the proof of the Theorem 3 (Appendix F.2), followed by the useful lemmas in the regret analysis.

### F.1   Algorithm Details for ATLAS-ADA

This part provides algorithm details of the ATLAS-ADA. The main procedures are presented in Algorithm 4 (base-algorithm) and Algorithm 5 (meta-algorithm).

**Hint Functions.**   As shown in Section 3.3, a guidance for designing the hint function is to approximate the true priors $\boldsymbol{\mu}_{y_t}$ with hint priors $\boldsymbol{h}_{y_t}$. A direct way is to use current data $S_t$ to estimate the hint priors, which we call *Forward Hint*. However, when the sample size $|S_t|$ is small at each round, the estimated priors can be far from the true priors, which motivates us to reuse the historical information. The intuition is that we can benefit from the historical data if there are some regular shift patterns. To see this, we design three additional hint functions. Details are listed in the following.

- **Forward Hint**. We perform transductive learning with the current data $S_t$ by $\boldsymbol{h}_{y_t} = \widehat{\boldsymbol{\mu}}_{y_t}$.
- **Window Hint**. With the belief that recent data are useful, we set a sliding window size of $L_m$ and use the average of the estimated priors in the window as the hint priors:

$$\boldsymbol{h}_{y_t} = \frac{1}{L_m}\sum_{s=t-L_m}^{t-1}\widehat{\boldsymbol{\mu}}_{y_s}.$$

| **Algorithm 4** ATLAS-ADA: base-algorithm | **Algorithm 5** ATLAS-ADA: meta-algorithm |
|---|---|
| **Input:** step size $\eta_i \in \mathcal{H}$ | **Input:** step size pool $\mathcal{H}$; learning rate $\varepsilon$ |
| 1: Let $\mathbf{w}_{1,i}$ be any point in $\mathcal{W}$ | 1: initialization: $\forall i \in [N], p_{1,i} = 1/N$ |
| 2: **for** $t = 2$ **to** $T$ **do** | 2: **for** $t = 2$ **to** $T$ **do** |
| 3:    construct the risk estimator $\widehat{R}_{t-1}$ as (3) | 3:    receive $\{\mathbf{w}_{t,i}\}_{i=1}^N$ from base-learners |
| 4:    update the model of base-learner by | 4:    update weight $\mathbf{p}_t \in \Delta_N$ according to |

$$\widehat{\mathbf{w}}_{t,i} = \Pi_{\mathcal{W}}[\widehat{\mathbf{w}}_{t-1,i} - \eta_i \nabla \widehat{R}_{t-1}(\mathbf{w}_{t-1,i})]$$
$$\mathbf{w}_{t,i} = \arg\min_{\mathbf{w} \in \mathcal{W}} H_t(\mathbf{w}) + \frac{1}{2\eta_i}\|\mathbf{w} - \widehat{\mathbf{w}}_{t,i}\|_2^2.$$

$$p_{t,i} \propto \exp\left(-\varepsilon\left(\sum_{s=1}^{t-1}\widehat{R}_s(\mathbf{w}_{s,i}) + H_t(\mathbf{w}_{t,i})\right)\right)$$

| 5:    send $\mathbf{w}_{t,i}$ to the meta-algorithm | 5:    predict final output $\mathbf{w}_t = \sum_{i=1}^N p_{t,i}\mathbf{w}_{t,i}$ |
| 6: **end for** | 6: **end for** |

- **Periodic Hint**. For a periodic situation where a similar pattern recurs at a specific time interval, we consider reusing the historical data in a periodic manner by maintaining a buffer $\{\bar{\boldsymbol{\mu}}_i \mid i \in [L_p]\}$ with the size of $L_p$. At round $t$, we utilize the estimated prior $\widehat{\boldsymbol{\mu}}_{y_t}$ to update the priors in the buffer:

$$\bar{\boldsymbol{\mu}}_i^{(t)} = \begin{cases} \bar{\boldsymbol{\mu}}_i^{(t-1)} + \lambda_t\left(\widehat{\boldsymbol{\mu}}_{y_t} - \bar{\boldsymbol{\mu}}_i^{(t-1)}\right) & i = t \bmod L_p \\ \bar{\boldsymbol{\mu}}_i^{(t-1)} & \text{otherwise} \end{cases},$$

  where $i \in [L_p]$ and $\bar{\boldsymbol{\mu}}_i^{(t)}$ denotes the value of $\bar{\boldsymbol{\mu}}_i$ at round $t$. $\lambda_t \in [0,1]$ is a coefficient to balance the historical and present information, which can be set to be $1/t$ or a constant value. Then we set $\boldsymbol{h}_{y_t} = \bar{\boldsymbol{\mu}}_{i_t}^{(t)}$, where $i_t = t \bmod L_p$.

- **Online KMeans Hint**. To further use the feature similarity of the historical data, we design a hint function with the online k-means clustering algorithms [81]. We maintain $L_k$ prototypes $\{(\bar{\mathbf{x}}_i, \bar{\boldsymbol{\mu}}_i) \mid i \in [L_k]\}$. At round $t$, we find the prototype $i_t$, which is the most similar to the current unlabeled data $S_t$:

$$i_t = \arg\min_{i \in [L_k]} \frac{1}{|S_t|} \sum_{\mathbf{x} \in S_t} \|\bar{\mathbf{x}}_i - \mathbf{x}\|^2,$$

  and then we update the corresponding prototype by

$$\bar{\mathbf{x}}_{i_t}^{(t)} = \bar{\mathbf{x}}_{i_t}^{(t-1)} + \kappa_t\left[\left(\frac{1}{|S_t|}\sum_{\mathbf{x} \in S_t}\mathbf{x}\right) - \bar{\mathbf{x}}_{i_t}^{(t-1)}\right],$$

$$\bar{\boldsymbol{\mu}}_{i_t}^{(t)} = \bar{\boldsymbol{\mu}}_{i_t}^{(t-1)} + \kappa_t\left(\widehat{\boldsymbol{\mu}}_{y_t} - \bar{\boldsymbol{\mu}}_{i_t}^{(t-1)}\right),$$

  where $\left(\bar{\mathbf{x}}_{i_t}^{(t)}, \bar{\boldsymbol{\mu}}_{i_t}^{(t)}\right)$ represents the value of $i_t$-th prototype at round $t$. $\kappa_t \in [0,1]$ is a coefficient to balance the historical and present information, which can be set to be $1/t$ or a constant value. Then we set $\boldsymbol{h}_{y_t} = \bar{\boldsymbol{\mu}}_{i_t}^{(t)}$.

## F.2   Proof of Theorem 3

The proof of Theorem 3 shares a similar spirit with that of Theorem 2, where we decompose the overall dynamic regret into meta-regret and base-regret. The new challenge is that the ATLAS-ADA algorithm involves the hint function $H_t(\cdot)$ to exploit the structure of distribution change, which requires a more delicate analysis for both the meta- and base-regret.

*Proof of Theorem 3.* Similar to the proof of Theorem 1 and Theorem 2, we can decompose the dynamic regret into two parts with the piecewise-stationary sequence $\{\mathbf{u}_t\}_{t=1}^T$,

$$\mathbb{E}_{1:T}\left[\sum_{t=1}^T R_t(\mathbf{w}_t)\right] - \sum_{t=1}^T R_t(\mathbf{w}_t^*)$$

$$= \underbrace{\mathbb{E}_{1:T}\left[\sum_{t=1}^T R_t(\mathbf{w}_t)\right] - \sum_{m=1}^M \sum_{t\in\mathcal{I}_m} R_t(\mathbf{w}_{\mathcal{I}_m}^*)}_{\texttt{term (a)}} + \underbrace{\sum_{m=1}^M \sum_{t\in\mathcal{I}_m} R_t(\mathbf{w}_{\mathcal{I}_m}^*) - \sum_{t=1}^T R_t(\mathbf{w}_t^*)}_{\texttt{term (b)}}.$$

According to (20) and (22), we have `term (b)` $\leq 2B\Delta V_T$. Besides, when the parameter is set as $\Delta = 1$, we have `term (b)` $= 0$. Thus, the term (b) can be bounded as

$$\texttt{term (b)} \leq \mathbb{1}\{\Delta > 1\} \cdot 2B\Delta V_T. \tag{31}$$

As for term (a), it can be further decomposed into the meta-regret and base-regret.

$$\texttt{term (a)} = \mathbb{E}_{1:T}\left[\sum_{t=1}^T R_t(\mathbf{w}_t)\right] - \sum_{t=1}^T R_t(\mathbf{u}_t)$$

$$= \underbrace{\mathbb{E}_{1:T}\left[\sum_{t=1}^T R_t(\mathbf{w}_t) - \sum_{t=1}^T R_t(\mathbf{w}_{t,i})\right]}_{\texttt{meta-regret}} + \underbrace{\mathbb{E}_{1:T}\left[\sum_{t=1}^T R_t(\mathbf{w}_{t,i})\right] - \sum_{t=1}^T R_t(\mathbf{u}_t)}_{\texttt{base-regret}},$$

where meta-regret measures the performance gap between the model returned by ATLAS-ADA and that by the $i$-th base algorithm. Base-regret is regret of the $i$-th base algorithm with step size $\eta_i > 0$.

**Analysis for the meta-regret.** Since $\mathbf{w}_t$ is independent of the sample observed at iteration $t$, following the same arguments in obtaining (25), we can bound the meta-regret as

$$\mathbb{E}_{1:T}\left[\sum_{t=1}^T R_t(\mathbf{w}_t) - \sum_{t=1}^T R_t(\mathbf{w}_{t,i})\right] \leq \mathbb{E}_{1:T}\left[\sum_{t=1}^T \sum_{j=1}^N p_{t,j}\widehat{R}_t(\mathbf{w}_{t,j}) - \sum_{t=1}^T \widehat{R}_t(\mathbf{w}_{t,i})\right].$$

**Remark 5.** The main difference between the analysis of the meta-algorithm for ATLAS-ADA and that of ATLAS is that the former method involves an update step with the hint function. To obtain a meta-regret bound that adapts the quality of the hint function, the previous work [19] crucially relies on the *smoothness* of the loss function, which is generally hard to be satisfied in the OLaS problem (e.g., the commonly used hinge loss is not smooth). A more delicate analysis shows that the smoothness assumption is no longer necessary to achieve an adaptive meta-regret bound. The result is stated as the following lemma, whose proof is deferred to Appendix F.3.

**Lemma 8.** *By setting the learning rate* $\varepsilon = \sqrt{\frac{\ln N + 2}{1 + \sum_{t=1}^T (\max_{i\in[N]}\{|\widetilde{R}_t(\mathbf{w}_{t,i}) - \widetilde{H}_t(\mathbf{w}_{t,i})|\})^2}}$, *the meta-algorithm of* ATLAS-ADA *(Algorithm 5) satisfies*

$$\sum_{t=1}^T \sum_{j=1}^N p_{t,j}\widehat{R}_t(\mathbf{w}_{t,j}) - \sum_{t=1}^T \widehat{R}_t(\mathbf{w}_{t,i}) \leq \Gamma\sqrt{(\ln N + 2)\left(1 + \sum_{t=1}^T \sup_{\mathbf{w}\in\mathcal{W}}\|\nabla\widehat{R}_t(\mathbf{w}) - \nabla H_t(\mathbf{w})\|_2^2\right)}$$

*for any* $i \in [N]$, *where* $\widetilde{R}_t(\mathbf{w}_{t,i}) = \widehat{R}_t(\mathbf{w}_{t,i}) - \widehat{R}_t(\mathbf{w}_{\mathbf{ref}})$ *and* $\widetilde{H}_t(\mathbf{w}_{t,i}) = H_t(\mathbf{w}_{t,i}) - H_t(\mathbf{w}_{\mathbf{ref}})$ *are the reference loss functions built upon an arbitrary reference point* $\mathbf{w}_{\mathbf{ref}} \in \mathcal{W}$.

According to Lemma 8, we can bound the meta-regret by

$$\mathbb{E}_{1:T}\left[\sum_{t=1}^T R_t(\mathbf{w}_t) - \sum_{t=1}^T R_t(\mathbf{w}_{t,i})\right]$$

$$\leq \Gamma\mathbb{E}_{1:T}\left[\sqrt{(\ln N + 2)\sum_{t=1}^T \sup_{\mathbf{w}\in\mathcal{W}}\|\nabla\widehat{R}_t(\mathbf{w}) - \nabla H_t(\mathbf{w})\|_2^2}\right]$$

$$\leq \Gamma \sqrt{(\ln N + 2) \left(1 + \sum_{t=1}^{T} \mathbb{E}_t \left[\sup_{\mathbf{w} \in \mathcal{W}} \|\nabla \widehat{R}_t(\mathbf{w}) - \nabla H_t(\mathbf{w})\|_2^2\right]\right)},$$

where the last inequality is due to the Jensen's inequality and the concavity of the square root function.

**Analysis for the base-regret.** Since the model $\mathbf{w}_t$ is independent of the draw of dataset $S_t$ received at iteration $t$, following a similar argument in obtain (19), we can bound the base-regret as

$$\mathbb{E}_{1:T} \left[\sum_{t=1}^{T} R_t(\mathbf{w}_{t,i})\right] - \sum_{t=1}^{T} R_t(\mathbf{u}_t) \leq \mathbb{E}_{1:T} \left[\sum_{t=1}^{T} [\langle \nabla \widehat{R}_t(\mathbf{w}_{t,i}), \mathbf{w}_{t,i} - \mathbf{u}_t \rangle\right].$$

Different from the analysis for the base-algorithm of ATLAS, the base-algorithm of ATLAS-ADA exploit a hint function $H_t(\cdot)$ to better exploit the pattern of the distribution challenge. We have the following regret guarantees on the output $\mathbf{w}_{t,i}$ of the base-algorithm.

**Lemma 9.** *Suppose the hint function $H_t : \mathcal{W} \mapsto \mathbb{R}$ is convex. The base-algorithm of* ATLAS-ADA *(Algorithm 4) with a step size $\eta_{t,i} > 0$ satisfies*

$$\sum_{t=1}^{T} \langle \nabla \widehat{R}_t(\mathbf{w}_{t,i}), \mathbf{w}_{t,i} - \mathbf{u}_t \rangle \leq 2\eta_i \sum_{t=1}^{T} \sup_{\mathbf{w} \in \mathcal{W}} \|\nabla \widehat{R}_t(\mathbf{w}) - \nabla H_t(\mathbf{w})\|_2^2 + \frac{\Gamma^2 + 2\Gamma P_T}{2\eta_i}$$

*for any comparator sequence $\{\mathbf{u}_t\}_{t=1}^{T}$ with $\mathbf{u}_t \in \mathcal{W}$. Moreover, $P_T = \sum_{t=2}^{T} \|\mathbf{u}_t - \mathbf{u}_{t-1}\|_2$ measures the variation of the comparator sequence.*

By Lemma 9 and taking the expectation over both sides, we can bound the base-regret as

$$\mathbb{E}_{1:T} \left[\sum_{t=1}^{T} R_t(\mathbf{w}_{t,i})\right] - \sum_{t=1}^{T} R_t(\mathbf{u}_t) \leq 2\eta_i \sum_{t=1}^{T} \mathbb{E}_t \left[\sup_{\mathbf{w} \in \mathcal{W}} \|\nabla \widehat{R}_t(\mathbf{w}) - \nabla H_t(\mathbf{w})\|_2^2\right] + \frac{\Gamma^2 + 2\Gamma P_T}{2\eta_i}.$$

Then, combining the meta- and base-regret, we can obtain that

$$\mathtt{term(a)} \leq 2\eta_i G_T + \frac{\Gamma^2 + 2\Gamma P_T}{2\eta_i} + \Gamma\sqrt{(\ln N + 2)(1 + G_T)}$$

for any base-algorithm $i \in [N]$, where $G_T = \sum_{t=1}^{T} \mathbb{E}_t \left[\sup_{\mathbf{w} \in \mathcal{W}} \|\nabla \widehat{R}_t(\mathbf{w}) - \nabla H_t(\mathbf{w})\|_2^2\right]$ measures the reusability of the history information.

It remains to tune the step size to make the bound tight. We can bound the term $G_t$ by

$$G_T = \sum_{t=1}^{T} \mathbb{E}_t \left[\sup_{\mathbf{w} \in \mathcal{W}} \|\nabla \widehat{R}_t(\mathbf{w}) - \nabla H_t(\mathbf{w})\|_2^2\right]$$

$$\leq \sum_{t=1}^{T} \mathbb{E}_t \left[2 \sup_{\mathbf{w} \in \mathcal{W}} \|\nabla \widehat{R}_t(\mathbf{w})\|_2^2 + 2 \sup_{\mathbf{w} \in \mathcal{W}} \|\nabla H_t(\mathbf{w})\|_2^2\right] \leq \frac{4TG^2K}{\sigma^2},$$

where the last inequality is due to the fact that $\sup_{\mathbf{w} \in \mathcal{W}} \|\nabla \widehat{R}_t(\mathbf{w}_t)\|_2 \leq (G\sqrt{K})/\sigma$ (shown as (28)) and the assumption that $\sup_{\mathbf{w} \in \mathcal{W}} \|\nabla H_t(\mathbf{w})\|_2 \leq (G\sqrt{K})/\sigma$. In such a case, we can show that the optimal step size $\eta_* = \frac{1}{2}\sqrt{\frac{\Gamma^2 + 2\Gamma P_T}{1 + G_T}}$ lies in the range $[\frac{1}{2}\sqrt{\frac{\Gamma^2\sigma^2}{\sigma^2 + 4TG^2K}}, \frac{1}{2}\sqrt{\Gamma^2 + 2\Gamma^2T}]$. So, we can construct the following step size pool to cover the optimal step size

$$\mathcal{H} = \left\{\frac{\Gamma\sigma}{\sqrt{\sigma^2 + 4TG^2K}} \cdot 2^{i-1} \mid i \in [N]\right\}$$

with $N = 1 + \lceil \frac{1}{2}\log_2\left((1 + 2T)(1 + 4TG^2K/\sigma^2)\right)\rceil$. Even better, due to the logarithmic construction of the pool, we claim that there must exist an $i_* \in [N]$ satisfies that $\eta_{i_*}/2 \leq \eta_* \leq \eta_{i_*}$. So, we further bound term (a) as

$$\mathtt{term(a)} \leq 3\sqrt{(1 + G_T)(\Gamma^2 + 2\Gamma P_T)} + \Gamma\sqrt{(\ln N + 2)(1 + G_T)}.$$

**Overall dynamic regret bound.** We can then use similar arguments in the proof of Theorem 2 to obtain the final dynamic regret bound. Combining term (a) and term (b) and noticing that $\{\mathbf{u}_t\}_{t=1}^T$ is the piecewise stationary comparator sequences with $P_T = \sum_{t=2}^T \|\mathbf{u}_t - \mathbf{u}_{t-1}\|_{t=1}^T \leq \Gamma T/\Delta$, we can bound the overall dynamic regret as

$$
\mathbb{E}_{1:T}\left[\mathbf{Reg}_T^{\mathbf{d}}\right] = \mathbb{E}_{1:T}\left[\sum_{t=1}^T R_t(\mathbf{w}_t)\right] - \sum_{t=1}^T R_t(\mathbf{w}_t^*)
$$

$$
\leq \mathbb{1}\{\Delta > 1\} \cdot 2B\Delta V_T + 3\sqrt{(1+G_T)(\Gamma^2 + 2\Gamma P_T)} + \Gamma\sqrt{(\ln N + 2)(1+G_T)}
$$

$$
\leq \mathbb{1}\{\Delta > 1\} \cdot 2B\Delta V_T + 3\Gamma\sqrt{\frac{2(1+G_T)T}{\Delta}} + 3\Gamma\sqrt{1+G_T} + \Gamma\sqrt{(\ln N + 2)(1+G_T)},
$$

where the last inequality is due to the fact that $\sqrt{a+b} \leq \sqrt{a} + \sqrt{b}$ for any $a, b \geq 0$. To obtain an upper bound of the above inequality, we consider the two following cases,

**Case 1**: $3\Gamma\sqrt{2(1+G_T)T} > 4BV_T$, in such a case, we can set $\Delta = \lceil \frac{3\Gamma\sqrt{2(1+G_T)T}}{4BV_T} \rceil$. Then, we can bound the dynamic regret by

$$
\mathbb{E}_{1:T}\left[\mathbf{Reg}_T^{\mathbf{d}}\right] \leq 3\left(9B\Gamma^2\right)^{\frac{1}{3}} \cdot V_T^{1/3} T^{1/3}(1+G_T)^{1/3} + (3 + \sqrt{2 + \ln N})\Gamma\sqrt{1+G_T}
$$

$$
= \mathcal{O}(V_T^{1/3} G_T^{1/3} T^{1/3} + \sqrt{1+G_T}).
$$

**Case 2**: $3\Gamma\sqrt{2(1+G_T)T} \leq 4BV_T$, in such a case, we can set $\Delta = 1$. Then, we have

$$
\mathbb{E}_{1:T}\left[\mathbf{Reg}_T^{\mathbf{d}}\right] \leq 3\Gamma\sqrt{2(1+G_T)T} + 3\Gamma\sqrt{1+G_T} + \Gamma\sqrt{(\ln N + 2)(1+G_T)}
$$

$$
\leq (6\Gamma(1+G_T)T)^{1/3}(12\Gamma BV_T)^{1/3} + 3\Gamma\sqrt{1+G_T} + \Gamma\sqrt{(\ln N + 2)(1+G_T)}
$$

$$
= \mathcal{O}(V_T^{1/3} G_T^{1/3} T^{1/3} + \sqrt{1+G_T}),
$$

where the second inequality is due to the inequality that $3\Gamma\sqrt{2(1+G_T)T} \leq 4BV_T$. We complete the proof by combining the two cases. $\qquad\square$

### F.3 Useful Lemmas in Analysis of Theorem 3

**Lemma 8.** *By setting the learning rate* $\varepsilon = \sqrt{\frac{\ln N + 2}{1 + \sum_{t=1}^T (\max_{i \in [N]}\{|\widetilde{R}_t(\mathbf{w}_{t,i}) - \widetilde{H}_t(\mathbf{w}_{t,i})|\})^2}}$, *the meta-algorithm of* ATLAS-ADA *(Algorithm 5) satisfies*

$$
\sum_{t=1}^T \sum_{j=1}^N p_{t,j} \widehat{R}_t(\mathbf{w}_{t,j}) - \sum_{t=1}^T \widehat{R}_t(\mathbf{w}_{t,i}) \leq 2\Gamma\sqrt{(\ln N + 2)\left(1 + \sum_{t=1}^T \sup_{\mathbf{w} \in \mathcal{W}} \|\nabla \widehat{R}_t(\mathbf{w}) - \nabla H_t(\mathbf{w})\|_2^2\right)}
$$

*for any* $i \in [N]$, *where* $\widetilde{R}_t(\mathbf{w}_{t,i}) = \widehat{R}_t(\mathbf{w}_{t,i}) - \widehat{R}_t(\mathbf{w}_{\mathtt{ref}})$ *and* $\widetilde{H}_t(\mathbf{w}_{t,i}) = H_t(\mathbf{w}_{t,i}) - H_t(\mathbf{w}_{\mathtt{ref}})$ *are the reference loss functions built upon an arbitrary reference point* $\mathbf{w}_{\mathtt{ref}} \in \mathcal{W}$.

*Proof of Lemma 8.* The key challenge in the proof is that the meta-algorithm is updated with the loss function $\widehat{R}_t(\mathbf{w}_{t,i})$ and the hint function $H_t(\mathbf{w}_{t,i})$, while we desire a regret that can scale with the gradient to make it compatible with the base-regret bound. To this end, we introduce the reference loss $\widetilde{R}_t(\mathbf{w}_{t,i}) = \widehat{R}_t(\mathbf{w}_{t,i}) - \widehat{R}_t(\mathbf{w}_{\mathtt{ref}})$ and $\widetilde{H}_t(\mathbf{w}_{t,i}) = H_t(\mathbf{w}_{t,i}) - H_t(\mathbf{w}_{\mathtt{ref}})$. It is easy to verify that the update procedure (4) with the original loss function is identical to that with the reference loss:

$$
p_{t,i} = \frac{\exp\left(-\varepsilon\left(\sum_{s=1}^{t-1} \widehat{R}_s(\mathbf{w}_{s,i}) + H_t(\mathbf{w}_{t,i})\right)\right)}{\sum_{i=1}^K \exp\left(-\varepsilon\left(\sum_{s=1}^{t-1} \widehat{R}_s(\mathbf{w}_{s,i}) + H_t(\mathbf{w}_{t,i})\right)\right)} = \frac{\exp\left(-\varepsilon\left(\sum_{s=1}^{t-1} \widetilde{R}_s(\mathbf{w}_{s,i}) + \widetilde{H}_t(\mathbf{w}_{t,i})\right)\right)}{\sum_{i=1}^K \exp\left(-\varepsilon\left(\sum_{s=1}^{t-1} \widetilde{R}_s(\mathbf{w}_{s,i}) + \widetilde{H}_t(\mathbf{w}_{t,i})\right)\right)}.
$$

As a consequence, we can bound the meta-regret by

$$\sum_{t=1}^{T}\sum_{j=1}^{N} p_{t,j}\widehat{R}_t(\mathbf{w}_{t,j}) - \sum_{t=1}^{T}\widehat{R}_t(\mathbf{w}_{t,i})$$

$$= \sum_{t=1}^{T}\sum_{j=1}^{N} p_{t,j}\widetilde{R}_t(\mathbf{w}_{t,j}) - \sum_{t=1}^{T}\widetilde{R}_t(\mathbf{w}_{t,i})$$

$$\leq \varepsilon \sum_{t=1}^{T}\left(\max_{i\in[N]}\{|\widetilde{R}_t(\mathbf{w}_{t,i}) - \widetilde{H}_t(\mathbf{w}_{t,i})|\}\right)^2 + \frac{\ln N + 2}{\varepsilon}$$

$$\leq 2\sqrt{\sum_{t=1}^{T}\left(\max_{i\in[N]}\{|\widetilde{R}_t(\mathbf{w}_{t,i}) - \widetilde{H}_t(\mathbf{w}_{t,i})|\}\right)^2 (\ln N + 2)}, \tag{32}$$

where the first inequality is due to Lemma 11 and the second inequality is by the step size setting of
$\varepsilon = \sqrt{(\ln N + 2)/(1 + \sum_{t=1}^{T} \max_{i\in[N]}\{|\widetilde{R}_t(\mathbf{w}_{t,i}) - \widetilde{H}_t(\mathbf{w}_{t,i})|\})}$.

Then, we show that the deviation between the loss function $\max_{i\in[N]}\{|\widetilde{R}_t(\mathbf{w}_{t,i}) - \widetilde{H}_t(\mathbf{w}_{t,i})|\}$ can be converted to the deviation between the gradient. Let $G_t(\mathbf{w}_{t,i}) = \widehat{R}_t(\mathbf{w}_{t,i}) - H_t(\mathbf{w}_{t,i})$, for any $i \in [N]$, we have

$$|\widetilde{R}_t(\mathbf{w}_{t,i}) - \widetilde{H}_t(\mathbf{w}_{t,i})| = |G_t(\mathbf{w}_{t,i}) - G_t(\mathbf{w}_{\mathbf{ref}})| = |\langle \nabla G_t(\boldsymbol{\xi}_{t,i}), \mathbf{w}_{t,i}\rangle|$$

$$\leq \Gamma\|\nabla\widehat{R}_t(\boldsymbol{\xi}_{t,i}) - \nabla H_t(\boldsymbol{\xi}_{t,i})\|_2 \leq \Gamma \sup_{\mathbf{w}\in\mathcal{W}}\|\nabla\widehat{R}_t(\mathbf{w}) - \nabla H_t(\mathbf{w})\|_2, \tag{33}$$

where the first equality is due to the mean value theorem and $\boldsymbol{\xi}_{t,i} = c_{t,i}\mathbf{w}_{t,i} + (1 - c_{t,i})\mathbf{w}_{\mathbf{ref}}$ with a certain constant $c_{t,i} \in [0,1]$. Plugging (33) into (32), we obtain that

$$\sum_{t=1}^{T}\sum_{i=1}^{N} p_{t,i}\widehat{R}_t(\mathbf{w}_{t-1,i}) - \sum_{t=1}^{T}\widehat{R}_t(\mathbf{w}_{t-1,i}) \leq 2\Gamma\sqrt{(\ln N + 2)\sum_{t=1}^{T}\sup_{\mathbf{w}\in\mathcal{W}}\|\nabla\widehat{R}_t(\mathbf{w}) - \nabla H_t(\mathbf{w})\|_2^2},$$

which completes the proof. $\qquad\square$

Then, we provide the proof of the regret of the base-algorithm for ATLAS-ADA.

**Lemma 9.** *Suppose the hint function $H_t : \mathcal{W} \mapsto \mathbb{R}$ is convex. The base-algorithm of* ATLAS-ADA *(Algorithm 4) with a step size $\eta_{t,i} > 0$ satisfies*

$$\sum_{t=1}^{T}\langle\nabla\widehat{R}_t(\mathbf{w}_{t,i}), \mathbf{w}_{t,i} - \mathbf{u}_t\rangle \leq 2\eta_i \sum_{t=1}^{T}\sup_{\mathbf{w}\in\mathcal{W}}\|\nabla\widehat{R}_t(\mathbf{w}) - \nabla H_t(\mathbf{w})\|_2^2 + \frac{\Gamma^2 + 2\Gamma P_T}{2\eta_i},$$

*for any comparator sequence $\{\mathbf{u}_t\}_{t=1}^{T}$ with $\mathbf{u}_t \in \mathcal{W}$. Moreover, $P_T = \sum_{t=2}^{T}\|\mathbf{u}_t - \mathbf{u}_{t-1}\|_2$ measures the variation of the comparator sequence.*

*Proof of Lemma 9.* We decompose the instantaneous loss of the base algorithm as,

$$\langle\nabla\widehat{R}_t(\mathbf{w}_{t,i}), \mathbf{w}_{t,i} - \mathbf{u}_t\rangle$$
$$\leq \underbrace{\langle\nabla\widehat{R}_t(\mathbf{w}_{t,i}) - \nabla H_t(\mathbf{w}_{t,i}), \mathbf{w}_{t,i} - \widehat{\mathbf{w}}_{t+1,i}\rangle}_{\texttt{term (a)}} + \underbrace{\langle\nabla H_t(\mathbf{w}_{t,i}), \mathbf{w}_{t,i} - \widehat{\mathbf{w}}_{t+1,i}\rangle}_{\texttt{term (b)}} + \underbrace{\langle\nabla\widehat{R}_t(\mathbf{w}_{t,i}), \widehat{\mathbf{w}}_{t+1,i} - \mathbf{u}_t\rangle}_{\texttt{term (c)}}.$$

For term (a), by Cauchy-Schwarz inequality and AM-GM inequality, we have

$$\texttt{term (a)} \leq \|\nabla\widehat{R}_t(\mathbf{w}_{t,i}) - \nabla H_t(\mathbf{w}_{t,i})\|_2\|\mathbf{w}_{t,i} - \widehat{\mathbf{w}}_{t+1,i}\|_2$$

$$\leq 2\eta_i\|\nabla\widehat{R}_t(\mathbf{w}_{t,i}) - \nabla H_t(\mathbf{w}_{t,i})\|_2^2 + \frac{1}{2\eta_i}\|\mathbf{w}_{t,i} - \widehat{\mathbf{w}}_{t+1,i}\|_2^2.$$

Then, we bound term (b) by Proposition 4.1 of Campolongo and Orabona [38],

$$\texttt{term (b)} \leq \frac{1}{2\eta_i} \left( \|\widehat{\mathbf{w}}_{t+1,i} - \widehat{\mathbf{w}}_{t,i}\|_2^2 - \|\widehat{\mathbf{w}}_{t+1,i} - \mathbf{w}_{t,i}\|_2^2 - \|\widehat{\mathbf{w}}_{t,i} - \mathbf{w}_{t,i}\|_2^2 \right).$$

The term (c) is bounded by Lemma 7 of Zhao et al. [19],

$$\texttt{term (c)} \leq \frac{1}{2\eta_i} \left( \|\mathbf{u}_t - \widehat{\mathbf{w}}_{t,i}\|_2^2 - \|\mathbf{u}_t - \widehat{\mathbf{w}}_{t+1,i}\|_2^2 - \|\widehat{\mathbf{w}}_{t,i} - \widehat{\mathbf{w}}_{t+1,i}\|_2^2 \right).$$

Combining the three upper bounds yields

$$\sum_{t=1}^{T} \langle \widehat{R}_t(\mathbf{w}_{t,i}), \mathbf{w}_{t,i} - \mathbf{u}_t \rangle$$

$$\leq 2\eta_i \sum_{t=1}^{T} \|\nabla \widehat{R}_t(\mathbf{w}_{t,i}) - \nabla H_t(\mathbf{w}_{t,i})\|_2^2 + \frac{1}{2\eta_i} \sum_{t=1}^{T} \left( \|\mathbf{u}_t - \widehat{\mathbf{w}}_{t,i}\|_2^2 - \|\mathbf{u}_t - \widehat{\mathbf{w}}_{t+1,i}\|_2^2 \right) + \frac{\Gamma^2}{2\eta_i}$$

$$\leq 2\eta_i \sum_{t=1}^{T} \sup_{\mathbf{w} \in \mathcal{W}} \|\nabla \widehat{R}_t(\mathbf{w}) - \nabla H_t(\mathbf{w})\|_2^2 + \frac{\Gamma^2 + 2\Gamma P_T}{2\eta_i},$$

which completes the proof. $\qquad \square$

## G Technical Lemmas

**Lemma 10.** *Let $A \in \mathbb{R}^{K \times K}$ be an invertible matrix and $\Delta \in \mathbb{R}^{K \times K}$ be a squared matrix such that $\|A^{-1}\Delta\|_2 \leq 1/2$. For $B = A + \Delta$, we have the following two claims:*

- *$B$ is an invertible matrix.*
- *$\|B^{-1} - A^{-1}\|_2 \leq 2\|A^{-1}\|_2^2 \cdot \|\Delta\|_2$.*

*Proof.* The proof follows the same arguments in Horn and Johnson [82, Page 381]. We first show that $B = A(I + A^{-1}\Delta)$ is an invertible matrix. Since $A$ is an invertible matrix by the assumption, it is sufficient to show that $I + A^{-1}\Delta$ is invertible. The statement is true since the spectral radius of $A^{-1}\Delta$ satisfies $\rho(A^{-1}\Delta) \leq \|A^{-1}\Delta\|_2 \leq 1/2$. The bounded spectral radius implies that $-1$ is not an eigenvalue of the matrix $A^{-1}\Delta$, and thus it is invertible.

Then, we can upper bound the norm

$$\|B^{-1} - A^{-1}\|_2 = \|A^{-1}\Delta B^{-1}\|_2 \leq \|A^{-1}\Delta\|_2 \|B^{-1}\|_2, \tag{34}$$

where the term $\|B^{-1}\|_2$ can be further bounded by

$$\|B^{-1}\|_2 \leq \|A^{-1}\|_2 + \|B^{-1} - A^{-1}\|_2 \leq \|A^{-1}\| + \|A^{-1}\Delta\|_2 \|B^{-1}\|_2.$$

Rearranging the above inequality, we have

$$\|B^{-1}\|_2 \leq \frac{\|A^{-1}\|_2}{1 - \|A^{-1}\Delta\|_2}. \tag{35}$$

Then, plugging (35) into (34) and rearranging the terms, we have

$$\|B^{-1} - A^{-1}\|_2 \leq \frac{\|A^{-1}\Delta\|_2 \|A^{-1}\|_2}{1 - \|A^{-1}\Delta\|_2} \leq \frac{\|A^{-1}\|_2^2 \|\Delta\|_2}{1 - \|A^{-1}\Delta\|_2} \leq 2\|A^{-1}\|_2^2 \|\Delta\|_2,$$

which completes the proof. $\qquad \square$

**Lemma 11** (Theorem 19 of Syrgkanis et al. [83]). *Let $\boldsymbol{\ell}_t \in \mathbb{R}^N$ be the loss vector taking $\ell_{t,i} \in \mathbb{R}$ as its $i$-th entry and $\mathbf{m}_t \in \mathbb{R}^N$ the hint vector taking $m_{t,i}$ as its $i$-th entry. The Hedge algorithm updating with $p_{t,i} \propto \exp\left( -\varepsilon \left( \sum_{s=1}^{t-1} \ell_{s,i} + m_{t,i} \right) \right)$ satisfies*

$$\sum_{t=1}^{T} \sum_{j=1}^{N} p_{t,j} \ell_{t,j} - \sum_{t=1}^{T} \ell_{t,i} \leq \frac{\ln N + 2}{\varepsilon} + \varepsilon \sum_{t=1}^{T} \|\boldsymbol{\ell}_t - \mathbf{m}_t\|_\infty^2$$

*for any $i \in [N]$, where $\varepsilon > 0$ is the step size.*