# OpenReview forum: "Adapting to Online Label Shift with Provable Guarantees"
_NeurIPS.cc/2022/Conference — NeurIPS 2022 Accept_

### Official Review · Reviewer_bK7W · 2022-07-08

**Rating:** 5
**Confidence:** 2
**Soundness:** 3 good
**Presentation:** 2 fair
**Contribution:** 2 fair

**Summary:**

This paper addresses the problem that the label distribution in testing data is different from training data. The problem is interesting and with practical interest. The authors propose to include the testing data in the training process in the goal to reduce error.

The proposed method has two stages: first selecting data and corresponding labels from the initial distribution; including unlabeled data from another distribution. If the label shift is not considered, would the model produce wrong prediction results?

The goal of the work is to make the model adaptive to the new label distribution. Is it a two way door design? That is, when there is no label distribution shift, the model performs as well as the vanilla one trained on the data following the initial distribution.

In Theorem 1, the constant \sigma >0 denotes the minimum singular value of confusion matrix. What is the assumption to make the minimum singular value of confusion matrix to be bigger than 0, instead of 0?

V_T measures the intensity of the label distribution shift. Is there any threshold that, if V_T is smaller than the threshold, the label distribution shift does not have effect about the performance of the vanilla model.

Could the method be applied to both classification and regression?

**Questions:**

If the label shift is not considered, would the model produce wrong prediction results?

Does the proposed model follow a two way door design? That is, when there is no label distribution shift, the model performs as well as the vanilla one trained on the data following the initial distribution.

In Theorem 1, the constant \sigma >0 denotes the minimum singular value of confusion matrix. What is the assumption to make the minimum singular value of confusion matrix to be bigger than 0, instead of 0?

V_T measures the intensity of the label distribution shift. Is there any threshold that, if V_T is smaller than the threshold, the label distribution shift does not have effect about the performance of the vanilla model.

Could the method be applied to classification and regression?

**Limitations:**

yes

**Strengths And Weaknesses:**

Strength:
-The problem is interesting and with practical interest.
-The paper is well written.

Weakness:
- The assumptions of the work are not clarified.
- The data in the experiments is not in large scale.

---

> ### Author Response · Authors · 2022-08-02
> **Response to Reviewer bK7W: addressing your main concerns (1/2)**
>
> Thanks for your helpful comments. We will address your concerns as follows. If your concerns are properly addressed, please consider updating your score for this submission.
>
> **Q1**: "If the label shift is not considered, would the model produce wrong prediction results?"
>
> **A1**: Yes.  Below we explain the reasons.
>
> - Theoretically, the optimal classifier (Bayes decision boundary) for a classification problem depends on the class prior $\mathcal{D}(y)$. Thus, when the class prior changes, the optimal classifier for the original distribution will become suboptimal if the label shift is not considered.
>
> - Empirically, in our experiments (see Table 1 and Table 3), the FIX model (which directly predicts the label using the vanilla models without considering label shift) is clearly inferior to the other algorithms (including our method and prior work [15]). Actually, even in the offline setting, it is necessary to consider label shift, as demonstrated in previous works [11,12].
>
> Thanks for your questions. We will add those discussions in the revised version.
>
> ---
>
> **Q2**: "Does the proposed model follow a two way door design? That is, when there is no label distribution shift, the model performs as well as the vanilla one trained on the data following the initial distribution."
>
> **A2**: Thank you for bringing this point up. Our method follows a two way door design as it can outperform other methods when label shift happens while performing as well as (even slightly better than) the vanilla model.
>
> We conducted additional experiments on the CIFAR10 dataset during the rebuttal period to support this claim. We report the averaged error (\%) of the compared algorithms when there is no label shift as follows,
>
> |     FIX    |    UOGD    |    ATLAS   |  ATLAS-ADA |
> |:----------:|:----------:|:----------:|:----------:|
> | 23.40±0.01 | 22.73±0.01 | 22.75±0.02 | 22.82±0.02 |
>
>
> The results show that our algorithms can be even slightly better than FIX (the vanilla model) when there is no label shift. Such a phenomenon is also observed in the previous work (please kindly refer to the "Constant" column of Table 1 in [15]), where the online algorithm also outperforms the initial model.
>
> We agree that if we have *infinite* samples in the offline stage, the vanilla model is the best choice when there is *no* label shift. Nevertheless, in practice, we can only collect finite samples in the offline stage. So the new (unlabeled) data that appear in the online stage are helpful to boost the model’s performance, especially when the model can be updated with a certain unbiased risk estimator.

---

> > ### Author Response · Authors · 2022-08-02
> > **Response to Reviewer bK7W: addressing your main concerns (2/2)**
> >
> >
> > **Q3**: "The assumptions of the work are not clarified." and "In Theorem 1, the constant $\sigma >0$ denotes the minimum singular value of confusion matrix. What is the assumption to make the minimum singular value of confusion matrix to be bigger than 0, instead of 0?"
> >
> > **A3**: Thanks for the question. The condition $\sigma>0$ is actually *equivalent* to the assumption of invertible confusion matrix $C\_{f\_0}$, and the latter one was stated in Lemma 1 of the main text. We note that this assumption is commonly adopted in the literature on label shift [11, 12, 15], and the condition can be easily satisfied if the model is well trained on the offline dataset. In the revised version, we will further justify the relationships between these two conditions and state assumptions more clearly.
> >
> > ---
> >
> > **Q4**: $V\_T$ measures the intensity of the label distribution shift. Is there any threshold that, if $V\_T$ is smaller than the threshold, the label distribution shift does not have effect about the performance of the vanilla model.
> >
> > **A4**: As discussed in A2 to Q2, the performance of the vanilla model can be improved by the online learning process even if there is no label shift. So, the model update is also necessary when the distribution change $V\_T$ is small.
> >
> > ---
> >
> > **Q5**: Could the method be applied to classification and regression?
> >
> > **A5**: Thank you for the constructive comments. Our paper studies the label shift problem in the multi-class classification scenario. It is an interesting direction to explore label shift in the regression problem. The problem is more challenging because the label space $\mathcal{Y}$ is no longer discrete. In our opinion, a possible solution is to employ the distribution matching method in [Nguyen et al., 2015] to estimate the label distribution $\mathcal{D}_t(y)$ and then use it to construct the risk estimator. We will take this as future work and add  it in the discussion.
> >
> > Reference: Nguyen et al. Continuous Target Shift Adaptation in Supervised Learning. In ACML 2015.
> >
> > ---
> >
> > **Q6**: "The data in the experiments is not in large scale."
> >
> > **A6**: Our experiments are conducted on various synthetic and real-world datasets, including the ArXiv dataset (296, 708 papers from 23 classes) and SHL dataset (107, 000 mobile sensor data from 6 classes). We are not sure whether this scale is “large” enough, but we believe all those experimental results have already demonstrated the effectiveness of our approach, as appreciated by Reviewer P75U "*experiments seem very extensive and show clearly what the authors want to show*". We hope this can address your concern, and we will be happy if our methods are deployed in even larger-scale scenarios such as industrial applications in the future.

---

> > > ### Author Response · Authors · 2022-08-08
> > > **Thanks for the review! Have we properly addressed the concerns?**
> > >
> > > We thank the reviewer again for the constructive feedback. Given that the author-reviewer discussion period is soon coming to an end, please let us know if our response has properly addressed the main concerns, and whether the reviewer has any further questions. Thanks!

---

> > > > ### Comment · Reviewer_bK7W · 2022-08-09
> > > > **thank authors for the clarification**
> > > >
> > > > I appreciate the clarification of authors. I would like to raise the score to weak accept since there is still some concern not well-addressed.

---

> > > > > ### Author Response · Authors · 2022-08-09
> > > > > **Thanks for your feedback!**
> > > > >
> > > > > Once again, we are grateful to the reviewer for the constructive feedback, and for all the time spent during the reviewing and the discussion periods. We will add those clarifications in the revised version to avoid potential misunderstandings. Thanks!

---

### Official Review · Reviewer_P75U · 2022-07-13

**Rating:** 9
**Confidence:** 4
**Soundness:** 4 excellent
**Presentation:** 4 excellent
**Contribution:** 4 excellent

**Summary:**

This work studies the problem of online adapting to label shift (e.g. where P(Y|X) is fixed, but the priors change) using the framework of online learning. It features a very strong theoretical analysis and excellent experimental evaluation. The presentation is great, and the Appendix even includes a brief intro to OCO!

**Questions:**

1) Remark 1; can you make it more formal what you mean by that you should focus on dynamic regret? Could you formally prove the advantage? Or is this already clear from the experiments?

2) Line 162 - Should this really be \mathbb{E}_{x \sim D_t} ? I have the feeling that it should not be dependent on t here.

3) "method still hardly suitable for the OCO framework" (line 195) - why? Can you be more precise why that is the case? CUrrently a bit vague.

4) I could not understand the remark in line 202 - where is it shown that the risk estimator is convex in expectation?

5) In line 260; the risk is used in the Hedge algorithm. Is there not a overfitting concern here? Should this risk not be estimated using validation data or fresh samples?

**Limitations:**

The authors reflect and conclude sufficiently on their own work - indeed high probability regret bounds would be nice to have as a future work, but the current work is already multi-contribution and definitely high quality.

**Strengths And Weaknesses:**

1) Very strong theoretical and seemingly sound analysis (I did not have time to go through the proofs) and various different variants of the algorithm are proposed and evaluated to show their pro's and cons.

2) A very clear comparison is made to related methods and its discussed in great detail what are the pro's and cons of the proposed method compared to the related work

3) The experiments seem very extensive and show clearly what the authors want to show

4) I was a bit sad to see the conclusions being postponed to the appendix; but the related work was sufficiently covered in the main text according to me.

I would maybe Transpose Table 1 if there is space to make it easier to read (easier to compare numbers vertically than horizontally).

---

> ### Author Response · Authors · 2022-08-02
> **Response to Reviewer P75U: Thank you for the appreciation!**
>
> Many thanks for your great appreciation and very constructive comments! In the following, we will address your questions. We will further improve the paper according to your suggestions.
>
> **Q1:** "...more formal what you mean by that you should focus on dynamic regret? Could you formally prove the advantage? Or is this already clear from the experiments?"
>
> **A1:** Thank you for the question. Briefly, dynamic regret can ensure algorithms be competitive with time-varying comparators rather than a fixed benchmark used in static regret. As such, it is very suitable for algorithm design under the online label shift scenario, provided that one can adequately model the comparator to depict the label shift. By contrast, simply optimizing static regret would lead to significantly undesired performance. Our experiments also illustrate the advantage of optimizing dynamic regret over static regret as the performance measure, as clearly shown in Table 1 (in the main text) and Table 3 (in Appendix B.1). We will add more discussions to formally state this advantage.
>
> ---
>
> **Q2** : "Line 162 - Should this really be $\mathbb{E}\_{x \sim \mathcal{D}\_t(\mathbf{x}|y)}$ ? I have the feeling that it should not be dependent on $t$ here."
>
> **A2**: You are correct, the distribution $\mathcal{D}\_t(\mathbf{x}|y)$ is actually independent of $t$. Under the label shift assumption, we have $\mathcal{D}\_t(\mathbf{x}|y) = \mathcal{D}\_0(\mathbf{x}|y)$ for all $t\in[T]$. We will change $\mathcal{D}\_t(\mathbf{x}|y)$ to $\mathcal{D}\_0(\mathbf{x}|y)$ for a more clear notation. Many thanks.
>
> ---
>
> **Q3**: "method still hardly suitable for the OCO framework" (line 195) - why? Can you be more precise why that is the case?"
>
> **A3**: The original risk estimator proposed by [15] can hardly be convex due to the 0/1 loss and the argmax operator. Furthermore, [15] also proposed a soft version of the risk estimator by replacing the 0/1 loss with a *smooth surrogate* and removing the argmax operator by a probability output. However, this soft version is still hard to be convex because it is directly established on the probability output, which would involve a normalization operation. The authors of [15] handle this issue by imposing an assumption on the convexity (please kindly refer to Assumption 3 in [15]). Thus, we find the soft risk estimator is unsuitable for the OCO framework. We will provide more explanations in the main paper and add more detailed introductions to their method in the appendix.
>
>
> ---
>
> **Q4**: "...where is it shown that the risk estimator is convex in expectation?"
>
> **A4**:  In Lemma 1, we show that the expectation of the risk estimator $\mathbb{E}\_{S\_t\sim\mathcal{D}\_t}[\hat{R}\_t(\mathbf{w})]$ is identical to the expected risk $R\_t(\mathbf{w}) = \mathbb{E}\_{(\mathbf{x},y)\sim \mathcal{D}\_t}[ \ell(f(\mathbf{w},\mathbf{x}),y)]$. Since $\ell(f(\mathbf{w},\mathbf{x}),y)$ can be taken as any convex surrogate loss w.r.t. the model $\mathbf{w}$, we can obtain that the risk estimator is convex in expectation. We will make this clear in the next version.
>
> ---
>
> **Q5**: "In line 260; the risk is used in the Hedge algorithm. Is there not an overfitting concern here? Should this risk not be estimated using validation data or fresh samples?"
>
> **A5**: Thank you for the constructive comments! We totally agree that there is a chance to promote the performance of our algorithm if a few labeled data are available in the online stage. But, we suspect that using those labeled data directly might be suboptimal, as the number of labeled data could be too few in the online setting. It would be an interesting and promising direction to study how to use labeled and unlabeled data together to promote performance, which is one of our major future works!
>
> We further want to clarify that the Hedge algorithm actually performs a certain “implicit” regularization, which might be suitable for mitigating overfitting. Specifically, the update rule of Hedge is identical to a linear optimization problem with a regularization term
> $$
> \mathbf{p}\_t = {\arg\\,\min}\_{\mathbf{p}\in\Delta}\sum\_{i=1}^{N} p\_{i}\sum\_{s=1}^{t-1}\left(\widehat{R}\_{s,i}(\mathbf{w}\_{s,i})\right) + \frac{1}{\varepsilon}\psi(\mathbf{p}),
> $$
> where the regularizer $\psi(\mathbf{p}) = \sum\_{i=1}^N p\_i\ln p\_i$ achieves its minimum value when $p\_i = 1/N$ for all $i\in[N]$. Such a regularizer encourages the algorithm to assign a small amount of weight to the base-algorithm even if it performs badly, which helps us to mitigate overfitting and is the key to deriving the regret bound. We will add discussion in this regard in the next version.
>
> **Q6**: "I would maybe Transpose Table 1 if there is space to make it easier to read (easier to compare numbers vertically than horizontally)."
>
> **A6**: Many thanks! We will modify Table 1 in the final version.

---

### Official Review · Reviewer_Z3UA · 2022-07-14

**Rating:** 6
**Confidence:** 3
**Soundness:** 3 good
**Presentation:** 3 good
**Contribution:** 2 fair

**Summary:**

This paper aims to minimize the dynamic regret (instead of the static regret in previous work) for the problem of online label shift (OLS). Specifically, to handle the non-stationarity environment and the lack of supervision, this paper constructs a new unbiased risk estimator that utilizes the unlabeled data and then proposes online ensemble algorithms to deal with the non-stationarity nature.  Besides, theoretical analyses about the dynamic regret bounds are provided. Experiments are conducted to validate the effectiveness of the proposed method.

**Questions:**

1. As claimed by the authors, [15] mainly utilizes the $0/1$ loss while this paper uses the surrogate loss in the construction of the unbiased risk estimator. However, I find the provided dynamic regret bound is about the surrogate loss while we originally care about the $0/1$ loss (just the same as the evaluation measure in experiments). Thus, a natural question is what is about the dynamic $0/1$ regret bound? Please give more discussions.
2. In this paper, the hypothesis set (or the model domain) $\mathcal{W}$ is assumed to be known but nearly impossible in practice. I also checked the experimental part but find no useful information. How to select or set $\mathcal{W}$? Please give more discussions.

**Limitations:**

While this paper is built on related work, I am afraid that the novelty may be a bit limited.

**Strengths And Weaknesses:**

I am not an expert in this area and may neglect related literature. Below are my comments.

### Originality
The novelty is limited. This work is mainly built on [15]. While [15] mainly aims to minimize the static regret for the OLS problem, this work aims to minimize the dynamic regret, which needs to consider more suitable online learning algorithms.

### Quality
Overall, this paper is sound. The proposed methods are natural and built on related work. The claims are supported by formal theoretical results. I have not carefully checked the proofs and the techniques seem standard. Experimental results are also provided to illustrate the effectiveness. Please see my main concerns in the Questions part.

### Clarity
This paper is well written and organized.

### Significance
This paper should be of interest to the community of label distribution shift, and online learning. Besides, the non-stationary setting may be very practical in some applications although this paper considers the simple OLS setting.

---

> ### Author Response · Authors · 2022-08-02
> **Response to Reviewer Z3UA: addressing your main questions (1/2)**
>
> Thanks for your appreciation of our work and careful review. In the following, we first highlight the novelty of our work and then address your major questions. We will improve our paper according to your comments.
>
> **Q1**: "The novelty is limited. This work is mainly built on [15]. While [15] mainly aims to minimize the static regret for the OLS problem, this work aims to minimize the dynamic regret..."
>
> **A1**: We respectfully disagree with your comment. Our work is *not* a simple extension of [15] from static regret to dynamic regret. *Please check Lines 80-90 of the main text for a summary of our technical contributions*. We here briefly summarize our novel contributions over [15].
>
> * **Novel risk estimator: a convex formulation.** We have proposed a convex formulation of the risk estimator, which better fits the OCO framework and can lead to provable guarantees. By contrast, the convexity of the risk estimator proposed by [15] is hard to be verified as it involves several non-convex operators.
>
>
> * **Novel online algorithm: advanced algorithm design and regret analysis.**  We highlight our algorithms are not direct applications of standard online learning methods but require many technical innovations.
>     + For dynamic regret (Theorem 2): our bound can adapt to variation of class prior $V\_T$ without any prior knowledge of $V\_T$. In contrast, without prior knowledge, existing online learning arts can only adapt to the variation of the comparator sequence $P_T$, which is hard to interpret for OLS problems.
>     + For adaptive result (Theorem 3): our bound is free from the smoothness assumption. By contrast, such an assumption is required by the previous work [18] to achieve the same adaptive dynamic regret bound. Our improvement roots in an advanced algorithm design based on the implicit update and a novel regret analysis.
>
> Finally, we note that [15] only provides static regret. In our paper, besides the dynamic regret bound (Theorem 2), we propose a more adaptive algorithm to exploit the structure of label shift (Theorem 3). All those results are obtained non-trivially. Please refer to Lines 80-90 (Technical Contribution), Remark 3, and Appendix D.2 for more comprehensive discussions. Many thanks!

---

> > ### Author Response · Authors · 2022-08-02
> > **Response to Reviewer Z3UA: addressing your main questions (2/2)**
> >
> >
> > **Q2**: "...the provided dynamic regret bound is about the surrogate loss while we originally care about the 0/1 loss... Thus, a natural question is what is about the dynamic 0/1 regret bound?"
> >
> > **A2**: We discuss the relationship between 0/1 loss and surrogate loss in the following three points.
> >
> > - First, we note that directly optimizing 0/1 loss is generally hard, not only in the online learning case but also in statistical learning theory. A (consistent) surrogate loss is usually adopted as the proxy in the optimization process. Thus, in this work, we use the dynamic regret over the surrogate loss as the performance measure to guide algorithm design.
> >
> > - Besides, although our dynamic regret is established over the surrogate loss, it still ensures an upper bound for the cumulative risk over 0/1 loss $\sum\_{t=1}^T R^{0/1}\_{t}(\mathbf{w}\_t)$. To see this, we have
> > $$
> >     \sum\_{t=1}^T R^{0/1}\_{t}(\mathbf{w}\_t) \leq \sum\_{t=1}^T R\_{t}(\mathbf{w}\_t) \leq \sum\_{t=1}^T R\_{t}(\mathbf{w}\_t^{\star}) + \mathrm{regret~bound},
> > $$
> > as the surrogate loss serves as an upper bound of the 0/1 loss. The above derivation usually appears in online classification literature to obtain mistake bounds (i.e., cumulative 0/1 loss); for example, [20, Theorem 12.1] uses hinge loss to achieve so.
> >
> >
> > - Finally, we emphasize that [15] seemingly presents a static regret bound over the 0/1 loss, but the result builds on the convexity assumption, which can not be verified (and actually can hardly be true). So we believe their regret guarantee is less attractive not only because they didn't take the non-stationarity into account but also due to the relatively strong and unrealistic assumption.
> >
> >
> > ---
> >
> > **Q3:** "...the hypothesis set (or the model domain) W is assumed to be known but nearly impossible in practice. I also checked the experimental part but find no useful information. How to select or set W?."
> >
> > **A3:** In experiments, for simplicity, we choose the decision domain $\mathcal{W}$ as a ball with a fixed diameter for each dataset (all the methods use the same decision domain for fair comparison; the diameter is properly set according to the complexity of the dataset). We will state it in the main text to make this clear.
> >
> > Theoretically, knowing the diameter of the feasible domain is standard for classic online learning, so as the previous works for dynamic regret minimization. Our current work inherits such an assumption. Admittedly, it would be much appreciated if the algorithm could work without knowing the upper bound of the domain diameter. Building upon the recent advance in non-stationary online learning [Luo et al., Section 5.2], we believe our method can be extended to the aforementioned scenario. Specifically, by designing a more sophisticated online ensemble structure to handle the uncertainty of the non-stationary environments simultaneously and explore the range of comparator sequences, we can achieve a so-called “comparator-adaptive” dynamic regret guarantee, bypassing the requirement of knowing the diameter. We will add this discussion in the next version. Many thanks!
> >
> > *Reference*: Haipeng Luo, Mengxiao Zhang, Peng Zhao, and Zhi-Hua Zhou. Corralling a Larger Band of Bandits: A Case Study on Switching Regret for Linear Bandits. In COLT 2022.

---

### Meta-Review · Area_Chair_9n9J · 2022-08-30

**Recommendation:** Accept
**Confidence:** Less certain

**Metareview:**

This paper considers online learning under a label shift scenario: an initial model is trained  on labeled data from the first distribution, and then the learner receives unlabeled data from shifting target distributions in rounds. The assumption here is that the underlying label-conditional densities (d(x|y)) do not change, only the "weighting" of the various labels changes between the rounds. This submission introduces the notion of dynamic regret for this setting where the learner is round-wise compared to the best predictor for the round-specific task. The submission provides an algorithm and bounds on this dynamic regret (that, naturally, involves a term measuring the amount of label-weighting-shift) as well as empirical evaluations of the proposed algorithm (both on synthetic and real datasets).

Overall this appears as a well-rounded submission on a problem setup that is clearly relevant to the NeurIPS community.

Since data is provided in batches per learning round, it would be appropriate to also compare to life-long learning setups, algorithms and guarantees, rather than just to online learning. It seems that the studied framework is more commonly referred to as life-long learning, but this connection and the corresponding literature is entirely ignored in this submission. This should be fixed/clarified before publication.

**Award:**

No

---

### Decision · Program_Chairs · 2022-09-14

Accept